# Open Boundary Conditions for Atmospheric Large Eddy Simulations and the Implementation in DALES4.4

Franciscus Liqui Lung[1], Christian Jakob[1], A. Pier Siebesma[2,3], and Fredrik Jansson[2]

[1]Monash University, Melbourne, Victoria, Australia
[2]Delft University of Technology, Delft, The Netherlands
[3]The Royal Netherlands Meteorological Institute, De Bilt, The Netherlands

**Correspondence:** Franciscus Liqui Lung (franciscus.liquilung@monash.edu)

**Abstract.** Open boundary conditions were developed for atmospheric large eddy simulation (LES) models and implemented into the Dutch Atmospheric Large Eddy simulation model. The implementation was tested in a "Big Brother"-like setup, in which the simulation with open boundary conditions was forced by an identical control simulation with periodic boundary conditions. The results show that the open boundary implementation has minimal influence on the solution. Both the mean state and the turbulent structures are close to the control simulation and disturbances at the in- and outflow boundaries are negligible. To emulate a setup in which the LES is coupled to a coarser model, the influence of coarse boundary input was tested by smoothing the output of the periodic control simulation both temporally and spatially before feeding it as input to the simulation with open boundary conditions. When smoothing is applied over larger/longer spatial/temporal scales, disturbances start to form at the inflow boundary and an area exists where turbulence needs to develop. Adding synthetic turbulence to the smoothed input reduces the size of this area and the magnitude of the disturbances.

## 1 Introduction

Large eddy simulation (LES) is a numerical simulation tool used to study turbulent motions in the atmospheric boundary layer (ABL). Employing resolutions ranging from $1 - 100$m, the largest turbulent eddies containing most turbulent kinetic energy (TKE) are resolved, whereas the effects of smaller unresolved eddies are parameterised. With most of the TKE being resolved, LES has the advantage over coarser limited area models (LAMs) when it comes to representing the effects of boundary layer turbulence. This advantage comes at the cost of domain size and/or simulation time. Idealised ABL studies using LES started in the late sixties/early seventies (e.g. Lilly, 1966; Deardorff, 1972; Sommeria, 1976). Traditionally, LES was mainly used to study ABLs with idealized homogeneous forcings, employing periodic lateral boundary conditions (LBCs).

With the increase in computational power, the use of LES has shifted from idealised cases to more complex and realistic scenarios. Some examples are the simulation of urban areas (e.g. Giometto et al., 2016; Kurppa et al., 2018), windfarms (e.g. Mehta et al., 2014) and very large case studies pushing towards domain sizes of $\mathcal{O}(1000\text{km})$ (e.g. Schalkwijk et al., 2015; Heinze et al., 2017). For the latter it is especially important to capture the heterogeneity present in the domain. Periodic LBCs are by definition not suited for this (Moeng et al., 2007). Furthermore, it is often desired to couple LES to a regional weather model to transfer large-scale atmospheric structures. For these reasons, it is desirable to have open LBCs in place. Ideally,

open LBCs allow the prescription of variables at inflow boundaries and propagate variables unperturbed out of the domain at outflow boundaries. Having the ability to use open BCs makes an LES model much more versatile in simulating a range of phenomena, especially over heterogenous terrain. While periodic BCs can sometimes be used to study large-scale phenomena over such terrain, the large domains required to do so quickly become computationally prohibitive.

There is no consensus on the "best" implementation of open boundary conditions for anelastic turbulent flow. In 1991 two mini-symposia were unsuccessfully dedicated to this topic and the effort was summarized as a frustrating one (Sani and Gresho, 1994). A popular choice is an outflow condition based on the radiation condition of Sommerfeld (1949). The radiation BC states that waves generated in the interior of the domain should propagate outwards with no reflections at the boundaries. It takes the form of a propagating wave and replaces the Navier-Stokes equations at the boundaries. The difficulty lies in determining the phase speed of the wave, which is required for applying the radiation boundary condition. Different implementations for the phase speed have been defined (e.g. Orlanski, 1976; Klemp and Wilhelmson, 1978; Hedley and Yau, 1988). Orlanski (1976) uses a variable phase speed that is defined upwind of the boundary and propagated to the boundary. The results of a 2D test case show that the implementation works well and results in minimal reflections. Klemp and Wilhelmson (1978) use radiation LBCs in their 3D storm model and evaluate their influence in a 2D version of the model. They define their phase speed as a constant plus the local boundary-normal velocity component. Using a similar test setup as Orlanski (1976), Klemp and Wilhelmson (1978) show that their implementation is capable of producing realistic results. They do note that the results are sensitive to the choice of the fixed part of the phase speed. Hedley and Yau (1988) compare the implementations of Orlanski (1976) and Klemp and Wilhelmson (1978) with their new implementation that is a hybrid version of the implementation of Orlanski (1976). They conclude that their hybrid implementation is superior to both. Craske and Van Reeuwijk (2013) give a summary of open BCs for incompressible turbulent flows and state that a radiation outflow condition results in the least amount of distortion for convection dominated flows. Incompressible LES models such as PALM (Maronga et al., 2015, 2020) and MESO-NH (Lac et al., 2018) and the fully compressible WRF-LES (Skamarock et al., 2021) have the option to use radiation boundary conditions for the boundary-normal velocity components based on one of the previously mentioned implementations. For the other variables (homogeneous) Neuman BCs are often used at outflow boundaries, which specify the boundary-normal derivative. For inflow conditions Dirichlet(-like) boundary conditions are common, which specify the fields at the boundary. The implementation of open boundary conditions in these LES models is summarised in Table 1.

**Table 1.** Summary of the open boundary implementations in the mentioned LES models.

| Model | Boundary-normal velocity components | Boundary-tangential velocity components and cell-centered variables | Phase velocity definition | Relaxation zone | References |
|---|---|---|---|---|---|
| PALM | Dirichlet inflow, radiation outflow | Radiation for velocity, other variables not described | Based on Orlanski (1976), averaged laterally | Yes | Maronga et al. (2015) |
| | Prescribed | Prescribed | Not applicable | No | Maronga et al. (2020); Kadasch et al. (2021) |
| MESO-NH | Radiation on perturbed fields | Weighted Dirichlet for inflow, Neumann (extrapolated) for outflow | Based on Carpenter (1982) | No | Lafore et al. (1998); Lac et al. (2018) |
| WRF-LES | Radiation on perturbed fields | Different definition for the boundary-normal flux term | Based on Klemp and Wilhelmson (1978) | No | Skamarock et al. (2021) |
| | Prescribed | Prescribed | Not applicable | Yes | |
| ICON | Prescribed | Prescribed | Not applicable | Yes | Heinze et al. (2017) |

The implementation of open LBCs make it possible to nest LES within both itself and mesoscale models (e.g. Moeng et al., 2007; Zhu et al., 2010; Talbot et al., 2012; Mazzaro et al., 2017; Heinze et al., 2017; Kadasch et al., 2021; Mirocha et al.,

2014). These studies used prescribed boundary conditions instead of radiation BCs to nest their LES. Prescribed boundary conditions are a Dirichlet boundary condition, where the LBCs of the child simulation are directly prescribed by the parent simulation. These type of prescribed LBCs are similar to what is used in the mesoscale modelling community and are intuitive to implement. Dirichlet boundary conditions are however known to create reflections and perturbations at outflow boundaries for turbulent flows (e.g. Wesseling, 2009; Ol'shanskii and Staroverov, 2000). For this reason Moeng et al. (2007); Zhu et al. (2010); Heinze et al. (2017) use a relaxation zone in combination with a prescribed boundary condition, in which the fields near the boundary are nudged towards the boundary values to dampen any numerical noise due to the LBCs. Moeng et al. (2007) use WRF-LES to conduct two-way nested simulations of LES nested within LES. They conclude that the nesting works well for LES within LES but state the challenges that will arise for both one-way and two-way nesting of LES within a mesoscale model. Mesoscale models will have different vertical profiles due to their turbulent transport parametrisations in the PBL as opposed to the 3D resolved turbulence of LES. Furthermore, since mesoscale models are non-turbulence resolving, the lack of turbulence at inflow boundaries will result in a spinup area required for turbulence to develop. The spinup area is further increased by the implementation of a relaxation zone, as it does not only dampen numerical artifacts but turbulence as well. Zhu et al. (2010) tested both a one-way and two-way nested setup with WRF-LES being forced by National Centers for Environmental Prediction (NCEP) reanalysis data. They found that the relaxation zone in the outermost model is able to mitigate potential problems with the large resolution jump between the coarsest WRF-LES domain and the NCEP data set. However, they found that the cloud fields can be strongly modulated by mesoscale organisation, especially in high wind conditions where the clouds align with the mean wind direction. They found little benefit of two-way nesting over one-way nesting. Talbot et al. (2012) coupled WRF-LES within WRF in a realistic one-way nesting setup using 3 LES domains with increasing resolution in 3 mesoscale domains. They found that the use of a nested LES setup mainly improves the surface fluxes and near surface fields, but the bulk ABL dynamics such as boundary layer height of the mesoscale models agreed better with observations. They also found that the initial and boundary forcings were most important for the results and had a much bigger influence than the choice of subgrid scheme. Mazzaro et al. (2017) studied the effect of unresolved mesoscale flows on LES. They forced three similar LES domains with different resolution mesoscale simulations. They test their results both with and without the addition of the cell perturbation method of Muñoz-Esparza et al. (2014, 2015) and find that the LES is capable to overcome erroneous features in the mesoscale output. The cell perturbation scheme helps to greatly reduce the distance required for turbulence to develop, especially for the coarser mesoscale forcings. The best results are obtained with the highest resolution mesoscale model. Heinze et al. (2017) used a one-way nesting approach to employ realistic LES over Germany. Three domains were used to step down from 625m horizontal resolution to 156m with a constant grid refinement factor of two. They compared their results to the observations of the HD(CP$^2$) campaign and conclude that when it comes to small-to-mesoscale variablity the use of LES drastically improves the results compared to their reference mesoscale model COSMO. PALM has also recently implemented an option for offline nesting within COSMO (Kadasch et al., 2021). They employ prescribed boundary conditions and impose synthetic turbulence in addition to the boundary fields. At the moment there is no relaxation zone implemented, but they do note that this might change in the future. In their test cases, they find that the boundary input has the largest impact on the main flow structures. Flow and updrafts rapidly develop with the help of the

synthetic turbulence routine. Fully developed turbulence was found after two to three times the distance corresponding to the eddy turnover time.

In this research we develop a set of open LBCs for anelastic LES and implement it in the Dutch Atmospheric Large Eddy Simulation model (DALES). The goal of the paper is threefold. First, we will give a clear and extensive description of the open LBCs developed in this research. Second, we will show the influence of the LBCs on the mean fields and turbulent characteristics. Third, we will see how, in an idealized setup, the results depend on the temporal and spatial resolution of the input data, as one would encounter when embedding the LES in a coarser, non-turbulence resolving, LAM. The LBCs are developed to minimize reflections and the area needed for turbulence to develop and to allow for potential future one-way nesting with coarser LAMs. To minimise reflections, the outflow boundary conditions will be based on the radiation boundary condition of Sommerfeld (1949) and for the inflow boundary a new set of Robin boundary conditions will be derived. To allow for one-way nesting with coarser LAMs, the open LBCs will be developed such that they allow time varying input. The LBCs are tested with a simplistic dry convective case in a "Big Brother" like setup (Denis et al., 2002). This allows us to single out the influence that the LBCs have on the fields in the interior of the domain. To study the influence of the spatial and temporal resolution of the boundary input data, the turbulence in the input data is filtered both in space and time simultaneously. This allows us to study the influence of the open LBCs in a setup where the LES is coupled to a non-turbulence resolving model and quantify the influence of the spatial and temporal resolution ratios between the parent and child model. We will investigate how long it takes for turbulence to fully develop. Furthermore, the influence of synthetic turbulence on generating inflow turbulence is explored.

## 2   Boundary condition implementation

This section will describe the implementation of the open boundary conditions in the Dutch Atmospheric Large Eddy Simulation (DALES) model (Heus et al., 2010). The presented open boundary implementation is applicable to any incompressible atmospheric LES and except for the discussion about mass conservation, could also be used for fully compressible LES. DALES solves the anelastic Navier-Stokes equations on a staggered Arakawa-C grid. The prognostic variables are the three velocity components $(u, v, w)$, liquid potential temperature $(\theta_l)$, total water specific humidity $(q_t)$, the rain water specific humidity $(q_r)$, the rain droplet number concentration $(N_r)$, the subfilter scale turbulence kinetic energy $(e)$ and up to 100 active or passive scalars. Appropriate boundary conditions are required for all the prognostic variables at the resolution of the simulation. The velocity components are located at their respective cell faces and the rest of the variables at the cell centres. The boundary is defined as the cell faces of the outermost grid cells. Therefore, the boundary-normal velocity components are located at the boundary, whereas the other variables are located offset from the boundary. If the boundary input is not at the same time intervals as the simulation, the input data is linearly interpolated in time to the model time. First, the implementation for the boundary-normal velocity components will be given and conservation of mass will be discussed. Second, the implementation for the other variables is described. Third, the algorithm used to add synthetic turbulence at the boundaries will be discussed.

## 2.1 Boundary-normal velocity components

The boundary condition for the boundary-normal velocity components depends on whether the cell is an in- or outflow cell. An inflow cell for the boundary-normal velocity component is defined as $\mathbf{u}^B \cdot \hat{n} < 0$, where $\mathbf{u}^B$ is the input velocity vector specified at the boundary, given by external data, and $\hat{n}$ the outward pointing boundary normal unit vector. An outflow cell is defined by $\mathbf{u}^B \cdot \hat{n} \geq 0$.

### 2.1.1 Outflow

The outflow boundary condition is based on the Sommerfeld radiation boundary condition (Sommerfeld, 1949), which states that disturbances should only be advected out of the domain with no reflections. The radiation boundary condition takes the form of a single propagating wave.

$$\frac{\partial u_n}{\partial t} = \begin{cases} -\frac{U}{\rho}\frac{\partial \rho u_n}{\partial n} + \epsilon, & \text{for lateral boundaries} \\ -\frac{U}{\rho}\frac{\partial \rho u_n}{\partial n} + g\frac{\theta - \langle\theta\rangle}{\langle\theta\rangle} + \epsilon, & \text{for top boundary} \end{cases} \tag{1}$$

In Eq. (1) $u_n$ is the boundary-normal velocity component, $U$ the advection/phase speed of the disturbances, $\partial/\partial n$ the boundary-normal derivative, $\rho$ the reference density profile used by DALES, $\theta$ the potential temperature, $g$ the gravitational acceleration, $\epsilon$ a correction factor required to conserve mass, which will be explained in more detail in Sect. 2.1.3 and $\langle\rangle$ denotes a horizontal slab average. For the vertical component at the top boundary the buoyancy force is added which works as a damping factor for the top boundary in stably stratified flows. The time derivative is discretised using DALES' third order Runga-Kutta scheme (Heus et al., 2010). The spatial derivative is discretised using a first order upwind scheme.

$$\left.\frac{\partial u_n}{\partial n}\right|_i \approx \begin{cases} \frac{u_i - u_{i-1}}{\Delta x_n}, & \text{for } u_B \geq 0 \\ \frac{u_{i+1} - u_i}{\Delta x_n}, & \text{for } u_B < 0 \end{cases} \tag{2}$$

For non-dispersive waves with a phase speed equal to $U$, the 1-D case of Eq. (1) without the correction factor $\epsilon$ will not generate any reflections. In the case of atmospheric simulations, which is a dispersive system, the transport velocity $U$ needs to be chosen carefully, such that reflections are minimised. Popular implementations for the phase speed are given by Orlanski (1976) used by PALM (Maronga et al., 2015, 2020) and by Klemp and Wilhelmson (1978) used in MESO-NH (Lafore et al., 1998; Lac et al., 2018) and WRF-LES (Skamarock et al., 2021). Here we will use a slightly adjusted version of the implementation given by Hedley and Yau (1988). The implementation of Hedley and Yau (1988) is a hybrid version of the implementation given by Orlanski (1976) and is shown to work better. Similar to Orlanski (1976), the velocity field and tendencies upstream of the boundary at the previous time step are used to define the local phase speed, which is then propagated to the boundary for the next time step. Additionally, Hedley and Yau (1988) set a fixed lower limit for the phase speed. We will set the lower limit to

the boundary input normal velocity component, $u_n^B$.

$$U^* = U|_{x_n - \hat{x} \cdot \hat{n} \Delta x_n}^{t - \Delta t} = \left\langle -\rho \frac{\partial u_n}{\partial t} \left( \frac{\partial \rho u_n}{\partial n} \right)^{-1} \right\rangle^{\text{int}},$$

$$U = \begin{cases} u_n^B, & \text{if } |U^*| \leq |u_n^B| \\ U^*, & \text{if } |u_n^B| < |U^*| < \frac{\Delta x_n}{\Delta t} \\ \text{sign}(U^*) \frac{\Delta x_n}{\Delta t}, & \text{if } |U^*| \geq \frac{\Delta x_n}{\Delta t} \end{cases} \tag{3}$$

In Eq. (3) $t - \Delta t$ denotes the previous time step and $x_n - \hat{x} \cdot \hat{n} \Delta x_n$ the location one gridsize upstream of the boundary. To avoid large fluctuations in the phase speed due to local gradients, the phase speed is averaged over the horizontal dimension perpendicular to the boundary vector over a distance of $\Delta x^{\text{int}}$ (north and south boundaries) or $\Delta y^{\text{int}}$ (west and east boundaries), denoted by $< >^{\text{int}}$. This is similar to PALM, which averages laterally over the entire boundary (Maronga et al., 2015). For stability reasons the upper bound of the phase velocity is set to the CFL condition. Equation (3) is discretised using a first order upwind scheme Eq. (2).

### 2.1.2 Inflow

For inflow cells the boundary-normal velocity at the boundary $u_n$ is nudged towards the input value $u_n^B$ with a relaxation time scale equal to the integration time scale used by DALES ($\Delta t$). The discretisation of the time derivative is given by the third-order Runga-Kutta scheme used by DALES (Heus et al., 2010).

$$\frac{\partial u_n}{\partial t} = \frac{u_n^B - u_n}{\Delta t} + \epsilon \tag{4}$$

### 2.1.3 Conservation of mass

The use of radiation boundary conditions means that continuity is not guaranteed and a correction factor, $\epsilon$ needs to be added. Hedley and Yau (1988) enforce that the height integrated mass flux through each boundary does not change in time. This limits however the functionality for time-varying wind fields, in which inflow boundaries can become outflow boundaries and vice versa. Here we derive a correction term that forces the mass flux through the boundary to the boundary input on a defined length scale. This allows the wind field to change in magnitude and direction over time. To conserve mass the following constrains are imposed.

1. The input boundary-normal velocity components integrated over the lateral and top boundaries S(B) satisfy the continuity equation conform to the reference density profile used by DALES.

$$\iint\limits_{S(B)} \rho \mathbf{u}^B \cdot \hat{n} \, \mathrm{d}S = 0 \tag{5}$$

2. The lateral and top boundaries are subdivided into patches $S^{\text{int}}$ defined by $\Delta y^{\text{int}}$ and $\Delta z$ for the west and east boundaries, $\Delta x^{\text{int}}$ and $\Delta z$ for the north and south boundaries and $\Delta x^{\text{int}}$ and $\Delta y^{\text{int}}$ for the top boundary. We enforce that the mass

flux integrated over each patch equals the mass flux given by the input velocities integrated over the same patch.

$$\iint\limits_{S^{\text{int}}} \rho\mathbf{u}\cdot\hat{n}\,\mathrm{d}S = \iint\limits_{S^{\text{int}}} \rho\mathbf{u}^B\cdot\hat{n}\,\mathrm{d}S \tag{6}$$

To obtain the correction factor $\epsilon$, we define $\epsilon$ to be constant (in space) within a single integration patch $S^{\text{int}}$, but can differ between patches. To obtain an expression for the correction term on a particular integration patch $\epsilon\left(S^{\text{int}}\right)$, we take the time derivative of Eq. (6). Further, we define $\frac{\partial\tilde{u}_n}{\partial t} = \frac{\partial u}{\partial t} - \epsilon$ as the tendency from either Eq. (1) or (4) minus the correction term. Within DALES the tendencies for the boundary-normal velocities are first calculated without the correction term. These tendencies are then used to calculate the correction term $\epsilon$ for each integration patch using Eq. (7). The correction factor is then added to the tendencies before applying them to make sure mass is conserved.

$$\epsilon\left(S^{\text{int}}\right) = \frac{\iint_{S^{\text{int}}} \rho\left(\frac{\partial u_n^B}{\partial t} - \frac{\partial\tilde{u}_n}{\partial t}\right)\mathrm{d}S}{\iint_{S^{\text{int}}} \rho\,\mathrm{d}S} \tag{7}$$

The correction factor $\epsilon$ can be physically interpreted as the correction required to force the mass flux through the integration patch $S^{\text{int}}$ to the mass flux integrated over the patch as given by the input. Since the constrain is set on the integrated quantity, fluctuations smaller than the set integration patch are conserved. Smaller values for $\Delta x^{\text{int}}$ and $\Delta y^{\text{int}}$ impose more strict boundary conditions, with Dirichlet conditions in the limit where $\Delta x^{\text{int}} = \Delta x$ and $\Delta y^{\text{int}} = \Delta y$. When used in a nested simulation, $\Delta x^{\text{int}}$ and $\Delta y^{\text{int}}$ could be set to the gridsize used by the parent model. In this setup the total mass flux through a parent cell at the boundary of the child model (DALES) is conserved, while the child model is free to generate turbulence on smaller scales. This is illustrated in 2D in Fig. 1 in which the blue cells correspond to the parent model and have a resolution of $\Delta x^{\text{parent}}$ and the brown cells to the child model (DALES).

The role of the correction term is to conserve mass integrated over the domain, such that the pressure solver, which needs to find a solution that conserves mass locally, can find a solution. It is possible to implement the tendency from the correction factor as a non-homogeneous Neumann boundary condition for the modified pressure (defined in Heus et al., 2010) $\frac{\partial\pi}{\partial n} = -\epsilon$, such that all the tendencies as a result of the continuity requirement are together. We chose however, to add the term in the equations for the boundary-normal velocity components and use homogeneous Neumann boundary conditions for the modified pressure $\frac{\partial\pi}{\partial n} = 0$, because this allows us to keep using the Fourier pressure solver present in DALES (Heus et al., 2010), by using cosine basis functions only.

At the moment the vertical length scale of the integration patch is fixed to the vertical grid resolution. This allows for a straightforward implementation when using stretched vertical grids. We have also experimented with setting the vertical length scale of the integration patch to the domain height. This couples the boundary layer with the column above the inversion layer and gave unwanted results. In the future the implementation can be extended to allow for a variable vertical integration length scale as well.

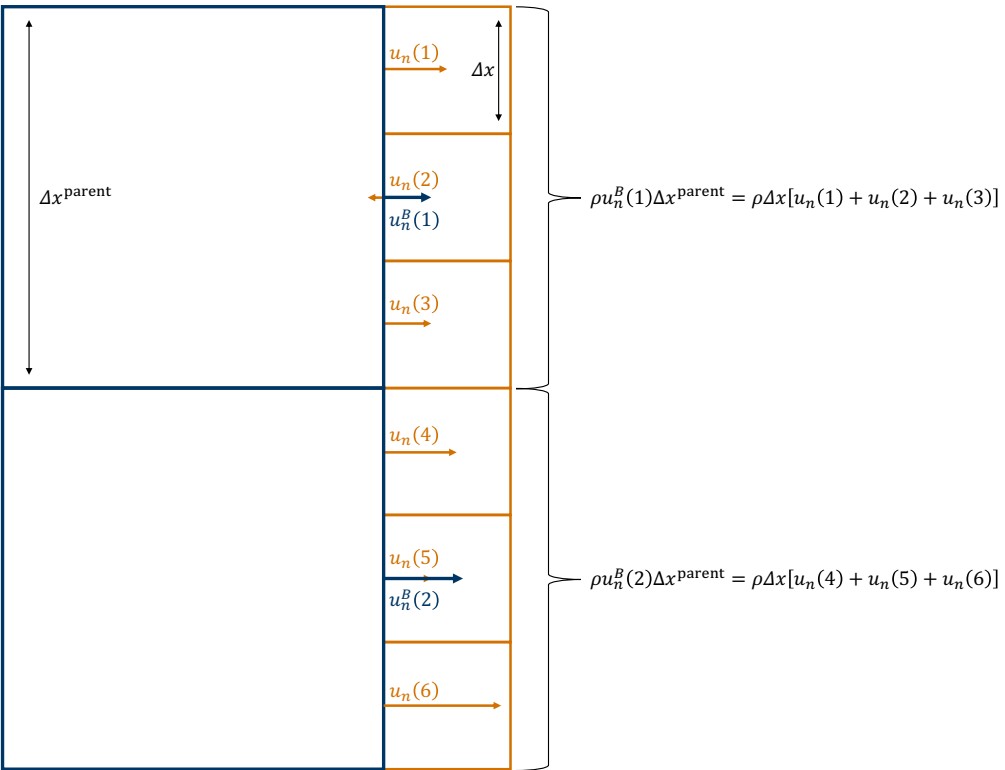

**Figure 1.** 2D illustration of a nested setup in which the integration length scales are set to the gridsize of the parent model. In this setup the mass flux through a parent cell (blue) at the boundary of the child model (brown) is conserved, while the child model is free to generate turbulence on smaller scales.

## 2.2 Boundary-tangential velocity components and cell-centered variables

This section will discuss the boundary conditions for the cell-centered variables and the tangential velocity components. These variables are not computed at the boundary. Instead, ghost cells are used together with a second order central discretisation to determine the behaviour of the variable at the boundary. The implementation is different for in- and outflow boundaries. For the cell-centered variables and tangential velocity components, a boundary is defined as inflow if $\mathbf{u} \cdot \hat{n} < 0$ and as outflow

otherwise. Note that this is different from the definition for the boundary-normal velocity components, where the nature of the boundary is determined by the input velocity $u_n^B$. These two can differ for outflow boundaries when the advection velocity is low and turbulence strong enough to reverse the local flow direction, as the radiation boundary condition does not enforce outflow on the local scale.

### 2.2.1 Outflow

For outflow cells homogeneous Neumann conditions, Eq. (8), are specified at the lateral boundaries.

$$\frac{\partial \psi}{\partial n} = 0 \tag{8}$$

In Eq. (8) $\psi$ is any of the cell centered variables ($\theta_l$, $q_t$, $q_r$, $N_r$, $e$) or tangential velocity components. At the top of the domain Neumann boundary conditions are set, which take the slab averaged vertical derivative into account,

$$\frac{\partial \psi}{\partial z} = \frac{\partial \langle \psi \rangle}{\partial z}, \tag{9}$$

in which $<>$ denotes a slab average. The decision to use homogeneous Neumann boundary conditions for all but the boundary-normal velocity components has been based on the results of Sani and Gresho (1994) and Craske and Van Reeuwijk (2013). Sani and Gresho (1994) state that Neumann boundary conditions tend to produce less perturbations in comparison to a boundary condition on the variable itself (Dirichlet). Setting homogeneous Neumann conditions for the boundary-normal velocity components results in an ill-posed system with fluctuations in the pressure field and is not suited for turbulent flows (Sani and Gresho, 1994; Craske and Van Reeuwijk, 2013).

### 2.2.2 Inflow

For inflow boundaries, Dirichlet boundary conditions are a common choice (e.g. Maronga et al., 2015; Lac et al., 2018). However, for flows in which boundary cells change from in- to outflow boundaries and in which the outflow boundary is free to diverge from the boundary input, Dirichlet boundary conditions can result in large gradients over the boundary when they instantaneously set the value at the boundary to the boundary input value. For models that use radiation boundary conditions, this can result in unrealistic large tendencies at the boundary. MESO-NH poses a less strict Dirichlet inflow boundary condition by setting the boundary value to a weighted average between the input value and the nearest LES domain value, with a weight of $0.8$ for the interior values (Lac et al., 2018). In this research we take a different approach and implement a Robin boundary condition, which will be derived in this section. The Robin boundary condition is a weighted average between a Dirichlet and Neumann boundary condition.

To derive the inflow boundary condition, we assume that advection is the only process taking place at the boundary.

$$\frac{\partial \psi}{\partial t} + u_n \frac{\partial \psi}{\partial n} = 0 \tag{10}$$

We also impose that the boundary value is nudged towards a given input value $\psi^B$ over a timescale $\tau$.

$$\frac{\partial \psi}{\partial t} = \frac{\psi^B - \psi}{\tau} \tag{11}$$

Combining these two constrains gives,

$$\frac{\psi^B - \psi}{\tau} + u_n \frac{\partial \psi}{\partial n} = 0, \tag{12}$$

which can be rewritten in the form of a Robin boundary condition.

$$\psi - u_n \tau \frac{\partial \psi}{\partial n} = \psi^B \tag{13}$$

The behaviour of Eq. (13) is determined by the value of $u_n \tau$. Dirichlet and homogeneous Neumann conditions correspond to different limits.

$$\lim_{u_n \tau \to 0} \psi = \psi^B \text{ (Dirichlet)}$$

$$\lim_{u_n \tau \to \pm\infty} \frac{\partial \psi}{\partial n} = 0 \text{ (Homogeneous Neumann)} \tag{14}$$

The classical Dirichlet inflow conditions can thus be obtained by setting $\tau = 0$. When $\tau \neq 0$ the boundary condition transitions from Dirichlet to homogeneous Neumann conditions as the velocity increases, avoiding large fluxes into the domain. At $u_n = 0$ the transition point between in- and outflow conditions, the boundary condition changes from Dirichlet (inflow), $u_n \tau = 0$, to homogeneous Neumann (outflow). This transition can be smoothed by introducing a variable timescale for the inflow conditions. The inflow conditions were derived with the proposition that advection nudges the boundary over a fixed time scale. At very low velocities advection plays a minor role and this assumption breaks down. To overcome this, the time scale needs to increase as the velocity approaches $0$. The following requirements are set for $\tau$:

$$\lim_{u_n \to 0} u_n \tau = \infty$$

$$\lim_{u_n \to \infty} \tau = \tau_0$$

$$\tau_0 = 0 \Rightarrow \tau = 0 \tag{15}$$

The first condition is set such that Eq. (13) approaches homogeneous Neumann conditions for $u_n = 0$, which removes the discontinuity. The second condition specifies that for large advection velocities we would like to have a constant nudging time scale $\tau_0$. The third condition allows to set the Robin inflow condition to Dirichlet inflow conditions when the nudging time scale is set to $\tau_0 = 0$. A dependency of $\tau \sim (1/u_n)^p$, where $p \geq 2$ satisfies the conditions. The relation used is given by:

$$\tau = \tau_0 \left[ 1 + \left| \frac{u_s}{u_n} \right|^p \right], \tag{16}$$

in which $u_s$ is a subgrid velocity scale at the boundary. Here we used the square root of the subgrid turbulent kinetic energy taken from the SFS-TKE scheme used by DALES (Heus et al., 2010). A different estimate can be used as well. When the resolved velocity is larger than the subgrid velocity, $u_n \gg u_s$, the timescale reduces to $\tau = \tau_0$. When the resolved velocity drops below the subgrid velocity the time scale will increase, providing a transition from the Robin boundary condition to the homogeneous Neumann condition at $u_n = 0$. The final form of the Robin inflow boundary conditions are given by Eq. (17).

$$\psi - u_n \tau_0 \left[ 1 + \left| \frac{u_s}{u_n} \right|^p \right] \frac{\partial \psi}{\partial n} = \psi^B \tag{17}$$

At the top of the domain the slab-averaged vertical gradient is taken into account. The Robin boundary condition at the top of the domain is given by Eq. (18).

$$\psi - w \tau_0 \left[ 1 + \left| \frac{u_s}{w} \right|^p \right] \left( \frac{\partial \psi}{\partial z} - \frac{\partial \langle \psi \rangle}{\partial z} \right) = \psi^B \tag{18}$$

## 2.3 Synthetic turbulence routine

To investigate the potential of synthetic turbulence in reducing the turbulence spinup area, the Random Flow Generation (RFG) algorithm of Smirnov et al. (2001) is implemented. When used, $\psi^B$ in Eq. (18) and Eq. (17) and $u_n^B$ in Eq. (4) are replaced by $\psi^B + \psi^R$ and $u_n^B + u_n^R$ respectively, where the superscript $R$ denotes the perturbation given by the RFG algorithm. In the calculations for the mass conservation correction factor, $\epsilon$ Eq. (7), $u_n^B$ is still used to satisfy condition Eq. (5). The RFG algorithm involves scaling and orthogonal transformation to create non-homogeneous anisotropic (near) divergence free velocity perturbations for a given covariance matrix $\overline{u_i' u_j'}$, turbulent length scale $\lambda$ and turbulent time scale $\tau^R$ from the summation of $N$ harmonic functions. The RFG routine is extended to give correlated potential temperature perturbations as well. From personal experience it is known that potential temperature perturbations are more effective in initiating turbulence than momentum perturbations. To create the potential temperature perturbations, a perturbation field is created from the summation of $N$ harmonics,

$$\alpha = \sqrt{\frac{2}{N}} \sum_{i=1}^{N} p_i \cos\left( \mathbf{k_i} \cdot \frac{\mathbf{x}}{\lambda} + \omega_i \frac{t}{\tau^R} \right) + q_i \sin\left( \mathbf{k_i} \cdot \frac{\mathbf{x}}{\lambda} + \omega_i \frac{t}{\tau^R} \right),$$

$$p,q,\omega \in N\left(0,1\right),$$

$$\mathbf{k} \in N\left(0,0.5\right), \tag{19}$$

where $\mathbf{x}$ is the position vector, $t$ the time and $N\left(\mu,\sigma\right)$ samples from a normal distribution with mean $\mu$ and standard deviation $\sigma$. Next, the perturbation field is scaled for a given $\overline{\theta'^2}$ and correlated to $w^R$ for a given $\overline{w'\theta'}$.

$$\theta^R = \left( \rho \frac{w^R}{\sqrt{\overline{w'^2}}} + \alpha\sqrt{1 - \rho^2} \right)\sqrt{\overline{\theta'^2}},$$

$$\rho = \frac{\overline{w'\theta'}}{\sqrt{\overline{\theta'^2}\,\overline{w'^2}}} \tag{20}$$

The RFG algorithm is easy to implement and is computationally inexpensive. A downside of the RFG algorithm is that it produces a Gaussian-model-like power spectrum. Huang et al. (2010) developed an improved algorithm that allows for any power spectra, but it comes with an additional computational cost. There are many other techniques and routines that are being used to help generate turbulence at inflow boundaries. However, the aim of this paper is not to study the performance of different inflow turbulence routines, but to rather show the potential of adding perturbations to the boundary input fields in general.

## 3  Simulation set up and methodology

The test case setup used in this research is summarized in Fig. 2 and consists of a "Big Brother"-like setup (Denis et al., 2002) to test the performance of the open LBC implementation, simulations with spatially and temporally smoothed input to test the influence of turbulence present in the input data and simulations with added synthetic turbulence in addition to the

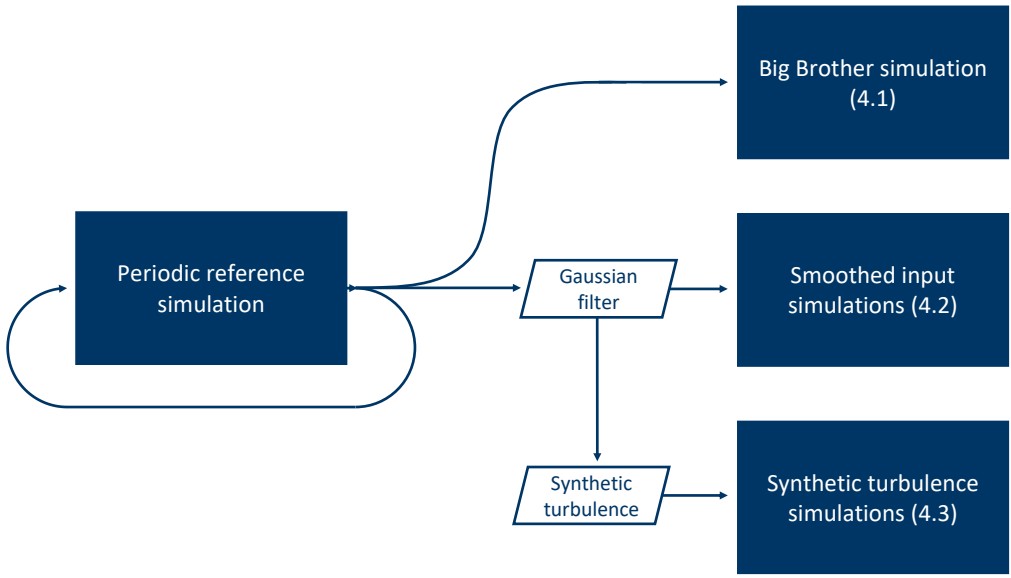

**Figure 2.** Illustration of the simulation setup. The solid blue rectangles show the different simulations and the sections in which their results are analyzed.

smoothed input to see how these algorithms can help generate turbulence. The simulation case used in the test setup is the
development of a dry convective boundary layer. This case is well understood and DALES is known to produce realistic results
(Heus et al., 2010). The dry convective boundary layer is forced with a constant surface heat flux of $\overline{w'\theta'}_s = 0.115 \mathrm{Kms}^{-1}$, a
zero surface momentum flux $u^* = 0 \mathrm{ms}^{-1}$ and a geostrophic forcing in the east-west direction corresponding to $u_g = 3 \mathrm{ms}^{-1}$.
The simulation is initialised with an east-west velocity of $U = 3 \mathrm{ms}^{-1}$, a north-south velocity of $V = 0 \mathrm{ms}^{-1}$ and an initial
potential temperature profile that consists of a boundary layer with a temperature of 300K, an inversion layer at 950m and an
inversion jump of $\Delta\theta = 8\mathrm{K}$ over 120m (linear interpolation between 300K and 308K over 120m) with a constant temperature
gradient of $\frac{\partial\theta}{\partial z} = 0.003 \mathrm{Km}^{-1}$ above. This corresponds to a convective velocity scale of $w^* = 1.5 \mathrm{ms}^{-1}$. The domain size is
$L_x \times L_y \times L_z = 15.36 \times 3.84 \times 1.92 \mathrm{km}$ with a horizontal resolution of $\Delta x = \Delta y = 60\mathrm{m}$ and a vertical resolution of $\Delta z = 20\mathrm{m}$.
The simulations last 6 hours and have an integration time step of $\Delta t = 5\mathrm{s}$. The subgrid scheme used is the SFS-TKE scheme
described in Heus et al. (2010). For the advection of all variables DALES' second order central scheme was used (Heus et al.,
2010). This setup is very close to the dry (strong) convective boundary layer shown in Heus et al. (2010), which was already
studied by Sullivan et al. (1998). The differences are the addition of a mean background wind, a weaker surface heat flux, a
higher horizontal resolution, the use of second-order advection schemes and a fixed integration time step. The initial profiles
and the evolution over time of the potential temperature, east-west wind velocity, potential temperature flux and east-west wind
variance are shown in Fig. 3.

The "Big Brother"-like experiment, as was first proposed by Denis et al. (2002), consists of a simulation with open boundary
conditions that is directly coupled to an identical reference simulation with periodic boundary conditions. The boundary fields

**Table 2.** Setup parameters for the reference case. From left to right; grid spacing, domain size, integration time step, surface heat flux, surface momentum flux geostrophic wind forcing.

| $\Delta x/y, \Delta z$ (m) | $L_x, L_y, L_z$ (m) | $\Delta t$ (s) | $\overline{w'\theta'}_s$ (Kms$^{-1}$) | $u^*$ (ms$^{-1}$) | $u_g, v_g$ (ms$^{-1}$) |
|---|---|---|---|---|---|
| 60, 20 | 15360, 3840, 1930 | 5 | 0.115 | 0 | 3, 0 |

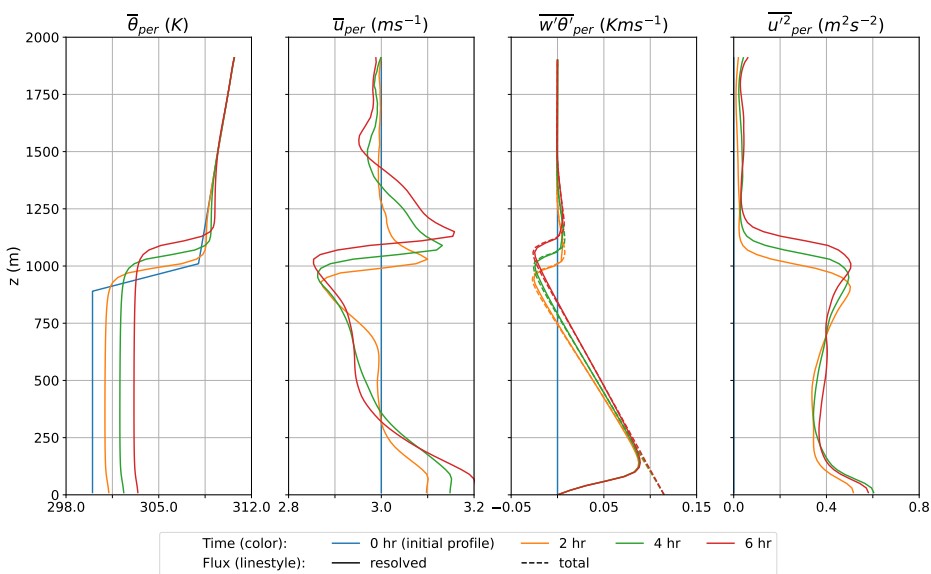

**Figure 3.** Evolution of the periodic reference case from initial profiles to end of simulation (6hr). Left to right; slab averaged potential temperature, slab averaged east-west velocity, slab averaged resolved and total heat flux, slab averaged east-west velocity variance.

of the periodic simulation are communicated every time step to the simulation with open boundaries. This allows us to directly study the influence of the open boundary implementation, since both the periodic and open boundary simulation are now identically forced and only differ in the implementation of their boundary conditions. The coupling is done offline, which means that the periodic simulation is done first and the boundary output is saved for every time step. This output is then used to force the simulation with open boundary conditions. In this setup the west boundary is (mainly) an inflow boundary, the east boundary (mainly) an outflow and the north and south boundaries will be in- and outflow boundaries changing for each grid cell and with time. The periodic simulation uses periodicity for the lateral boundaries and a no-stress boundary condition at the top (Heus et al., 2010). The simulation with open boundary conditions uses open boundary conditions for the lateral and top boundaries. First, we carry out a sensitivity analysis to study the dependence of the solution on the parameters introduced in the open boundary implementation. The parameters will be individually perturbed around a reference set. Next, a more in depth analysis is conducted on the results of the simulation with the reference parameters. The parameters for the sensitivity analysis are listed in Table 3 with the default parameters highlighted in green.

**Table 3.** Settings of the open boundary implementation for the sensitivity runs. The default settings are highlighted in green.

| $\Delta x^{\text{int}}/y^{\text{int}}$ (m) | $p$ (−) | $\tau_0$ (s) | Buoyancy term top boundary |
|---|---|---|---|
| $\Delta x/y$, $0.5L_{x/y}$, $L_{x/y}$ | 2, 3, 4 | 0, 20, 60 | on, off |

In practice, the open boundary conditions will often be used to couple the LES to a coarser resolution model, such as a meso-scale weather model. To study the impact of coarse resolution (in space and time) boundary data, the periodic output is smoothed with a Gaussian filter before it is used to force the open boundary simulation. The simulation with open boundary conditions is repeated for different degrees of spatial and temporal smoothing. This setup emulates a one-way nesting setup and moves from the LES being nested in a turbulence-resolving model to a non-turbulence-resolving model. It also allows us to study the influence of resolution ratios between parent and child model in a nested setup for both the spatial and tempo-

ral resolutions. Since the smoothed fields come from the same model with the same model physics, resolution and subgrid parametrisations, any differences between the results of the simulation with the smoothed input and the reference (periodic) simulation must be caused by the boundary implementation and the smoothing. Comparison to the case without smoothing allows us to see the influence of smoothing, which relates to the resolution/turbulent scales present in the emulated parent model.

Different techniques exist to artificially add turbulence or increase the turbulent scales present in coarse data. To demonstrate the potential of one such technique, the synthetic turbulence algorithm of Smirnov et al. (2001) is implemented and expanded to give perturbations for the potential temperature as well (Sect. 2.3). The smoothed-input open boundary simulations are repeated with the addition of synthetic turbulence. The perturbations are created using height-depended covariance matrices for $\mathbf{u}$ and $\theta$ obtained from the differences between the smoothed and non-smoothed input fields. The turbulent length scale

is set to the boundary layer height, which represents the largest turbulent eddies. The turbulent time scale is calculated as the turbulent length scale over the mean advection velocity. The covariance information would not be available in a real case setup, but it allows us to see how the algorithm would perform in a best case scenario. The purpose of these simulations is not to find the best synthetic turbulence implementation, nor to fine tune the implementation used, but to give an impression on how these routines can potentially improve the results.

**4   Results and Discussion**

This section will describe the results of the test case described in Sect. 3. First, the performance of the open boundary implementation is evaluated using the coupled periodic/open boundary simulations. Second, the influence of input turbulence scales is described using the smoothed-input simulations. Third, the prospects of synthetic turbulence are explored.

## 4.1 Big Brother simulation

In this section the results of the "Big Brother" experiment are shown. In this setup the periodic boundary output is given to the simulation with open boundary conditions at the same spatial and temporal resolution. This setup allows us to investigate the definition and implementation of the boundary conditions. Any disturbances present in the simulation with open boundary conditions must be a direct result of the boundary implementation, as the periodic simulation supplies "perfect" boundary fields. It is a first necessary test that needs to be passed. The challenging areas are mainly the outflow (east) boundary and the north

and south boundaries. At the outflow boundary, fields should leave the domain unperturbed and the area affected by reflections upstream of the outflow boundary should be minimal. The north and south boundaries are both in- and outflow boundaries and will therefore challenge the capability of the boundary conditions to switch from in- to outflow in time and space. The results from the simulation with open boundary conditions are compared to the reference case with periodic boundary conditions. We would like the mean field and the turbulence properties such as the length scales and energy distribution to be unaffected by

the numerics of the boundary condition implementation. The two simulations don't have to match from a deterministic point of view, as the chaotic nature of the system will result in different placement of eddies between both simulations.

To investigate the sensitivity of the solution on the parameters of the open boundary implementation, the simulation is repeated for different sets of parameters. Each of the parameters is individually perturbed around the default values. The parameters and their values are shown in Table 3. Figure 4 shows the slab average profiles calculated over the last half hour

of the simulation as a perturbation from the periodic profiles for potential temperature, eastward velocity, vertical potential temperature flux and eastward velocity variance. The profiles for the periodic simulation can be seen in Fig. 3. The black line represents the solution for the default values. Each color represent a simulation where one of the parameters is perturbed and the dashed or dotted line the perturbation value. Within the boundary layer, below $1000$m, the solution does not significantly depend on the values chosen for the parameters. All simulations are very close to the periodic simulation (within $1\%$), indicating

that the open boundary implementation does not have a significant impact on the solution.

At and above the inversion height the simulation with a larger timescale for the Robin inflow conditions, $\tau_0 = 60$s, and the simulation without the buoyancy term in the top radiation boundary condition perform significantly worse then the other simulations. Without the buoyancy term in the top radiation boundary conditions, reflections from the top boundary result in distortions in the top layer of the simulation. Sometimes a sponge layer is implemented to dampen these type of reflections,

but we don't need it here as, when used, the buoyancy term in the top radiation boundary condition solves the problem. The longer time scale for the Robin inflow conditions corresponds to a Robin boundary condition that is more weighted towards a Neumann boundary condition. A too long time scale gives too much freedom at the inflow boundary and allows for waves to build up around the inversion layer. A shorter timescale such as used in the default settings therefore works better. The default timescale is not set to zero, which corresponds to Dirichlet conditions, because a slightly relaxed condition works better for

simulations with lower mean background wind speeds.

For the integration length scale $\Delta x^{\mathrm{int}}$ and $\Delta y^{\mathrm{int}}$ the simulation where they are set to the grid resolution shows the best results. This corresponds to Dirichlet boundary conditions for the boundary-normal velocity components. These settings work

well for this setup, because the boundary input is turbulent, at the same resolution and from the same model. In other words, the simulation with open boundary conditions can find a solution that fits these boundary conditions. A larger integration length

scale gives the LES more freedom and works better when the boundary input does not contain turbulence or is from a different model. The simulations have also been done with a shorter timestep of 2s, the results for all but the Robin boundary condition time scale remain the same. For the Robin boundary condition the optimum time scale is lower for a shorter time step, which requires further research. All the results shown from here on are obtained with the default settings.

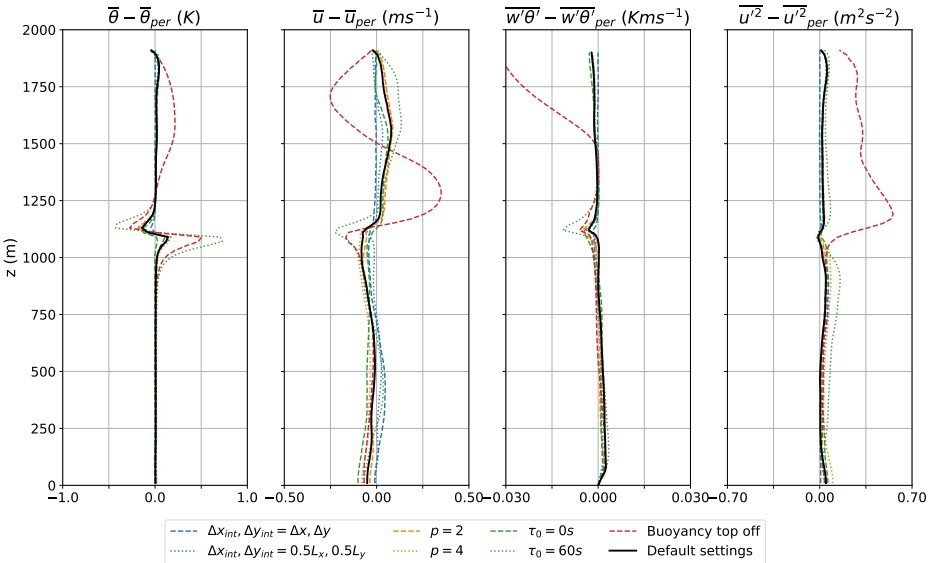

**Figure 4.** Sensitivity analysis for the open boundary implementation parameters. Slab average profiles for simulations that have parameters perturbed around a default configuration (Table 3). The profiles are calculated over the last half hour of the simulation as a perturbation from the periodic profiles (Fig. 3). Left to right; potential temperature, eastward velocity, vertical potential temperature flux, eastward velocity variance.

Figures 5 and 6 show a top (xy) view at $110\mathrm{m}$ and a side (xz) view of the potential temperature respectively. The top view is

shown as a perturbation with respect to the periodic slab average. The cross-sections are a snapshot after 6 hours of simulation time. The top panel shows the results for the periodic simulation and the bottom panel for the simulation with open boundary conditions. The location of the xz cross-section within the xy cross-section (and vice versa) is shown by the dashed line. The slope of the solid line in the xz cross-section of the simulation with open boundary conditions corresponds to the ratio of the advective velocity scale ($U = 3\mathrm{ms}^{-1}$) and convective velocity scale ($w^* = 1.5\mathrm{ms}^{-1}$). Left (upstream) of this line, fields will

be mainly dominated by information advected from the inflow boundary, whereas right of the line (downstream) the fields will be mainly influenced by convection originating from the surface boundary. The cross-sections are used to visually inspect the results to see if there are any discrepancies in the mean fields or turbulent structures. The simulations don't have to be similar from a deterministic point of view as the smallest differences at the boundaries would result in a different solution due

to the chaotic nature of the system. The results of the open boundary simulation are very similar to the periodic simulation.

The spatial scales and magnitude of the turbulent features resemble those of the periodic simulation. Up to 3km from the inflow boundary (left) the turbulent features of the open boundary simulation are almost identical in shape and location to the periodic simulation, which shows that the turbulent boundary input fields at the inflow boundary are communicated well to the open boundary simulation. Further downwind they start to deviate as a result of the chaotic nature of the system. No clear disturbances at any of the boundaries are seen and at the outflow boundary (right) the turbulent fields leave the domain without any significant reflections.

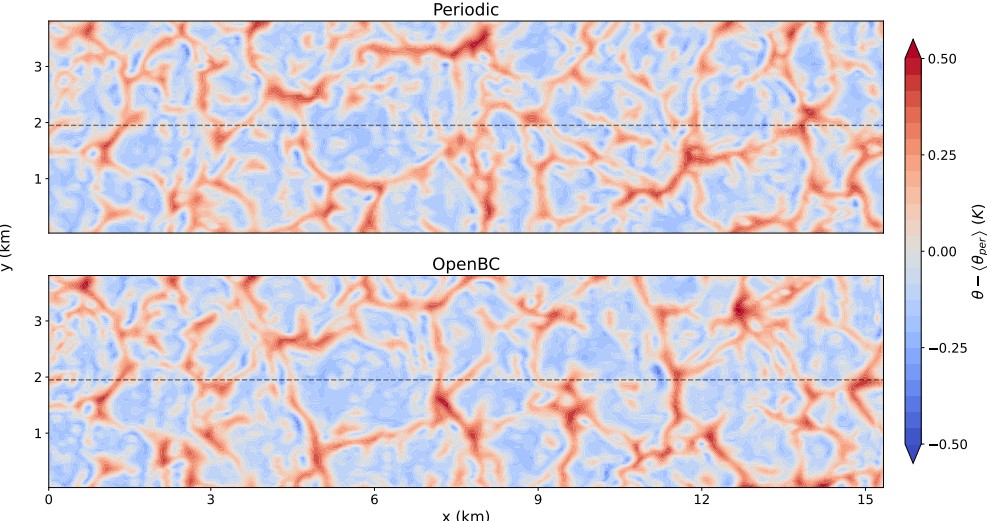

**Figure 5.** Horizontal cross-section of the potential temperature perturbation with respect to the periodic slab average at a height of 110m for the periodic simulation (top) and open boundary simulation (bottom). The dotted line shows the location of the xz cross-section shown in Fig. 6.

A more quantitative comparison of the influence of the open boundary conditions on the magnitude of the turbulent perturbations is obtained by calculating $\frac{1}{2}\left[\sigma_y^2(u) + \sigma_y^2(v) + \sigma_y^2(w)\right]$ for every time step and averaging it over the last half an hour of the simulation. $\sigma_y^2()$ denotes the variance in the cross-wind ($y$) direction. This quantity is very close to the definition of turbulent kinetic energy (TKE) and will therefore be referred to as TKE from hereon. Figure 7 shows cross-sections of TKE

for the periodic and open boundary condition experiments. The top panel shows the TKE for the periodic simulation and the bottom panel for the simulation with open boundary conditions. The grey dotted (dashed) contour lines mark the areas where the TKE values are smaller (larger) than the 2.5% (97.5%) percentile of the periodic simulation for that height. The slope of the solid black line corresponds to the ratio of the advective velocity scale ($U = 3\mathrm{ms}^{-1}$ and the convective velocity scale ($w^* = 1.5\mathrm{ms}^{-1}$) and can be used as a measure of where the information from the surface boundary condition meets the infor-

mation from the inflow boundary (left). The mean TKE values have similar magnitudes for both simulations. The simulation with open boundary conditions produces larger TKE values above the boundary layer and just before the outflow boundary

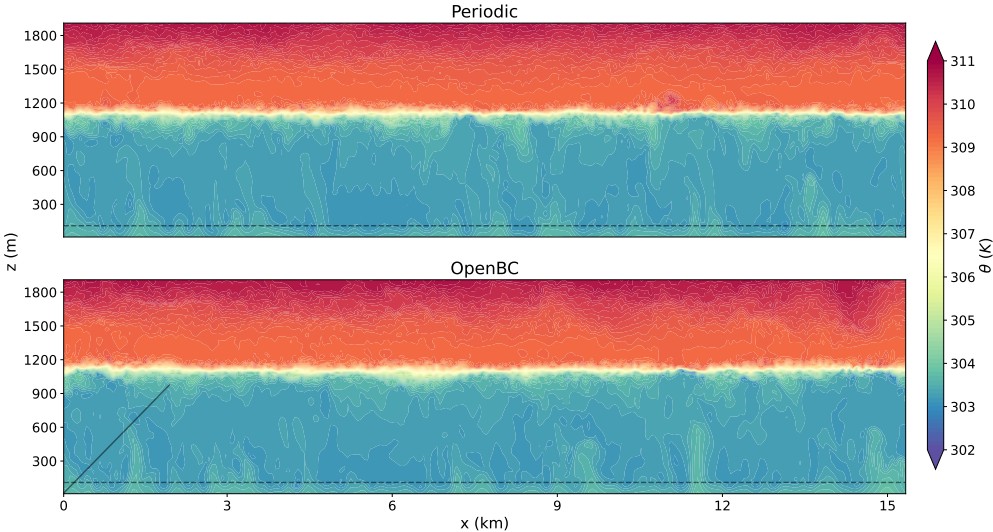

**Figure 6.** Vertical cross-section of the potential temperature for the periodic simulation (top) and open boundary simulation (bottom). The dotted line shows the location of the xy cross-section shown in Fig. 5 and the slope of the solid line corresponds to the ratio of the advective velocity scale ($U = 3\mathrm{ms}^{-1}$) and convective velocity scale ($w^* = 1.5\mathrm{ms}^{-1}$).

(right). The increase in TKE above the boundary layer might be caused by the higher wind speeds present in the open boundary simulation (Fig. 4). The increased TKE values at the outflow boundary are the result of reflections and disappear 1km upwind of the outflow boundary. To further quantify the differences between the simulations we vertically integrate the TKE over the boundary layer (Fig. 8) along the cross-section shown in Fig. 7. We find that the magnitudes of TKE between the two simulations is very similar, indicating that the boundary conditions have virtually no influence on the Big Brother simulation, once again with the small exception of a slight accumulation of TKE at the outflow boundary.

A wavelet analysis of the potential temperature field is used to quantify the influence of the open boundary conditions on the power spectrum of the turbulence. Figure 9 shows a wavelet analysis for the periodic (top) and open boundary (bottom) simulations. A one dimensional wavelet analysis is performed on an instantaneous xy-slab after 6 hours of simulation time. The wavelet analysis is done in the along-wind ($x$) direction. The results for each along-wind line are averaged over the cross-wind direction. A Morlet wavelet was used as the mother wavelet. The vertical axis shows the wavelength of the features on a logarithmic axis. The colors denote the wavelet power on a logarithmic scale. The hatched area indicates the cone of influence (COI), the COI describes the area that is potentially affected by boundary effects. These boundary effects result from the stretched wavelet extending beyond the edges of the domain and results within the COI should therefore be ignored. The grey dotted (dashed) contour lines mark the areas where the wavelet energy is smaller (larger) than the $2.5\%$ ($97.5\%$) percentile of the periodic simulation for that wavelength. The wavelet analysis shows similar results for both simulations. As expected, least energy is contained in the smallest wavelengths and most energy is contained in features with a wavelength similar to

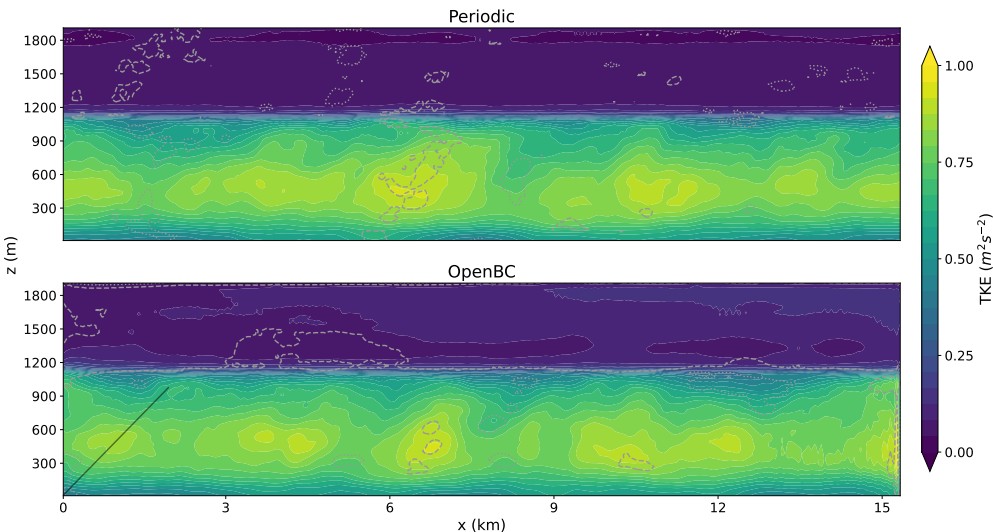

**Figure 7.** TKE profile derived from the cross-wind direction for the periodic (top) and open boundary (bottom) simulations. The slope of the solid line corresponds to the ratio of the advective velocity scale ($U = 3\mathrm{ms}^{-1}$) and convective velocity scale ($w^* = 1.5\mathrm{ms}^{-1}$). The grey dotted (dashed) contour lines mark the areas where the TKE is smaller (larger) than the $2.5\%$ ($97.5\%$) percentile of the periodic simulation for that height.

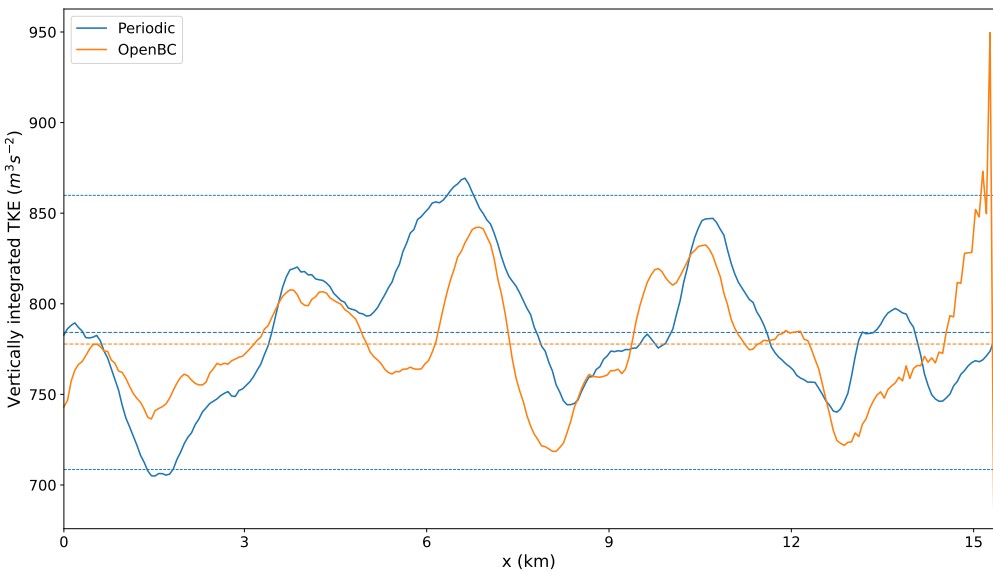

**Figure 8.** TKE integrated over the boundary layer. The dashed lines show the mean and the mean plus-minus two times the standard deviation.

the boundary layer height ($\approx 10^3\mathrm{m}$). There are no clear differences visible between the periodic and open boundary wavelet analysis, which indicates that the open boundary implementation does not influence the turbulent power spectrum.

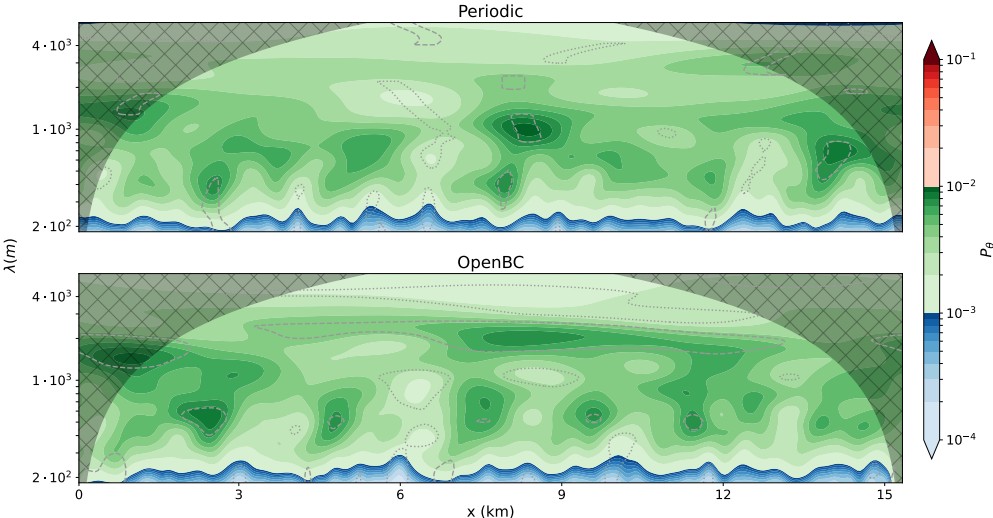

**Figure 9.** Wavelet analysis of the potential temperature at a height of $110$m for the periodic (top) and open boundary (bottom) simulations. The vertical axis shows the wavelengths of the features, the horizontal axis the distance from the inflow boundary and the coloring the energy present. The hatched area is the cone of influence and indicates the area that is potentially affected by boundary effects and results within should be ignored. The grey dotted (dashed) contour lines mark the areas where the wavelet energy is smaller (larger) than the $2.5\%$ ($97.5\%$) percentile of the periodic simulation for that wavelength.

From Figs. 5-9 it is concluded that the influence of the open boundary implementation on the simulation is minimal. The slab averaged fields, turbulent energy and spectral signature of the simulation are minimally perturbed by the implementation. Furthermore, the results of the sensitivity analysis show that solution is not sensitive to the values of the parameters, as long as they are within a reasonable range and the buoyancy term in the top radiation boundary condition is used.

## 4.2 Smoothed input simulations

This section will show and discuss the results of the smoothed-input simulations for different degrees of horizontal and temporal smoothing. This setup emulates the situation where the outer model provides boundary fields at a coarser spatial and/or temporal resolution than the LES. The panels in Figs. 10 and 11 show the same cross-sections as the bottom panels of Figs. 5 and 6 respectively for different degrees of smoothing. The horizontal axis of the panels shows the amount of smoothing in the temporal dimension and the vertical axis the amount of smoothing in the horizontal direction. The top left cross-section is the result without smoothing and is the same as the bottom panels from Figs. 5 and 6. For low degrees of smoothing, $\sigma_t \leq 30\Delta t$ and $\sigma_x \leq 4\Delta x$, the open boundary simulations resemble the periodic simulation and the solution is not significantly disturbed. For higher degrees of smoothing wavelike structures emerge at the inflow boundary (left) that persist up to $5$km into the domain. These structures become more prominent with increased smoohting. Horizontal smoothing (vertical axis) induces features that are aligned in the cross-wind direction. Temporal smoothing result in similar disturbances with the addition of some along-

wind disturbances. The spatial-temporal smoothing does not affect the outflow boundary, where turbulent structures leave the domain unperturbed with no visual reflections.

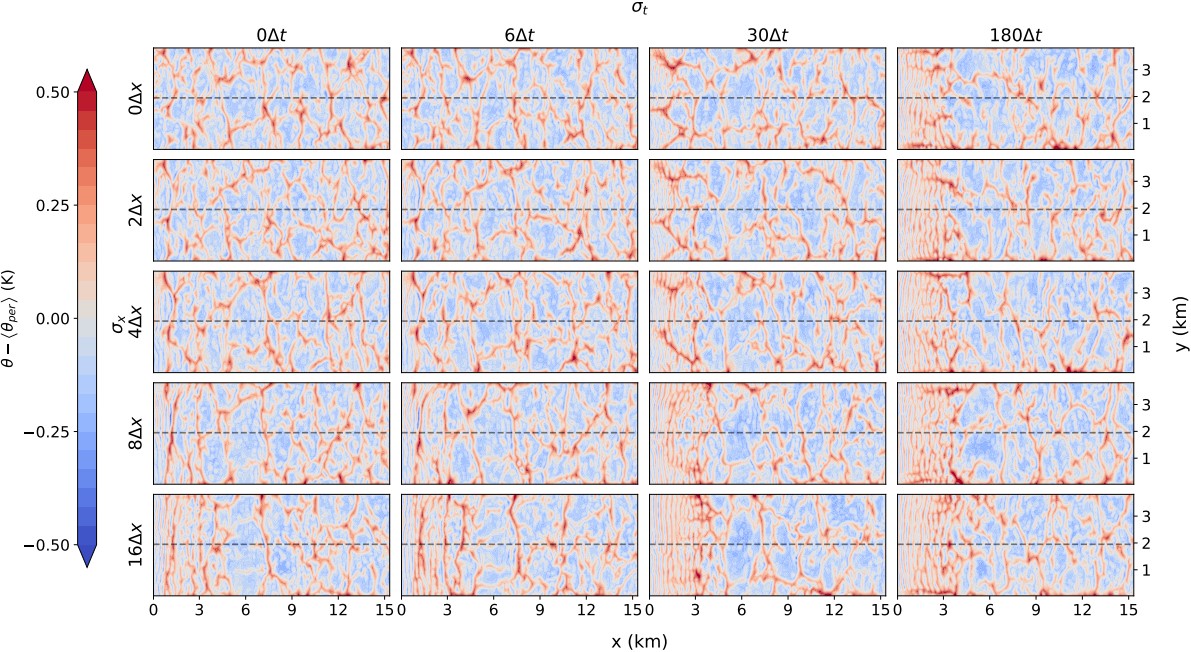

**Figure 10.** Horizontal cross-section of the potential temperature perturbations with respect to the periodic simulation at a height of 110m (similar to Fig. 5) for different degrees of smoothing. The horizontal axis of the panel shows the amount of smoothing in the temporal dimension and the vertical axis the amount of smoothing in the horizontal direction.

Figure 12 shows the TKE cross-sections for the smoothed-input simulations. Smoothing the input reduces the turbulent scales present in the input data. This results in an area of reduced TKE downwind of the inflow boundary (left). The slope of the black line in Fig. 12 indicates the ratio between the advective and convective velocity scales ($U/w^*$). It is expected that upwind (left) of this line the solution will be predominately dominated by information advected from the inflow boundary, whereas downwind (right) of this line convection would take over. The area of reduced TKE values downwind of the inflow boundary increases with increased smoothing and for large degrees of smoothing, the reduced TKE values extent much further than the line given by the $U/w^*$ ratio. For temporal smoothing of $\sigma_t \geq 30\Delta t$ a burst of TKE is present downwind of the reduced TKE area before settling to a TKE cross-section similar to that of the periodic simulation. This burst in TKE was also found by Muñoz-Esparza and Kosović (2018) and Kadasch et al. (2021). Our hypothesis is that the burst in TKE is a result of the clash between non turbulent fields that are mainly governed by information supplied at the lateral inflow boundary and turbulent fields originating from surface convection. We believe that the sudden transition from non turbulent flow to turbulent flow causes an overshoot in TKE. This phenomena is also seen during the spinup time of (periodic) turbulent simulations. During the first hour the turbulence in the boundary layer needs to build up. Only after this is developed it is capable of transporting the accumulated

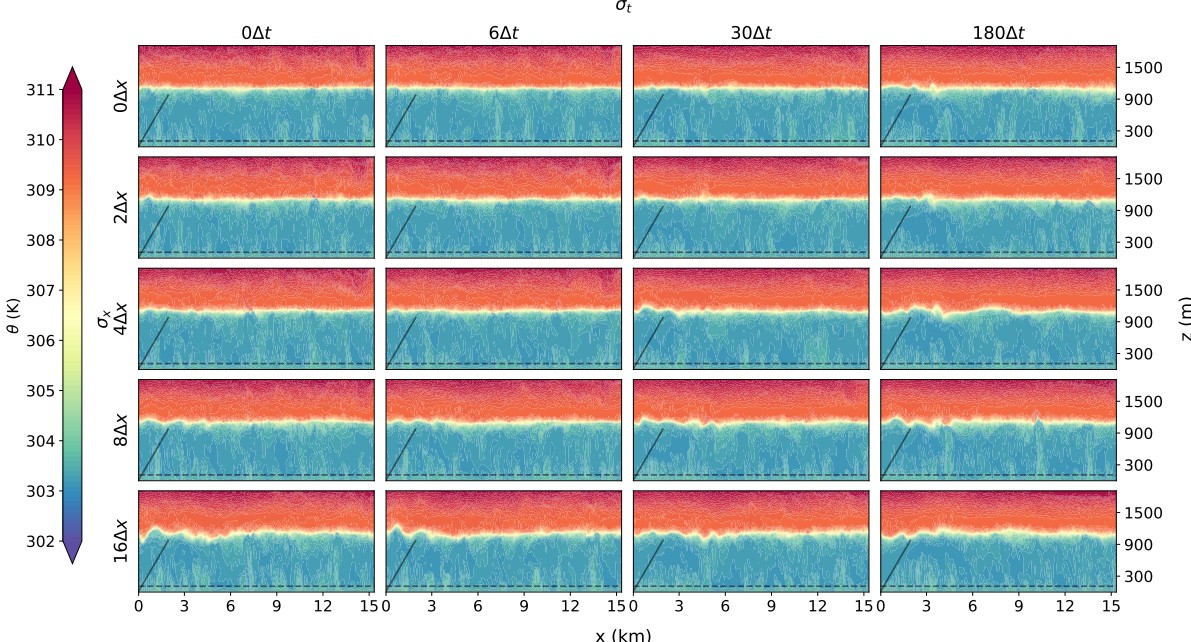

**Figure 11.** Vertical cross-section of the potential temperature (similar to Fig. 6) for different degrees of smoothing. The horizontal axis of the panel shows the amount of smoothing in the temporal dimension and the vertical axis the amount of smoothing in the horizontal direction.

surface moisture and heat flux through the boundary layer causing a peak in TKE but also in cloud fraction if clouds are formed on the top of the boundary layer (e.g. Siebesma and Cuijpers, 1995). In the worst cases (highest degrees of smoothing) it can take up to $6 - 7$ km before the TKE settles to values similar to those of the periodic simulation. The TKE field near the outflow boundary is not affected by the smoothing. The wavelike structures seen in Fig. 10 are not visible in the TKE cross-sections,

as they are aligned in the cross-wind direction, the same direction over which the TKE is calculated. Once again, the results are quantified further by vertically integrating the TKE over the boundary layer along the cross-section for all simulations (Fig. 13). Each simulation is shown as a thin line, with the control (no smoothing) and representative simulations for strong temporal and/or spatial smoothing highlighted in colour. Compared to the Big Brother experiment (Fig. 8), deviations from the control (periodic) simulations are much larger, in particular at the inflow boundary but also elsewhere in the domain. Comparing Figs.

8 and Fig. 13 once again highlights that the limitations in the open boundary simulations are mostly introduced by the spatial and temporal smoothing of the boundary values and not by the implementation of the boundary conditions themselves.

The wavelet analysis for the smoothed-input simulations is shown in Fig. 14. For low horizontal and temporal smoothing, $\sigma_x \leq 4\Delta x$ and $\sigma_t \leq 30\Delta t$, the influence on the results is small and the wavelet cross-section remains close to the periodic cross-section. For higher degrees of smoothing an increase in energy for wavelengths around $300$m is seen at the inflow boundary.

The energy increase at these wavelengths represent the waves seen in Fig. 14. The energy increase is larger for increased smoothing. For high degrees of smoothing a decrease in energy is seen for wavelengths around $1$km at the inflow boundary.

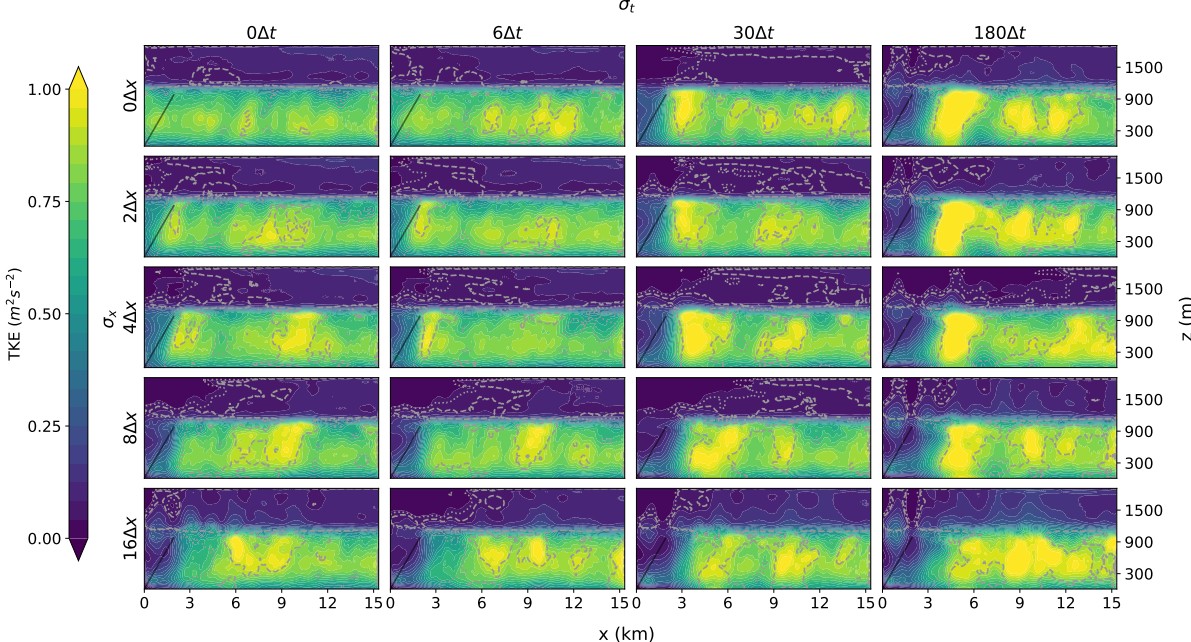

**Figure 12.** TKE cross-section derived from the cross-wind direction (similar to Fig. 7) for different degrees of smoothing. The horizontal axis of the panel shows the amount of smoothing in the temporal dimension and the vertical axis the amount of smoothing in the horizontal direction.

This represents the lack of developed turbulence near the inflow boundary as a result of the missing turbulence in the input data. The energy distribution moves towards the periodic profile downstream from the inflow boundary, with the maximum energy moving towards turbulence of the scale of the boundary layer height. For the highest degrees of smoothing this takes around 7km. The smoothing does not influence the wavelet spectrum at the outflow boundary.

The results analysed in Figs. 10-14 show that the input smoothing deteriorates the solution. For high degrees of smoothing, turbulent structures are missing at the inflow boundary and cross-wind oriented wavelike disturbances form. In the worst cases it can take up to 7km before the turbulent intensity and spectral signal evolves towards values close to the results of the periodic simulation. These results are important to take into account when coupling LES models to regional weather models. The latter usually have a spatial resolution on the order of kilometers and common output intervals are on the order of hours. This means that the ratios with the LES gridsize and timestep are at the bottom right of the shown panels. To avoid large ratios between the resolution of the input data and the LES model, repeated nesting can be used. With repeated nesting the LES can step down from the regional weather model resolution towards the desired resolution in steps with a determined refinement ratio. The results in this section suggest that a ratio of $4$ between the spatial resolutions and a ratio of $30$ between the temporal output and LES timestep should not be exceeded. In practice this is often hard to achieve, especially the temporal constraint as weather model data is often saved on a hourly interval. Another approach is to artificially add finer turbulent scales to the input data.

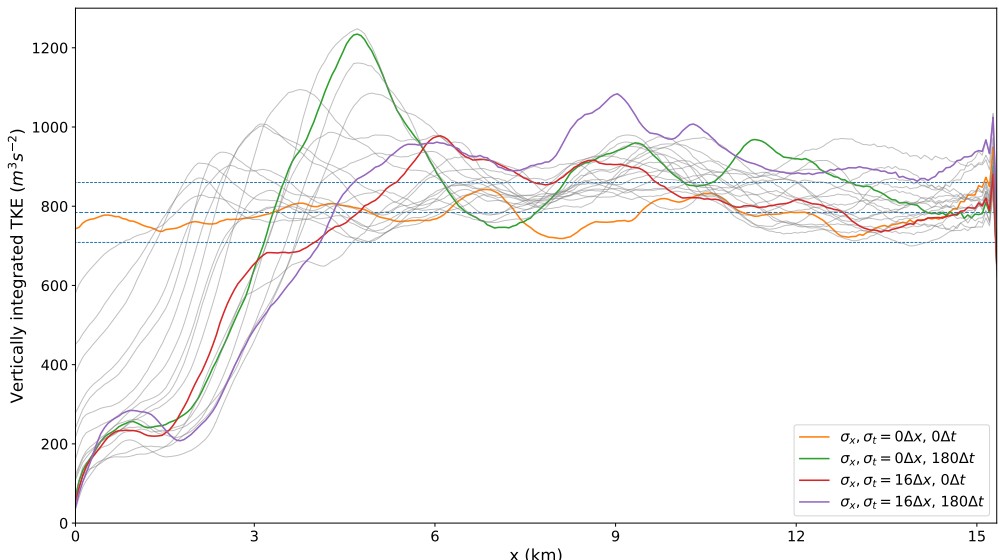

**Figure 13.** TKE integrated over the boundary layer. Each line represents one of the simulations from Fig. 12, with the corners from the panels being highlighted in colour. The blue dashed lines represent the mean TKE from the periodic simulation and the mean plus-minus two times the standard deviation.

This can be done by turbulence recycling, dedicated turbulence simulations or synthetic turbulence (e.g. Tabor and Ahmadi, 2010).

### 4.3 Synthetic turbulence simulations

The previous section has highlighted significant issues at the inflow boundary when the boundary values are smoothed in space and/or time, resulting in a more laminar flow near that boundary. A potential approach to reduce these issues (Smirnov et al., 2001) is to add synthetic turbulence to the boundary values. The purpose of this section is to investigate how the results in our simulations are affected by doing so. The algorithm of Smirnov et al. (2001) is implemented and extended to give potential temperature perturbations as well (Sect. 4.3). Figures 15 and 16 show the cross-sections for potential temperature. Compared

to Figs. 10 and 11 three things stand out. First, the addition of perturbations seems to remove the persistent wavelike structures at the inflow boundary (left). There are still disturbances present such as the perturbation at the inversion, but the persistent wavelike structure is gone. Second, the disturbances seem to disappear more quickly downstream. Third, the disturbances don't increase in magnitude with increased smoothing.

    Figure 17 shows the TKE cross-sections for the simulations with synthetic turbulence. The addition of synthetic turbulence

increases the TKE values directly downstream of the inflow boundary. The values are still below the developed-turbulence values of the periodic simulation. The synthetic turbulence does help to generate developed turbulence faster, which results in a smaller downstream area where the TKE is too low. The overshoot after the reduced TKE is also smaller in magnitude and

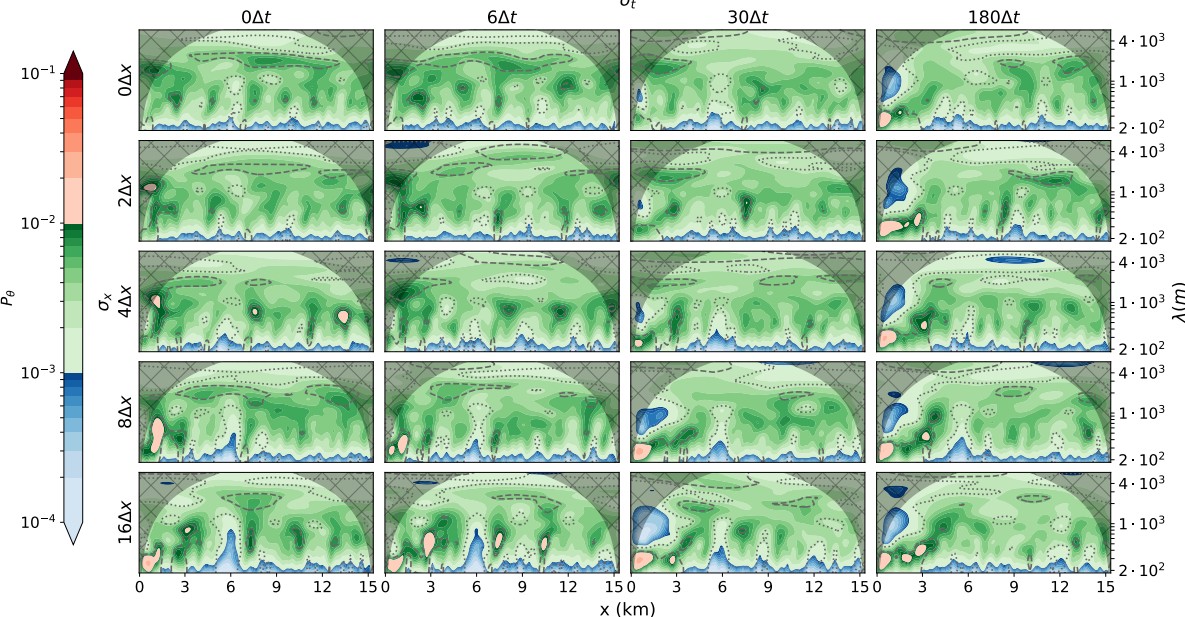

**Figure 14.** Wavelet analysis of the potential temperature at a height of $110\mathrm{m}$ (similar to Fig. 9) for different degrees of smoothing. The horizontal axis of the panel shows the amount of smoothing in the temporal dimension and the vertical axis the amount of smoothing in the horizontal direction.

area compared to Fig. 12. Furthermore, the overshoot no longer increases with increased smoothing in contrast to the results without added synthetic turbulence. The overshoot is similar in shape, magnitude and location for all smoothed simulations and

is located near the line that represents the ratio of the convective and advective velocity scales, where the information from the inflow boundary meets with the information from the surface. This means that it can be predicted where the overshoot is and which part of the simulation should be ignored. All simulations settle to a profile close to the periodic simulation within $5\mathrm{km}$ of the inflow boundary. The simulations with high degrees of smoothing do have an area with too much TKE. The addition of synthetic turbulence does not seem to have an influence on the outflow boundary. A quantitative comparison once again using

the vertical integral of TKE over the boundary layer (Fig. 18) confirms the positive influence of adding synthetic turbulence, showing a much reduced discrepancy from the control simulation at the inflow boundary in comparison to the simulations without it (Fig. 13).

    Figure 19 shows the wavelet analysis for the smoothed-input open boundary simulations. The increase in energy for wavelengths around $300\mathrm{m}$ is still visible at the inflow boundary, but the magnitude has been reduced by the turbulent perturbations.

The decrease in energy for wavelengths around $1\mathrm{km}$ is no longer there. Furthermore, the wavelet profile converges much faster to the periodic profile and the results don't seem to worsen with increased smoothing. The addition of synthetic turbulence does not affect the outflow boundary.

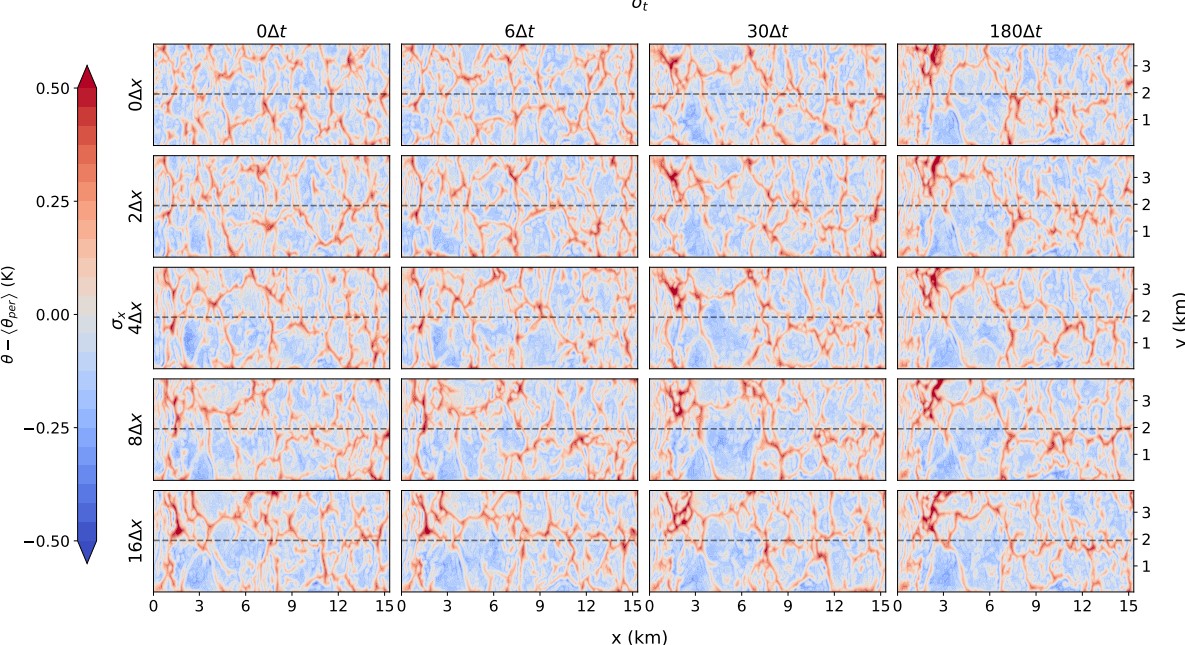

**Figure 15.** Horizontal cross-section of the potential temperature perturbations with respect to the periodic simulation at a height of 110m for different degrees of smoothing (similar to Fig. 10) with the addition of synthetic turbulence.

The results analysed in Figs. 15-19 show that the addition of synthetic turbulence on top of coarse input data can improve the simulation results. All of the inflow disturbances found in Sect. 4.2, as a result of coarse input data, were reduced in size and/or magnitude by the addition of synthetic turbulence. Furthermore, the location of the disturbances became predictable and their magnitude and size no longer increased with increased smoothing. The better performance when using synthetic turbulence may appear trivial. However, as we cannot add turbulence that is directly compatible with the LES solution, the synthetic turbulence could be dampened or generate artefacts near the inflow boundary. The fact that it does not, shows the value of using it in our implementation.

## 5 Conclusions

This paper introduced an open boundary implementation for atmospheric large eddy simulation models that was implemented in the Dutch Atmospheric Large Eddy Simulation model (DALES). The goal of this research was to give a detailed description of the implementation, investigate it's performance and show the influence of open boundary conditions and boundary input on the solution.

Radiation boundary conditions were implemented as an outflow condition for the boundary-normal velocity components at the lateral and top boundaries. At the top boundary buoyancy was also taken into account, which negated the need to add a

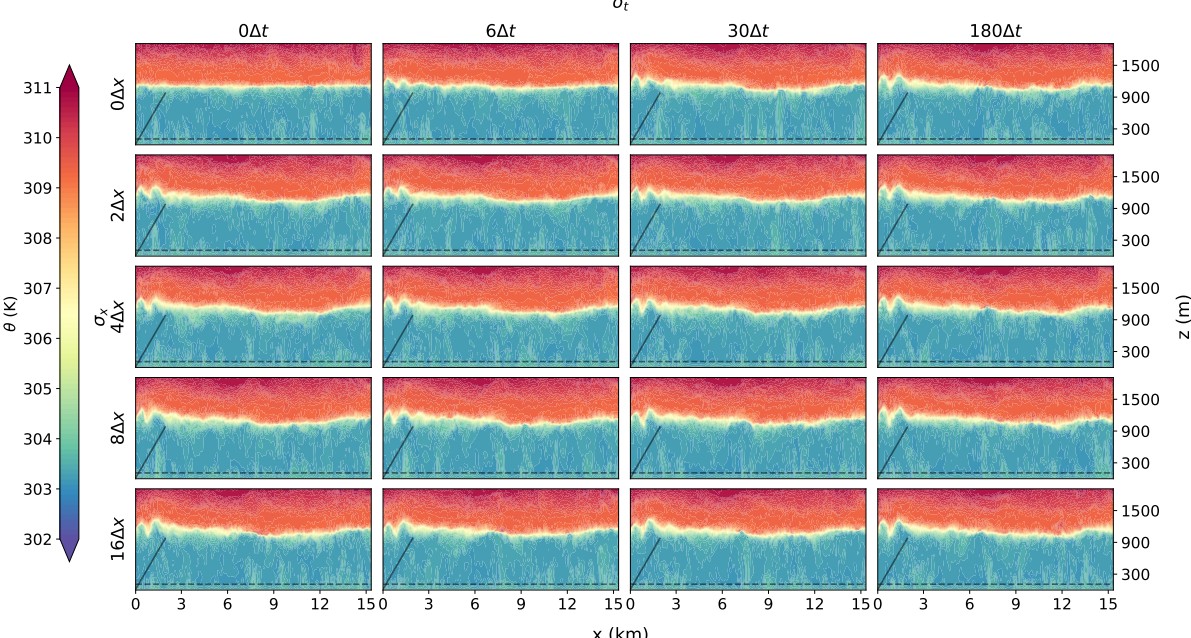

**Figure 16.** Vertical cross-section of the potential temperature for different degrees of smoothing (similar to Fig. 11) with the addition of synthetic turbulence.

sponge layer in the upper parts of the domain. Neumann conditions were used for the other variables at outflow boundaries. For inflow boundaries a Robin boundary condition was derived for the cell-centered variables and tangential velocity components to allow for a smooth transition between in- and outflow boundaries and a nudging condition was implemented for the boundary-normal velocity components.

Using a "Big Brother"-like setup, where a simulation with open boundary conditions was forced by an identical control simulation with periodic boundary conditions on the same spatial and temporal resolution, it was shown that the influence of the boundary implementation on the solution was minimal. Slab averaged profiles showed that the mean profiles are conserved. Furthermore, cross-sections of the potential temperature field showed that the turbulent input data was communicated well through the inflow boundary and that the turbulent fields left the domain without reflections or perturbations at the outflow boundary. Cross-wind turbulent kinetic energy cross-sections showed that the energy in the turbulent perturbations were the same in the simulation with open boundary conditions and the control simulation with periodic boundary conditions. The energy spectrum of the perturbations was also unchanged, which was shown with a wavelet analysis.

To investigate the influence of the spatial and temporal resolution of the input data, the output of the periodic simulation was smoothed before feeding it to the simulation with open boundary conditions. Different degrees of spatial and temporal smoothing showed that a mismatch between input turbulent scales and model scales results in the generation of wavelike disturbances downstream of the inflow boundary. The disturbances grow in size and magnitude when the ratio between input

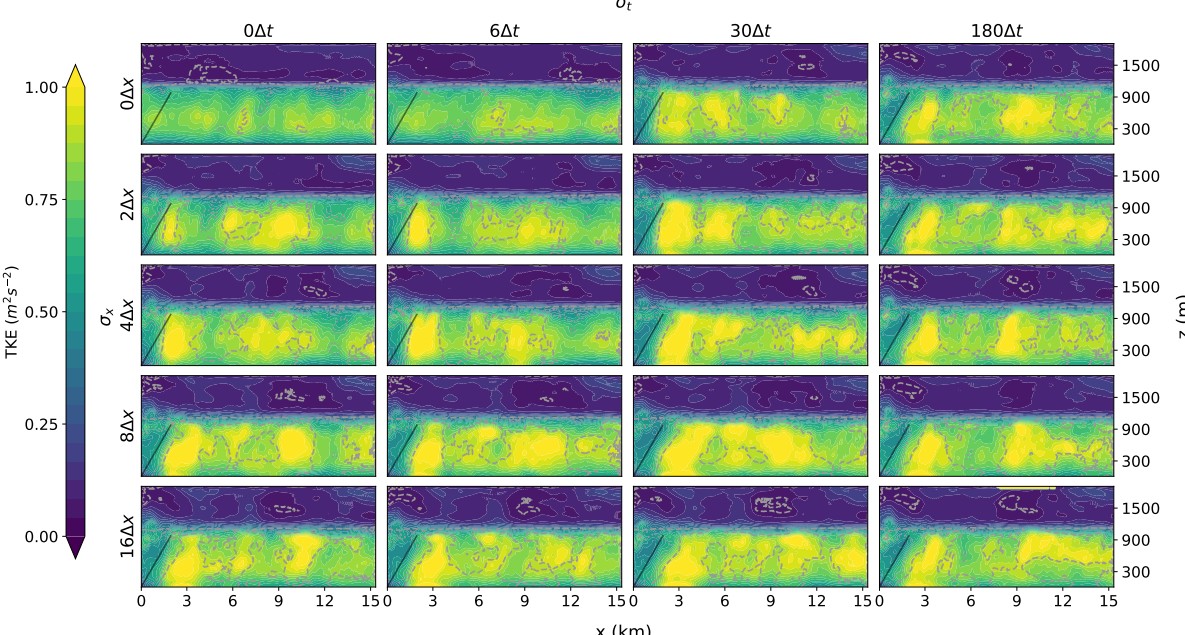

**Figure 17.** TKE cross-section derived from the cross-wind direction for different degrees of smoothing (similar to Fig. 12) with the addition of synthetic turbulence.

and model scales grows. The lack of turbulence in the input data also results in an area of reduced turbulent kinetic energy downstream of the inflow boundary, where there is no developed turbulence. This area growed as the smoothing increased. For large degrees of smoothing it was found that the turbulent energy overshoots before settling to values similar to the periodic control simulation. For these reasons, it is advised to be careful when coupling a large eddy simulation model with open boundary conditions to a coarser model. Repeated nesting can be used and is currently being explored to step down in multiple steps from coarse data to the desired resolution. The results of this research indicate that the refinement factor when nesting should not exceed 4 in the spatial dimension and 30 in the time dimension.

The potential of adding synthetic turbulence to the LBCs was explored and the results show that it can help to reduce the found disturbances in size and magnitude and to speed up the process of obtaining developed turbulence by artificially reducing the gap between the input turbulent scales and model scales. The strong wavelike character of the disturbances were removed and the length of the inflow area required for turbulence to develop was reduced. The disturbances and development area also became less dependent on the degree of smoothing and the development area is given by the ratio of the advective and convective velocity scales. However, if possible, we would still advise to keep the spatial and temporal ratios between the input data and the LES below the earlier mentioned values.

In summary, the implementation of open BCs described in this study provides a suitable framework for further investigating the use of the DALES model in "nested" mode. This provides a major advance in its utility as a science tool, as it increases

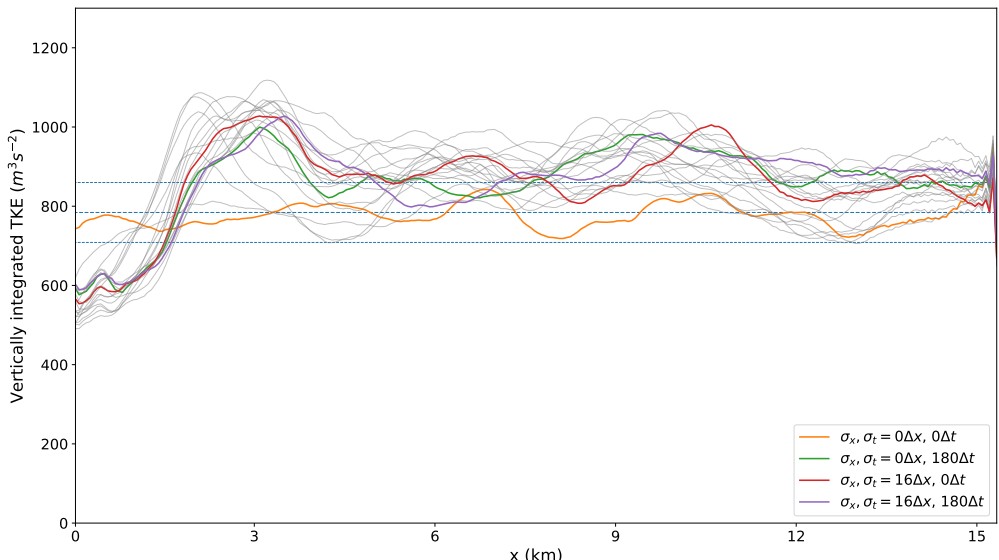

**Figure 18.** TKE integrated over the boundary layer. Each line represents one of the simulations from Fig. 17, with the corners from the panels being highlighted in colour. The blue dashed lines represent the mean TKE from the periodic simulation and the mean plus-minus two times the standard deviation.

its applicability to problems for which periodic BCs have strong limitations, such as over heterogeneous terrain. Spatial and temporal averaging of the boundary values, as is typical for embedding an LES into coarser resolution meso-scale models, deteriorates the results. The smoothing effects are much larger than those from the implementation of the open BCs themselves. Some of the deterioration can be overcome by adding synthetic turbulence at the inflow boundaries.

*Code and data availability.* The current version of DALES is available from the project website: https://github.com/dalesteam/dales under the GNU General Public License. The exact version of the model used to produce the results used in this paper is archived on Zenodo, as are input data and scripts to run the model and produce the plots for all the simulations presented in this paper (Liqui Lung et al., 2023).

*Author contributions.* FLL conceptualized the paper, did the formal analysis and visualization, implemented the methodology and software and wrote and edited the draft. CJ and PS supervised during the project, reviewed the draft and supported in the conceptualization. FJ supported the software implementation and reviewed the draft.

*Competing interests.* The authors declare that there are no competing interests

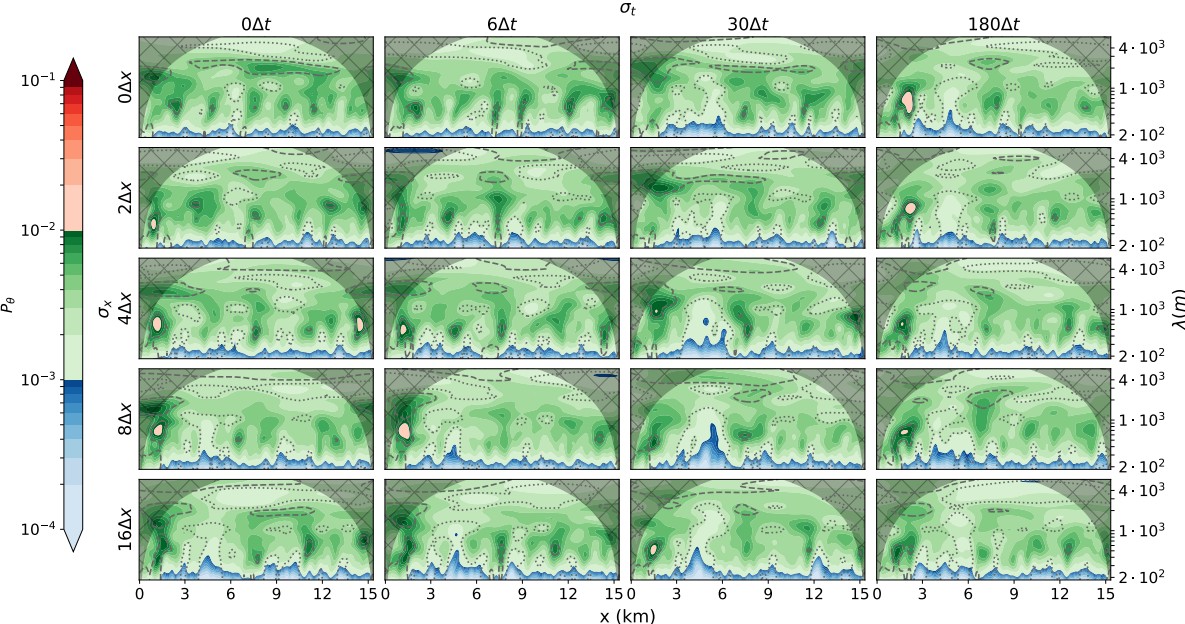

**Figure 19.** Wavelet analysis of the potential temperature at a height of 110m for different degrees of smoothing (similar to Fig. 14) with the addition of synthetic turbulence.

*Acknowledgements.* We acknowledge the use of ECMWF's computing and archive facilities, the funding provided by the Australian Research Council Centre of Excellence for Climate Extremes (CE170100023) and the support of the Ruisdael Observatory, a scientific research infrastructure which is (partly) financed by the Dutch Research Council (NWO, grant number 184.034.015)

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



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
