# Peer review of "Open Boundary Conditions for Atmospheric Large Eddy Simulations and the Implementation in DALES4.4"

_Geoscientific Model Development, 2023_

## Referee Comment (RC1)

**Review of the paper**
*Open Boundary Conditions for Atmospheric Large Eddy Simulations and the
Implementation in DALES4.4*
**by Franciscus Liqui Lung, Christian Jakob, A. Pier Siebesma, and Fredrik
Jansson, submitted to** *GMD*

My opinion about this paper is split. On the one hand, I find the topic and the numerical experiments interesting, due in particular to the very high resolution. On the other hand, there are several aspects that I feel need to be reworked:

A Several points in the description of the open boundary conditions are unclear or even not mentioned.

B The reference test case is not sufficiently described, which prevents from really evaluating the performance of the boundary conditions.

C Many statements may seem rather weak, or even quite obvious, in the comments of the simulation results. I think that the conclusions should be strengthened

**A - Description of the open boundary conditions**

1. Eq. (1) does not make sense, since it adds scalar values, like $\partial u_n/\partial t$ or $\epsilon$, and a vector value $\hat{z}$

2. Line 139: $x_n - \hat{x}.\hat{n}\,\Delta x_n$ is a location, not a cell.

3. Line 143, *Equation (2) is discretised using a second order forward scheme*: what does it mean exactly? Please provide the expression of the numerical scheme. Idem for the discretisation of (1).

4. Line 145, *a Dirichlet boundary condition is used for the boundary-normal velocity components*: I do not agree. A Dirichlet boundary condition for the boundary-normal velocity component would read $u_n = u_n^B$. And a Dirichlet boundary condition for the tendency of the boundary-normal velocity component would read $\dfrac{\partial u_n}{\partial t} = \dfrac{\partial u_n^B}{\partial t}$. (3) is actually some kind of nudging of $u_n$ towards $u_n^B$, with a relaxation time scale equal to $\Delta t$.
   Moreover the time discretisation of (3) should also be indicated.

5. Eq. (4): $S(B)$ is not defined.

6. Eq. (5): $S^{\mathrm{int}}$ is not defined. I understand that it is a patch around the boundary, but it should be defined exactly.

7. Eq. (6) is definitely unclear to me. Is $\epsilon$ a constant or does it depend on space and time? Is the $\epsilon(S^{\mathrm{int}})$ the same as $\epsilon$? If $\epsilon$ is a constant, (6) is indeed only the time derivative of (5), which does not involve any $\epsilon$. The way $\epsilon$ is actually estimated should be rewritten clearly.

8. Eq. (7): why do you choose a zero normal flux condition at outflow for all variables but $u_n$? You could have made other choices: please elaborate a little bit.

9. Line 206 and Eq. (9), *advection over an inflow boundary nudges the boundary value to a given input value*: this sentence corresponds to the equation

$$\frac{\partial \psi}{\partial t} + u_n \frac{\partial \psi}{\partial n} + \frac{\psi - \psi^B}{\tau} = 0$$

which is different from what is implemented. Actually (9) corresponds to the nudging inflow condition for $u_n$ (3) (without $\epsilon$, and with a more general relaxation time scale). But since $\psi$ is discretised one half-cell into the domain and not on the boundary, you have to decide what the value of $\psi$ is on the boudary. For this, you assume that $\psi$ is locally transported at speed $u_n$, i.e. $\frac{\partial \psi}{\partial t} = -u_n \frac{\partial \psi}{\partial n}$.

**B - Reference test case**

The reference test case is not really described. It is only said that it is a simulation of *the development of a dry convective boundary layer*, along with a three-line description of the vales of parameters.

10. A better overview of the solution should be given (e.g. some snapshots), and aspects which could have an impact on the performance of the OBCs should be emphasized (e.g. fluctuations in time of the direction — incoming or outgoing — of the flow near the open boundaries).

11. The objectives should be explained: what do the authors want from the OBCs ? What are the key properties and diagnostics that should not be impacted by open boundaries? In particular, do you expect to reproduce the behavior of the reference solution from a statistical point of view or from a deterministic point of view? What are then the quantitative criteria that will be used to assess the performance of the OBCs?

Some details:

12. line 271 *with periodic boundary conditions*: I suppose that periodicity is achieved in the $x$ and $y$ directions, but not in the $z$ direction?

13. lines 280 and 306: *boundary conditions* should be *boundary data*

**C - Weak statements and conclusions in Section 4**

In my opinion, the critical presentation of the numerical results (Section 4) should be improved, and the conclusions should be strengthened.

12. All figures visually compare reference fields with other ones obtained in simulations with OBCs, but no difference is never quantified.
For instance: *The TKE field near the outflow boundary is not affected by the smoothing* (line 387) , or *the wavelet cross-section remains close to the periodic cross-section* (line 390). Please quantify.

13. Figures 3 to 6: Those figures could be complemented with the difference between the two panels. And the conclusions fully depend on the criteria: do you want a statistical matching or a deterministic matching between the two panels? How could you quantify it?

14. Lines 358-360, *... shows similar results for both simulations... no clear differences visible...*: in my opinion, this is exaggerated. One should better explain why we can consider that the differences are not significant, which again depends on the criteria that have been chosen.

15. Section 4.1: boundary data are perfect in this experiment, with the same spatial and temporal resolution as the reference simulation. Dirichlet boundary conditions everywhere would therefore give a perfect result. So it is not surprising that the results are good in the vicinity of the inflow boundary. It is what happens near the output that is a priori the most interesting.

16. Several statements are quite obvious: smoothing the input data results in a reduced TKE downwind of the inflow boundary, and deteriorates the solution; adding synthetic turbulence helps to generate developed turbulence faster... Again defining, from the beginning, clear desirable quantitative criteria would help.

17. Lines 385 and 421: it is mentioned that a burst ok TKE is observed, but is there an explanation for it?

18. Section 4.3: the goal of this section is not clear to me. Do you expect for the solution to reproduce the reference solution from a deterministic point of view, or to have a correct level of turbulence? The key question is perhaps the following: which scales are closer (in some sense to be defined) to the reference ones when this artificial turbulence is added?

19. A suggestion: To the best of my knowledge, the introduction of a variable timescale $\tau$ for the inflow condition (Eq. (13)) is something new. In my opinion, this is a possible contribution, that is worth being discussed and emphasized.
    In other words, you could discuss more in depth this aspect, by comparing results with a Dirichlet inflow condition on $u^B$ ($\tau_0 = 0$), a Dirichlet condition for the tendency $\dfrac{\partial \psi}{\partial t} = \dfrac{\partial \psi^B}{\partial t}$, and intermediate conditions with several values of $\tau_0$ and $p$, including $p = 0$ (fixed timescale $\tau = 2\tau_0$). Relevant diagnostics should make it possible to decide if the time and space variability of the timescale has a significant effect.

---

## Author Comment (AC1)

**Authors response Liqui Lung et al. (2023) - Open Boundary Conditions for Atmospheric Large Eddy Simulations and the Implementation in DALES4.4**

Franciscus Liqui Lung
Christian Jakob
Pier Siebesma
Fredrik Jansson

January 2024

We want to thank the referees for their in depth review of our submitted manuscript and their comments and suggestions. In this response we aim to address their comments. For readability we have collected the comments from both reviewers and will discuss them per section.

**1    Major comments**

RC1 Several points in the description of the open boundary conditions are unclear or even not mentioned.

AC We agree with the concerns raised and addressed the comments in the *open boundary implementation* section of this document.
* * *
RC1 The reference test case is not sufficiently described, which prevents from really evaluating the performance of the boundary conditions.

AC We have elaborated on the reference case setup and included the initial profiles, a table with all the forcings and for the periodic simulation height profiles at different time intervals for the potential temperature, resolved vertical temperature flux $(\overline{w'\theta'})$, east-west wind velocity $(u)$ and east-west resolved wind variance $(\overline{u'^2})$.
* * *
RC1 Many statements seem rather weak, or even quite obvious, in the comments of the simulation results. I think that the conclusions should be strengthened.

AC We clarified the objectives on which we judge the performance of the boundary conditions. The simulations are evaluated statistically and not deterministically. The addition of synthetic turbulence is not to retrieve the same results as the unsmoothed simulation, but rather to mitigate artifacts as a result of the missing turbulence in the input data. To help quantify the influence of the boundary conditions on turbulence, the TKE (which will be renamed as it is strictly speaking not TKE see later comments) is integrated over the boundary layer. This gives an easier to read plot. Details of the adjustments can be found in the *Discussion and Presentation of Results* section of this document.
* * *
RC2 The author motivate their study by nesting LES domains into large-scale model domains. It is well known that other LES models which use Dirichlet boundary conditions for time-dependent mesoscale flow inputs sometimes suffer from wave-like structures near the boudaries, so better formulated boundary conditions to overcome this would be highly appreciated. However, as far as I understand, the boundary conditions described herein are only supposed to be used for idealized situations where the inflow and outflow boundary are fixed over the LES simulation period. For example, in a mesoscale-nested simulation, it is likely that the wind speed and direction continuously change in time, meaning that an inflow boundary can become an outflow boundary and so on. While this is still considered in the equations, though not supported by any analysis, the situation where a lateral boundary can become both, inflow and outflow boundary at the same time, is not considered in the equations. For example, this situation can occur if you want to model mesoscale phenomena like sea breezes, local wind systems, convective situations with weak winds, or situations like frontal passages. This is because the radiation boundary condition requires slab averages of the outward-pointing component. If there is a significant inflow at this boundary, the $\langle u_n \rangle$ can become negative. In case this happens, the flow becomes quickly unstable in conjunction with radiation boundary conditions, meaning that the proposed method is only applicable for idealized scenarios. Thus, the use of a slab average actually prohibits that a boundary can be both, inflow and outflow boundary at the same time. I recommend to rephrase the general motivation in this context, in order to avoid the impression that the proposed formulation of the boundary conditions solves the issue in general.

AC Our ultimate goal is to be able to nest DALES in mesoscale models. We agree that mesoscale-nested simulations involve time-varying boundary conditions and this has played an important part in how we defined our boundary conditions. We acknowledge that the presented test setup does not include all the challenges of a mesoscale-nested simulation. However, we believe that the presented setup is a first necessary set of tests that the implementation needs to pass before moving to more complicated test

cases in future publications as they may mask basic problems with the open boundary implementation. We are aware of the instabilities that can arise with radiation boundary conditions that use slab averages on time-varying boundaries. This is why we chose not to use slab averages, but instead defined the integration length scales over which we calculate the phase velocity and mass flux correction term. The integration length can conveniently be chosen to be the resolution of the "mother" model. This choice gives maximum freedom to the boundary conditions given the constrains imposed by the mother model (see *boundary implementation* section for more on this). We believe, that the mass correction term plays an important role in preventing any instabilities from building up. From other comments we do realize that the description of this correction term was not clear and we have elaborated on it in the *open boundary implementation* section. We do not believe the presented implementation can only be used in idealized setups and as this is also not our ultimate goal, we do not want to phrase it this way. The goal to be able to do mesoscale-nested simulations has motivated our implementation choices and we therefore do want to mention it. However, we do agree with the reviewer that the presented test case is not sufficient to claim that the setup will work in a mesoscale-nested setup and we will remove any such claims and mention that further testing is required. Since the submission of this manuscript, we have used this implementation to nest DALES in a mesoscale model, we will leave these results however for later publication.

RC2 The description of the boundary conditions lacks important information and is partly misleading. For example, the boundary conditions are formulated as tendencies instead of boundary values. However, the boundary value itself is required for the spatial descretization of the advection term, so I recommend to reformulate the equation towards boundary values. Further, the term slab average is not fully defined. It seems to have a different meaning at the outflow boundary compared to the inflow boundary. Moreover, the formula for the time-scale computation seems to be wrong because the second term in Eq. 13 does not become dimensionless.

AC We have addressed most of the comments that reviewers had on the description of the open boundary implementation in the *open boundary implementation* section. We however do not understand the comment on using tendencies instead of boundary values as we do not see a problem with using current time step values to calculate tendencies for the next time step. We have added the discretisation schemes and believe this will clarify the implementation. The $e$ in the time-scale computation represents a subgrid velocity scale and not the subgrid TKE. In this case we used the square root of the subgrid TKE as the velocity scale and have added this information to the manuscript.

RC2 The setup description of the test case lacks important information. Which surface boundary conditions did the authors use (momentum, heat, SGS-TKE, ...), which numerical schemes were applied (pressure solver, advection and time discretization, ...). Moreover, it is not clear to me how the north and south boundaries were treated (period BC vs. inflow/outflow BC?).

AC We agree with the reviewers that description of the test case lacks information and we have updated the description. We have included the initial profiles, surface boundary conditions, subgrid scheme and a table that includes all forcings. We have also referenced the DALES paper [Heus et al., 2010] for the used discretisation schemes and information on the pressure solver. In the simulations with open boundary conditions the north and south boundaries are treated as open boundaries. Depending on the input velocity, cells on these boundaries will either be inflow or outflow this will change between cells and with time.
* * *
RC2 I like the idea of a big-brother simulation to investigate the impact of the open boundary conditions in a systematic manner. However, the performance of the open BC is not sufficiently supported by the test case and the analysis. The authors only used a single setup for a convective boundary layer with a fixed inflow and outflow boundary. However, convection may easily masked systematic effects because instantaneous fluctuations may superimpose weaker systematic biases. For this purpose I think the evaluation of the model need to be extended towards purely neutral flows. Moreover, I think the test scenario should be also extended to a case with changing inflow conditions with respect to the wind speed to i) evaluate the performance of the mass-conservation scheme and ii) to demonstrate that proposed time-dependent relaxation time-scale algorithm works properly. Also a test case with changing wind direction is required to demonstrate that the boundary conditions can also deal with such situations.

AC The goal of this paper is to describe the current implementation of open boundary conditions in DALES and present a first necessary set of tests. We agree with the reviewer that the proposed cases all test and show different aspects and we have conducted some of them in the past (neutral and mesoscale-nested), however for readability we do not want to include them in the current manuscript and we will leave them for future publications. As mentioned before we will remove any claims that can not be supported by the current test case or state that they require further testing. We will also remove any references to simulations not presented in the manuscript.

**2 Introduction**

RC2 l8: The first part of the sentence sounds strange and should be rephrased.

  AC Will rephrase *The results show that when the ration between input and model resolution increases,* to *When smoothing is applied over larger/longer spatial/temporal scales,*
* * *
RC2 l12: I wouldn't say LES exists to study small scale weather phenomena but would formulate this in a more general way, e.g. to study turbulent motions.

  AC Agree, we will rephrase *study small scale weather phenomena* to *study turbulent motions*
* * *
RC2 l25: What do the authors mean by the term "fields"?

  AC We mean the variables. We will change *fields at inflow boundaries and propagate fields* to *variables at inflow boundaries and propagate variables.*
* * *
RC2 l43-45: It would be useful for the reader if the authors would be more specific, i.e. which model uses which kind of BC. The way the sentence if phrased is too general in my opinion. Also, concerning a description of inflow/outflow BC, the Maronga et al. (2015, https://doi.org/10.5194/gmd-8-2515-2015) paper is more suited reference.

  AC We will add a table with information on the different open boundary condition options in the mentioned models. We will also reference Maronga et al. [2015]. Maronga et al. [2015] describes the fixed in and outflow setting present in PALM 4.0, Maronga et al. [2020] however also describes the new possibility of self-and-rans nesting, for which they use prescribed boundary conditions, so we will reference both.
* * *
RC2 l48-l49: In addition to the Mazzaro paper it would be nice to add the original literature (Mirocha et al., 2014, https://doi.org/10.1175/MWR-D-13-00064.1, plus the follow-up literature - see also references in Mazzaro et al., 2017) of the cell perturbation method too. Also, to my knowledge, Heinze et al. (2017) used no prescribed boundary conditions as stated in the follow sentence but periodic boundary conditions in combination with a large-scale forcing term inferred from mesoscale model output.

RC2 l54-55: The reference to Heinze et al. (2017) at this point is misleading and not correct. As mentioned before, the study used period BC and the relaxation therein does not refer of a relaxation in space but in time, formulated as a nudging term.

AC  We will add Mirocha et al. [2014] as a reference. Heinze et al. [2017] describes the use of ICON-LES nested in COSMO for realistic simulations over Germany. The main simulation uses prescribed boundary conditions as described in the first paragraph of section 2: *Model description, set-up and simulation output.* For reference they do include results of smaller doubly periodic simulations, but they are only included for validation and are not the main simulation of the paper.
* * *
RC2  Intro: The manuscript would profit if the authors add some more text to introduce the term "open BC" and distinguish it from period boundary conditions with respect to its advantages and disadvantages. For example, also with periodic boundary conditions you can study larger-scale phenomena, even over heterogeneous land surfaces in particular cases.

AC  We will add to the manuscript what type of simulations can be done using periodic boundary conditions and for what type of simulations we need open boundary conditions and why we want to go there to the introduction.
* * *
RC2  l53: What do the authors mean with the term "numerical boundary layer"?

AC  We mean a thin layer upstream of the boundary where wiggles and perturbations are formed as a result from the very strict Dirichlet boundary condition. We will rephrase *Dirichlet boundary conditions are however known to create reflections and a numerical boundary layer at outflow boundaries* to *Dirichlet boundary conditions are however known to create reflections and perturbations at outflow boundaries.*
* * *
RC2  Moreover, a formulation like "often accompanied" is inappropriate here. The authors should be more specific in terms which model uses which strategy to mitigate boundary effects.

AC  The references in brackets indicate the models that report that they used a relaxation/nudging technique. For clarity we will rephrase *The prescribed boundary condition is therefore often accompanied with a relaxation zone (...)* to *For this reason Moeng et al., 2007; Zhu et al., 2010 and Heinze et al., 2017 use a relaxation zone in combination with a prescribed boundary condition,* We will also add this information to the table that describes which model uses which boundary conditions.

**3 Open boundary implementation**

RC1 Eq. (1) does not make sense, since it adds scalar values, like $\partial u_n/\partial t$ or $\epsilon$, and a vector value $\hat{z}$

AC We agree that the notation is wrong. In the revised version we will split the equation for the lateral and top boundaries

$$
\frac{\partial u_n}{\partial t} =
\begin{cases}
-\frac{U}{\rho}\frac{\partial \rho u_n}{\partial n} + \epsilon, & \text{for lateral boundaries} \\
-\frac{U}{\rho}\frac{\partial \rho u_n}{\partial n} + g\frac{\theta - \langle\theta\rangle}{\langle\theta\rangle} + \epsilon, & \text{for top boundary}
\end{cases}
$$

RC1 Line 139: $x_n - \hat{x} \cdot \hat{n}\Delta x_n$ is a location, not a cell.

AC We will change the wording from *the grid cell directly upstream of the boundary* to *the location one gridsize upstream of the boundary*
* * *
RC1 Line 143, *Equation (2) is discretised using a second order forward scheme*: what does it mean exactly? Please provide the expression of the numerical scheme. Idem for the discretisation of (1).

AC Here we made a mistake, this should be a first order upwind scheme and is defined as
$$
\left.\frac{\partial u}{\partial n}\right|_i \approx
\begin{cases}
\frac{u_i - u_{i-1}}{\Delta x_n}, & \text{for } u_B >= 0 \\
\frac{u_{i+1} - u_i}{\Delta x_n}, & \text{for } u_B < 0
\end{cases}
$$

For the time derivative discretisation the third order Runga Kutta method used by DALES [Heus et al., 2010] is used. We will add this information to the revised manuscript.
* * *
RC1 Line 145, *a Dirichlet boundary condition is used for the boundary-normal velocity components*: I do not agree. A Dirichlet boundary condition for the boundary-normal velocity component would read $u_n = u_n^B$. And a Dirichlet boundary condition for the tendency of the boundary-normal velocity component would read $\frac{\partial u_n}{\partial t} = \frac{\partial u_n^B}{\partial t}$. (3) is actually some kind of nudging of $u_n$ towards $u_n^B$, with a relaxation time scale equal to $\Delta t$. Moreover the time discretisation of (3) should also be indicated.

RC2 2.1.2 Inflow: What does it exactly mean that the Dirichlet condition is implemented as a tendency term? Suppose there is a mesoscale model input which changes over time and the LES model is in between 2 mesoscale model timesteps. How exactly are the BCs for the velocity vector and other quantities computed? I guess at the end DALES requires some kind of boundary values for each prognostic quantity for the spatial discretization rather than a tendency term? Moreover, as the authors mentioned

that a tendency term work well with the pressure solver, at what stage are the boundary values imposed, before or after invoking the pressure solver?

AC We agree that the description given by RC1 is more accurate than the current one and we will change *a Dirichlet boundary condition is used for the boundary-normal velocity components* to *the boundary-normal velocity at the boundary $u_n$ is nudged towards the input value $u_n^B$ with a relaxation time scale equal to the integration time scale used by DALES. The discretisation of the time derivative is given by a third-order Runga Kutta scheme used by DALES and is described in Heus et al. [2010].* In the given setup the input was given at the same spatial and temporal resolution as the simulation, so the case described by RC2 where DALES would be in between two input time steps does not occur. However, the current implementation has been used to simulate more realistic cases in which DALES was coupled to a mesoscale model. In this case, the boundary input data is linearly interpolated in time if DALES is in between two mesoscale time steps. This information will be added to the manuscript. Boundary input is required for all the prognostic variables of DALES. In the implementation the tendencies are applied before the pressure solver. The order does however not matter as the pressure solver uses homogeneous Neumann boundary conditions $\frac{\partial p}{\partial n} = 0$ and has therefore no influence on the tendencies of the boundary-normal velocity components at the boundaries.
* * *
RC1 Eq. (4): S(B) is not defined.

RC1 Eq. (5): $S^{\text{int}}$ is not defined. I understand that it is a patch around the boundary, but it should be defined exactly.

RC1 Eq. (6) is definitely unclear to me. Is $\epsilon$ a constant or does it depend on space and time? Is the $\epsilon(S^{\text{int}})$ the same as $\epsilon$? If $\epsilon$ is a constant, (6) is indeed only the time derivative of (5), which does not involve any $\epsilon$. The way $\epsilon$ is actually estimated should be rewritten clearly.

RC2 l170: I disagree with this interpretation. The boundary values enter the equations via the resolved- and subgrid-scale advection terms and not via the pressure term.

AC We agree that the section on the mass correction term $\epsilon$ needs clarity, especially since it's to our best knowledge a new approach. The following adjustments will be made:

- l156 rephrase to *The input boundary-normal velocity components integrated over the lateral and top boundaries $S(B)$ satisfy the continuity equation conform the reference density profile used by DALES.*
- l159 rephrase to *The lateral and top boundaries are subdivided into patches $S^{int}$ defined by $\Delta y^{int}$ and $\Delta z$ for the west and east boundaries, $\Delta x^{int}$ and $\Delta z$ for the north and south boundaries and $\Delta x^{int}$*

and $\Delta y^{int}$ for the top boundary. We enforce that the mass flux integrated over each patch equals the mass flux given by the input velocities integrated over the same patch.

- l162 rephrase to *To obtain the correction factor $\epsilon$, we define $\epsilon$ to be constant (in space) within a single integration patch $S^{int}$, but can differ between patches. To obtain an expression for the correction term on a particular integration patch $\epsilon\left(S^{int}\right)$, we take the time derivative of eq. (5). Further, we define $\frac{\partial \tilde{u}_n}{\partial t} = \frac{\partial \tilde{u}_n}{\partial t} - \epsilon$ as the tendency from either eq. (1) or (3) minus the correction term. Within DALES the tendencies for the boundary normal velocities are first calculated without the correction term. These tendencies are then used to calculate the correction term e using eq. (6) for each integration patch. The correction factor is then added to the tendencies before applying them to make sure mass is conserved.*

- l165-169 rephrase to *The correction factor $\epsilon$ can be physically interpreted as the correction required to force the mass flux through the integration patch $S^{int}$ to the mass flux given by the input. Since the constrain is set on the integrated quantity, fluctuations smaller than the set integration patch are conserved. Smaller values for $\Delta x^{int}$ and $\Delta y^{int}$ impose more strict boundary conditions, with Dirichlet conditions in the limit where $\Delta x^{int} = \Delta x$ and $\Delta y^{int} = \Delta y$. When used in a nested simulation, $\Delta x^{int}$ and $\Delta y^{int}$ could be set to the gridsize used by the mother model. In this setup the total mass flux through a mother cell at the boundary of the child model is conserved, while the child model is free to generate turbulence on smaller scales. This is illustrated in 2D in fig 1 in which the blue cells correspond to the mother model and have a resolution of $\Delta x^{mother}$ and the brown cells to the child mother (DALES).*

- l170-174 rephrase to *The role of the correction term is to conserve mass integrated over the domain, such that the pressure solver, which needs to find a solution that conserves mass locally, can find a solution. It is possible to implement the tendency from the correction factor as a non-homogeneous Neumann boundary condition for the pressure solver, such that all the tendencies as a result of the continuity requirement are together. We chose however, to add the term in the equations for the boundary-normal velocity components and use homogeneous Neumann boundary conditions for the pressure field, because this allows us to keep using the Fourier pressure solver present in DALES [Heus et al., 2010], by using cosine basis functions only.*

RC2 Headings of 2.1 and 2.2: The logical structure is misleading or the heading is poorly phrased. When 2.1 is about boundary-normal velocity components, I would expect that 2.2 is about boundary-parallel components and not about cell centered variables.

[Figure]

Figure 1: 2D illustration of a nested setup in which the integration length scales are set to the gridsize of the mother model. In this setup the mass flux through a mother cell (blue) at the boundary of the child model (brown) is conserved, while the child model is free to generate turbulence on smaller scales.

AC The sections are divided between variables located at the boundary, the boundary-normal velocity components, and variables that are located offset from the boundary, since they differ in the implementation of their boundary conditions. We agree that the tangential velocity components are strictly speaking not cell-centred variables and we will change the heading of 2.2 to *Boundary-tangential velocity components and cell-centered variables.*
* * *
RC2 l190: What does the term "homogeneous Neumann condition" exactly mean? I see it is defined later in Eq. 11, but should be mentioned already when first used.

RC1 Eq. (7): why do you choose a zero normal flux condition at outflow for all variables but $u_n$? You could have made other choices: please elaborate a little bit.

AC The term homogeneous Neumann condition means a zero normal flux condition $\frac{\partial \psi}{\partial n} = 0$, we will include this in the revised manuscript. The choice to use this condition for all variables except for the boundary-normal velocity component is based on the results of Sani and Gresho [1994]. The homogeneous Neumann conditions for the non boundary-normal velocity components at outflow boundaries are chosen, because they are less strict then a condition on the variable itself [Sani and Gresho, 1994]. Therefore, they tend to produce less disturbances. According to Sani and Gresho [1994] and Craske and Van Reeuwijk [2013] setting homogeneous Neumann conditions for the boundary-normal velocity components results in a ill-posed system with fluctuations in the pressure field and is not suited for turbulent flows. Craske and Van Reeuwijk [2013] conclude in their literature review in their introduction that radiation boundary conditions give the least amount of disturbances at outflow boundaries. Sani and Gresho [1994] also find that radiation boundary condition perform the best out of three boundary conditions tested. We will include this in the revised manuscript.
* * *
RC2 l200-201: To my knowledge this is exactly what is done in PALM (see Hellsten et al., 2021; Kadasch et al., 2021) and in WRF (Moeng et al., 2007; Mirocha et al., 2014), which does not seem to cause significant problems in both models. At least the authors should mention this. Furthermore, this raises the need to improve the argumentation why special Robin boundary conditions are required in conjunction to what happens in DALES when large gradients occur at the boundaries.

AC The potential issue of large gradients and tendencies is a result of less strict or "free" outflow boundary conditions. At outflow boundaries the LES can diverge from the mother model due to the radiative and homogeneous

Neumann boundary conditions. When an outflow boundary changes to an inflow boundary, Dirichlet boundary conditions instantly force the solution to be equal to the input. This can result in large tendencies. Palm uses prescribed boundary conditions in their nested setup, which means that at outflow boundaries the solution is also restricted by the prescribed values from the mother model. So the LES is not free to diverge, which means that the issue of large gradients/tendencies is not present. We however want the LES to be free and force it minimally at outflow boundaries. We will include this information in the revised manuscript to clarify the source of this potential problem.
* * *
RC1 Line 206 and Eq. (9), *advection over an inflow boundary nudges the boundary value to a given input value*: this sentence corresponds to the equation

$$\frac{\partial \psi}{\partial t} + u_n \frac{\partial \psi}{\partial n} + \frac{\psi - \psi^B}{\tau} = 0 \tag{1}$$

which is different from what is implemented. Actually (9) corresponds to the nudging inflow condition for $u_n$ (3) (without $\epsilon$, and with a more general relaxation time scale). But since $\psi$ is discretised one half-cell into the domain and not on the boundary, you have to decide what the value of $\psi$ is on the boundary. For this, you assume that $\psi$ is locally transported at speed $u_n$, i.e. $\frac{\partial \psi}{\partial t} = -u_n \frac{\partial \psi}{\partial n}$

RC2 l227: Isn't e usually being defined as the SGS-TKE? If yes, the units do not match (term in brackets needs to be dimensionless). If not, how is a subgrid-velocity being defined? SGS-models usually give estimations for the SGS-TKE but not for the velocities. There are formulations for SGS-velocites (see e.g. Weil et al., 2004; Weil, J.C.; Sullivan, P.P.; Moeng, C.H. The Use of Large-Eddy Simulations in Lagrangian Particle Dispersion Models. J. Atmos. Sci. 2004, 61, 2877–2887), but I have the impression that the authors mean something different.(?)

AC We get the confusion around the description of the origin of the Robin boundary condition. We will change line 206-208 to *To derive the inflow boundary condition, we assume that advection is the only process taking place at the boundary,*

$$\frac{\partial \psi}{\partial t} + u_n \frac{\partial \psi}{\partial n} = 0 \tag{2}$$

*We also impose that the boundary value is nudged towards a given input value $\psi^B$ over a timescale $\tau$,*

$$\frac{\partial \psi}{\partial t} = \frac{\psi - \psi^B}{\tau} \tag{3}$$

*Combining these two constrains gives*

$$-u_n \frac{\partial \psi}{\partial n} = \frac{\psi - \psi^B}{\tau} \tag{4}$$

In the definition for the variable timescale, $e$ is a subgrid velocity scale and not necessarily the TKE. We have used the square root of subgrid TKE as the velocity scale. We will make this clear in the manuscript and to avoid confusion, we will use a different symbol.
* * *
RC2 l244-245: Can the authors please specify if this is their personal experience, or if it is experience deduced from previous studies? To my knowledge, the current state of literature does not support to make such a statement - there exists no extensive quantitative comparison between different methods so far. Also, I strongly doubt that temperature fluctuations give perse a better solution than just adding perturbations onto the velocity components because the physical mechanisms of turbulence development differ and might not fit to the physical setup. For instance, in purely-shear driven flows this can lead to long persisting streak-like structures.

AC This is from personal experience. In our opinion the problem of starting turbulence at the inflow boundary is similar to spinup of turbulence in a periodic simulation. However, since there is no recycling due to the lack of periodicity, spinup time now equates distance from the inflow boundary. In our personal experience of getting turbulence started in neutral periodic cases, small random temperature fluctuations were more efficient. We will clarify that this is our personal experience.
* * *
RC2 Equations - general: punctuation is missing

AC We will address this.

**4 Test case setup description**

RC1 The reference test case is not really described. It is only said that it is a simulation of the development of a dry convective boundary layer, along with a three-line description of the vales of parameters.

RC1 line 271 with periodic boundary conditions: I suppose that periodicity is achieved in the x and y directions, but not in the z direction?

RC2 l265: Can the authors please be more specific? A w* = 1.5m/s can be achieved in different ways, e.g. by altering the surface flux or the boundary-layer depth. What was the prescribed heat flux in the simulations and how was the initial profile of potential temperature being defined?

RC2 l270 and following: If I understand right, you did perform a forcing where the open BC LES is driven by a period LES. In this regard, it is not clear to me how the coupling was realized. Did you take spatially resolved

data, or did you only took horizontal mean profiles? Did you prescribed boundary values at all lateral boundary, i.e. the east, west, north, south and top boundary, or only that the west boundary? I might be wrong, but according to Fig. 3 it looks like you used periodic BC along y. So my question: Does the north/south boundary act as inflow/outflow boundary at the same time? Does the left inflow boundary could be also an outflow boundary (in a CBL with 3m/s mean wind this can happen)? Same with the right "outflow" boundary.

RC2 I strongly recommend the authors revise the setup description and add more details to allow for a better understanding what was done. Furthermore, I am interested how the authors realized the coupling technically (some note in the text might be nice). Was is realized by an offline approach where the data is stored in a separate file or via an MPI coupling strategy between the big-brother and the open-BC simulation?

AC We agree with the referees that information is missing in the description of the reference case. We will include the initial profiles, a table with all the forcings (including the surface heat flux) and height profiles of the potential temperature at different time intervals to show the development over time of the boundary layer. For the simulation with periodic boundary condition periodicity is applied at all lateral boundaries and a no stress boundary condition at the top. For the simulation with open boundary condition, open boundary conditions are used at all lateral boundaries and the top boundary. So there's no periodicity at the north and south boundaries. Instantaneous cross-section output at every time step from the periodic simulation at the boundaries is used as input for the open boundary conditions. For the current setup this means that the west boundary will be mainly inflow and the east boundary outflow. The north and south boundary will be both in-and outflow at different sections of the boundary with cells changing from in-to output with time. The coupling is done offline, with the periodic output being stored and then used for the simulation with open boundary conditions. This information will be added to the manuscript.
* * *
RC2 l263-264: For demonstrating the benefit of a newly developed method it is inappropriate to say that other test cases are not shown because they yield similar good results. Either you have conducted these tests and show some results of them, or you don't. In my opinion, purely neutral tests give different insights in the performance of a method as just a convective case. Same with cloudy boundary layers, where it is not straightforward how cloud prognostic quantities provided by mesoscale scales are treated in the LES at the boundaries.

AC We agree that every test case will show different features of the implementation. However, the goal of this paper is not to test the boundary

conditions in every scenario, but to give a description of the implementation and a first necessary test case that we believe the implementation should pass. The implementation is currently being used in more advanced test cases (such as mesoscale-nested simulations), but we will leave those for future publications. We will adjust any statements or remarks that can not be backed up by the presented test case or state that further testing is required. We will also remove references to cases not presented in the current manuscript.
* * *
RC2 l267: Do the authors have arguments why they used such an anisotropic grid?

AC The anisotropic grid is chosen for computational reasons. More resolution is required in the vertical, since the vertical gradients of the mean temperature, moisture and wind as well as the corresponding turbulent fluxes are much stronger in the vertical than in the horizontal direction. Limiting the horizontal resolution to the same resolution as the vertical would significantly increase the computational costs.
* * *
RC2 l268: Was the dt really fixed to 5s? In a CBL the vertical component can become about 10 m/s. In conjunction with a dz = 20, time steps of 2s would be required to maintain numerical stability of the advection equation.

AC Yes, the time step was fixed at 5s. In these simulations the vertical velocity stays below 5 m/s. For the chosen time integration method (third order Runga Kutta) and the second order central discretisation scheme used, the critical Courant number for one-dimensional advection is $\sqrt{3}$ [Baldauf, 2008] (and not 1). This means that the upper limit for the time step is around $\frac{20}{5}\sqrt{(3)} \approx 7s$, which makes the chosen time step of 5s stable. We however agree that it is close to the critical value and have repeated the simulations for a 2s timestep. The results and figures have been changed with no influence on the conclusions.
* * *
RC1 lines 280 and 306: *boundary conditions* should be *boundary data.*

AC We will change this.

**5 Discussion and presentation of results**

RC1 A better overview of the solution should be given (e.g. some snapshots), and aspects which could have an impact on the performance of the OBCs should be emphasized (e.g. fluctuations in time of the direction - incoming or outgoing - of the flow near the open boundaries).

AC We will add profiles of the potential temperature field at different time intervals to show the evolution of the boundary layer. We will list the expected challenges for the open boundary conditions. These include reflections at outflow boundaries, changing from in to outflow conditions at the north and south boundaries and for the smoothed simulations the generation of turbulence downstream of the inflow boundary.
* * *
RC1 In my opinion, the critical presentation of the numerical results (Section 4) should be improved, and the conclusions should be strengthened.

RC1 All figures visually compare reference fields with other ones obtained in simulations with OBCs, but no difference is never quantified. For instance: *The TKE field near the outflow boundary is not affected by the smoothing* (line 387) , or *the wavelet cross-section remains close to the periodic cross-section* (line 390). Please quantify.

RC1 The objectives should be explained: what do the authors want from the OBCs ? What are the key properties and diagnostics that should not be impacted by open boundaries? In particular, do you expect to reproduce the behavior of the reference solution from a statistical point of view or from a deterministic point of view? What are then the quantitative criteria that will be used to assess the performance of the OBCs?

AC Ideally the boundary conditions have minimal influence on the solution from a statistical viewpoint. We would like the mean field and the turbulence properties such as the length scales and energy distribution to be unaffected by the numerics of the boundary condition implementation. We will highlight this in the description. To condense the information in the 2d panels into more directly quantifiable information, we will integrate the TKE cross-sections over the boundary layer and present them as well. For the panels we will only show the results for the corners and for $\sigma_t = 6\Delta t$ and $\sigma_x = 4\Delta x$. This would give a line for the simulation with open boundary conditions and for the simulation with periodic boundary conditions. Ideally we would like the mean of these lines to be comparable and the variation around the mean to be within the variance of the reference simulation.
* * *
RC2 Fig. 2: It would be easier to understand if you show absolute values rather than differences. Further, did you compute the profiles from the entire xy-domain or did you exclude some areas near the boundaries? In my opinion it does not make much sense to include areas where the flow is potentially affected by the boundaries because this can bias the result, even if the flow features in the interior of the model domain perfectly match.

AC Changing the panels to show absolute values will make it very difficult/impossible to see any differences as they are small compared to the absolute values.

Instead, we will include the absolute profiles separately when we show the development of the periodic simulation in time. The profiles are calculated from the entire xy-domain. We do agree that this can be very strict for the reason you mention. However, since we are looking for differences we wanted to include the boundaries as well.

RC1 Figures 3 to 6: Those figures could be complemented with the difference between the two panels. And the conclusions fully depend on the criteria: do you want a statistical matching or a deterministic matching between the two panels? How could you quantify it?

AC The solutions only need to agree in a statistical point of view. Due to the chaotic nature of the system they differ in the locations of their turbulent structures. We therefore think that showing differences does not show any relevant information. The fact that they are so similar in a deterministic point of view only goes to show how small the influence of the boundary conditions is, but it is not a requirement. We will clarify this in the manuscript.

RC2 l338-339: To thoroughly evaluate this, xz cross-sections are required. It could well be the case the authors just randomly picked a height which is only weakly affected, while other heights show significant up- or down-drafts near the boundaries.

AC We will include xz-cross sections to convince the reader we did not cherry pick a height level.

RC2 l339-340 and Fig. 4: Resolved or subgrid TKE? In the first case, how did you calculate the TKE (formula, time-averaging of the total fluxes, etc.)? In the next sentence you mention that the TKE is averaged over half an hour, which partly answers my question, but I have the impression that the calculation of TKE is not completely correct in this case. According to what you wrote, you computed instantaneous values of TKE from $\sum_i \left\langle (u_i'(t))^2 \right\rangle$ and average these over time. This only works when $u'$ refers to a phase average where homogeneous conditions along y apply. However, if the north/south boundaries are also in/outflow boundaries, this is strictly speaking not the case. Alternatively, $u_i'^2$ can be computed via a time average.

AC That is indeed how we compute what we call the TKE. To be completely correct, we will include the formula how we compute it and not call it the TKE.

RC2 caption Fig. 4: How can a black line indicate a "fixed" ratio? I guess you mean something like ratio between horizontal and vertical advection?

AC We agree that the description is not clear. We will change it to *The slope of the black line is given by the ratio between the horizontal velocity scale and the vertical velocity scale,* $\frac{U}{w^*}$.
* * *
RC2 l352-354: Which data was exactly used for the wavelet analysis? Did you use a spatial or a temporal data series for the wavelet analysis. In the latter case, at which distance from the inflow boundary? Did you use timeseries at at single point of time dependent yz cross section data. Which mother wavelet was employed? More specific information is required.

AC An instantaneous xy-slab has been used to calculate the wavelet analysis. For each x line a (spatial) wavelet analysis is performed. The average of these power spectra is shown. A morlet wavelet is used as the mother wavelet. We will add these specifications to the revised manuscript.
* * *
RC2 355: I do not understand why the analysis window is outside the domain. Actually the hatched area is defined by the cone-of-influence in the wavelet literature, describing the area in the scalogramm which is not affected by boundary effects. The sentence should be rephrased accordingly.

AC The reviewer is right that the hatched area is the cone-of-influence. The cone-of-influence describes the area that is potentially affected by boundary effects. These boundary effects result from the stretched wavelet extending beyond the edges of the domain. That's what we meant with "the analysis window is outside the domain". We will add the above information to the manuscript.
* * *
RC1 Lines 358-360, ... *shows similar results for both simulations... no clear differences visible...*: in my opinion, this is exaggerated. One should better explain why we can consider that the differences are not significant, which again depends on the criteria that have been chosen.

AC We tried to quantify this by using 2.5 upper and lower percentiles. We will use this information more in the text to support our claims. We will also state the criteria more clearly. Which is that they should match statistically. For any wavelength, the power present should be around the mean of the periodic simulation and with similar variation. They don't have to agree deterministically as turbulent structures can be located at different locations.
* * *
RC1 Section 4.1: boundary data are perfect in this experiment, with the same spatial and temporal resolution as the reference simulation. Dirichlet boundary conditions everywhere would therefore give a perfect result. So it is not surprising that the results are good in the vicinity of the inflow boundary. It is what happens near the output that is a priori the most interesting.

RC2 l363 and following: I agree, but this is not surprising as you simply forced an LES with output from another LES under idealized conditions (no changing wind direction, not much change in mass flux, etc.). The authors should put their statements into the context what their test case really shows.

AC It is true that perfect boundary information is given. Dirichlet conditions are however not employed everywhere so the solution is not predetermined and therefore not necessarily the same as the reference simulation. The inflow boundary in this case is indeed the least interesting and we would expect good results there. It is a sanity check that needs to be passed so that we have a good benchmark from which we can degrade by coarsening the temporal and spatial resolution of the input. Also, disturbances from the eastern, north and south boundaries could still propagate upstream and disturb the solution in the interior of the domain. We think this "best-case" scenario is a test that the implementations has to pass, even though it might not look too interesting. We will emphasise that this scenario is designed to test the implementation on the most basic level, as due to the ideal boundary information, any disturbances can only be blamed on the implementation.
* * *
RC1 Several statements are quite obvious: smoothing the input data results in a reduced TKE downwind of the inflow boundary, and deteriorates the solution; adding synthetic turbulence helps to generate developed turbulence faster... Again defining, from the beginning, clear desirable quantitative criteria would help.

AC We agree that a reduced TKE downwind of the inflow boundary is to be expected when smoothing the input fields. The goal of adding synthetic turbulence is to generate turbulence faster, however this does not mean that it would necessarily work. Since it is impossible to add "real" turbulence that the LES agrees with, it would be a realisitic possibility that the perturbations are dampened and wouldn't help. To make this clearer, we have added the following sentence to the text: *The better performance when using synthetic turbulence may appear trivial. However, as we cannot add turbulence that is directly compatible with the LES solution, the synthetic turbulence could be dampened or generate artefacts near the inflow boundary. The fact that it does not, shows the value of using it in our implementation.*

RC1  Lines 385 and 421: it is mentioned that a burst of TKE is observed, but is there an explanation for it?

RC2  l392-397: This is an interesting point because it systematically investigates the overshooting of turbulence also seen in previous studies (Munosz-Esparza and Kosovic, 2018 - https://doi.org/10.1175/MWR-D-18-0077.1 ; Kadasch et al., 2021). I would encourage the authors to also discuss their findings in the context of previous studies.

AC  Will have a look at these studies and compare their results to ours. We notice that the TKE burst roughly happens on the line with a the gradient given by the ratio between the convective (vertical) and horizontal velocity scales. Our hypothesis is therefore that it is a clash between fields that are mainly governed by information supplied at the lateral inflow boundary, which lack developed turbulence and the fields originating from surface convection that do have developed turbulence. We believe that the sudden transition from non turbulent flow to turbulent flow causes an overshoot in TKE. This phenomena is common during the spinup time of (periodic) turbulent simulations. During the first hour the turbulence in the boundary layer needs to build up. Only after this is developed it is capable of transporting the accumulated surface moisture and heat flux through the boundary layer causing a peak in TKE but also in cloud fraction if clouds are formed on the top of the boundary layer [i.e. Siebesma and Cuijpers, 1995]. We will add this explanation to the manuscript.
* * *
RC1  Section 4.3: the goal of this section is not clear to me. Do you expect for the solution to reproduce the reference solution from a deterministic point of view, or to have a correct level of turbulence? The key question is perhaps the following: which scales are closer (in some sense to be defined) to the reference ones when this artificial turbulence is added?

AC  The goal is not to reproduce the reference simulation, as it is not possible to recreate the full turbulence field from the few parameters that the synthetic turbulence routine uses. The goal is to speed up the turbulence generation and therefore shorten the turbulence build up length. Since it was found that the lack of developed turbulence resulted in the distortions found in the previous section we also wanted to see if adding synthetic turbulence is enough to mitigate these. We will describe these goals more clearly.
* * *
RC1  A suggestion: To the best of my knowledge, the introduction of a variable timescale $\tau$ for the inflow condition (Eq. (13)) is something new. In my opinion, this is a possible contribution, that is worth being discussed

and emphasized. In other words, you could discuss more in depth this aspect, by comparing results with a Dirichlet inflow condition on $u^B$ ($\tau_0 = 0$), a Dirichlet condition for the tendency $\frac{\partial \psi}{\partial t} = \frac{\partial \psi^B}{\partial t}$, and intermediate conditions with several values of $\tau_0$ and $p$, including $p = 0$ (fixed timescale $\tau = 2\tau_0$). Relevant diagnostics should make it possible to decide if the time and space variability of the timescale has a significant effect.

AC  We agree that these test would be interesting. However, we are afraid that extensive testing of all these different inflow conditions would distract from the main story. Furthermore, since the condition is developed with time-varying boundaries in mind, a different test case might be more suited for testing the different conditions. The Dirichlet limit is present in the sensitivity profiles.

**References**

Michael Baldauf. Stability analysis for linear discretisations of the advection equation with runge–kutta time integration. *Journal of Computational Physics*, 227(13):6638–6659, 2008. ISSN 0021-9991. doi: https://doi.org/10.1016/j.jcp.2008.03.025.

John Craske and Maarten Van Reeuwijk. Robust and accurate open boundary conditions for incompressible turbulent jets and plumes. *Comput. Fluids*, 86: 284–297, 11 2013. doi: 10.1016/j.compfluid.2013.06.026.

Rieke Heinze, Anurag Dipankar, Cintia Carbajal Henken, Christopher Moseley, Odran Sourdeval, Silke Trömel, Xinxin Xie, Panos Adamidis, Felix Ament, Holger Baars, Christian Barthlott, Andreas Behrendt, Ulrich Blahak, Sebastian Bley, Slavko Brdar, Matthias Brueck, Susanne Crewell, Hartwig Deneke, Paolo Di Girolamo, Raquel Evaristo, Jürgen Fischer, Christopher Frank, Petra Friederichs, Tobias Göcke, Ksenia Gorges, Luke Hande, Moritz Hanke, Akio Hansen, Hans-Christian Hege, Corinna Hoose, Thomas Jahns, Norbert Kalthoff, Daniel Klocke, Stefan Kneifel, Peter Knippertz, Alexander Kuhn, Thriza van Laar, Andreas Macke, Vera Maurer, Bernhard Mayer, Catrin I. Meyer, Shravan K. Muppa, Roeland A. J. Neggers, Emiliano Orlandi, Florian Pantillon, Bernhard Pospichal, Niklas Röber, Leonhard Scheck, Axel Seifert, Patric Seifert, Fabian Senf, Pavan Siligam, Clemens Simmer, Sandra Steinke, Bjorn Stevens, Kathrin Wapler, Michael Weniger, Volker Wulfmeyer, Günther Zängl, Dan Zhang, and Johannes Quaas. Large-eddy simulations over germany using icon: a comprehensive evaluation. *Q. J. Roy. Meteor. Soc.*, 143 (702):69–100, 2017. doi: 10.1002/qj.2947.

Thijs Heus, Chiel van Heerwaarden, Harmen Jonker, A.P. Siebesma, Simon Axelsen, K. Dries, Olivier Geoffroy, A.F. Moene, David Pino Gonzalez, S.R. Roode, and J. Arellano. Formulation of the dutch atmospheric large-eddy simulation (dales) and overview of its applications. *Geosci. Model Dev.*, 3: 415–444, 09 2010. doi: 10.5194/gmd-3-415-2010.

B. Maronga, M. Gryschka, R. Heinze, F. Hoffmann, F. Kanani-Sühring, M. Keck, K. Ketelsen, M. O. Letzel, M. Sühring, and S. Raasch. The parallelized large-eddy simulation model (palm) version 4.0 for atmospheric and oceanic flows: model formulation, recent developments, and future perspectives. *Geoscientific Model Development*, 8(8):2515–2551, 2015. doi: 10.5194/gmd-8-2515-2015.

B. Maronga, S. Banzhaf, C. Burmeister, T. Esch, R. Forkel, D. Fröhlich, V. Fuka, K. F. Gehrke, J. Geletič, S. Giersch, T. Gronemeier, G. Groß, W. Heldens, A. Hellsten, F. Hoffmann, A. Inagaki, E. Kadasch, F. Kanani-Sühring, K. Ketelsen, B. A. Khan, C. Knigge, H. Knoop, P. Krč, M. Kurppa, H. Maamari, A. Matzarakis, M. Mauder, M. Pallasch, D. Pavlik, J. Pfafferott, J. Resler, S. Rissmann, E. Russo, M. Salim, M. Schrempf, J. Schwenkel, G. Seckmeyer, S. Schubert, M. Sühring, R. von Tils, L. Vollmer, S. Ward, B. Witha, H. Wurps, J. Zeidler, and S. Raasch. Overview of the palm model system 6.0. *Geosci. Model Dev.*, 13(3):1335–1372, 2020. doi: 10.5194/gmd-13-1335-2020.

Jeff Mirocha, Branko Kosović, and Gokhan Kirkil. Resolved turbulence characteristics in large-eddy simulations nested within mesoscale simulations using the weather research and forecasting model. *Monthly Weather Review*, 142 (2):806 – 831, 2014. doi: 10.1175/MWR-D-13-00064.1.

R. Sani and P. Gresho. Résumé and remarks on the open boundary condition minisymposium. *Int. J. Numer. Meth. Fl.*, 18:983–1008, 05 1994. doi: 10.1002/fld.1650181006.

A. P. Siebesma and J. W. M. Cuijpers. Evaluation of parametric assumptions for shallow cumulus convection. *Journal of Atmospheric Sciences*, 52(6):650 – 666, 1995. doi: 10.1175/1520-0469(1995)052¡0650:EOPAFS¿2.0.CO;2.

---

## Author Response (AR1)

**Authors point to point response Liqui Lung et al. (2023) - Open Boundary Conditions for Atmospheric Large Eddy Simulations and the Implementation in DALES4.4**

Franciscus Liqui Lung
Christian Jakob
Pier Siebesma
Fredrik Jansson

February 2024

This document contains the point to point responses and a list of changes. Line references in the reviewer comments refer to the original document. Line references made by the authors refer to the track changes document. Text in blue means that it is added, text in red means that it is deleted.

**1 Major comments**

RC1 Several points in the description of the open boundary conditions are unclear or even not mentioned.

RC2 The description of the boundary conditions lacks important information and is partly misleading. For example, the boundary conditions are formulated as tendencies instead of boundary values. However, the boundary value itself is required for the spatial descretization of the advection term, so I recommend to reformulate the equation towards boundary values. Further, the term slab average is not fully defined. It seems to have a different meaning at the outflow boundary compared to the inflow boundary. Moreover, the formula for the time-scale computation seems to be wrong because the second term in Eq. 13 does not become dimensionless.

AC We agree with the concerns that information was missing in the description of the implementation. We have addressed most of the reviewers comments in the *open boundary implementation* section of this document, where the details of the changes can be found, and we believe the description is in a better state now. To summarize; we added information on the required boundary input, corrected where necessary the notation

used in the equations, added the discretisation schemes used, added extra clarification for the different averaging processes (slab vs patches), elaborated on the mass correction process and clarified the derivation of the Robin boundary condition and changed the subgrid velocity scale symbol to avoid confusion with the subfilter TKE. We however do not understand the comment of RC2 on using tendencies instead of boundary values as we do not see a problem with using current time step values to calculate tendencies for the next time step.
* * *
RC1 The reference test case is not sufficiently described, which prevents from really evaluating the performance of the boundary conditions.

RC2 The setup description of the test case lacks important information. Which surface boundary conditions did the authors use (momentum, heat, SGS-TKE, ...), which numerical schemes were applied (pressure solver, advection and time discretization, ...). Moreover, it is not clear to me how the north and south boundaries were treated (period BC vs. inflow/outflow BC?).

AC We agree with the reviewers that the description of the test case lacks information and we have updated the description. The details can be found in the *Test case setup description* section of this document. To summarize; we have included the simulation parameters and forcings and summarized them in a table, described the initial profiles and included a figure that shows the evolution of the profiles during the simulation, added information on how the coupling was done, clarified that all the boundaries are "open" in the open boundary simulation and added references to Heus et al. [2010] where more information can be found on the subgrid and surface schemes and the pressure solver.
* * *
RC1 Many statements seem rather weak, or even quite obvious, in the comments of the simulation results. I think that the conclusions should be strengthened.

AC We clarified the objectives on which we judge the performance of the boundary conditions, which include; that the simulations are evaluated statistically and not deterministically, the addition of synthetic turbulence is not to retrieve the same results as the unsmoothed simulation, but rather to mitigate artifacts as a result of the missing turbulence in the input data. To help quantify the influence of the boundary conditions on turbulence, we have included vertically integrated TKE plots. We have also added a summarizing paragraph in the conclusions. Details of the adjustments can be found in the *Discussion and Presentation of Results* section of this document.
* * *
RC2 The author motivate their study by nesting LES domains into large-scale model domains. It is well known that other LES models which use Dirichlet boundary conditions for time-dependent mesoscale flow inputs sometimes suffer from wave-like structures near the boudaries, so better formulated boundary conditions to overcome this would be highly appreciated. However, as far as I understand, the boundary conditions described herein are only supposed to be used for idealized situations where the inflow and outflow boundary are fixed over the LES simulation period. For example, in a mesoscale-nested simulation, it is likely that the wind speed and direction continuously change in time, meaning that an inflow boundary can become an outflow boundary and so on. While this is still considered in the equations, though not supported by any analysis, the situation where a lateral boundary can become both, inflow and outflow boundary at the same time, is not considered in the equations. For example, this situation can occur if you want to model mesoscale phenomena like sea breezes, local wind systems, convective situations with weak winds, or situations like frontal passages. This is because the radiation boundary condition requires slab averages of the outward-pointing component. If there is a significant inflow at this boundary, the $\langle u_n \rangle$ can become negative. In case this happens, the flow becomes quickly unstable in conjunction with radiation boundary conditions, meaning that the proposed method is only applicable for idealized scenarios. Thus, the use of a slab average actually prohibits that a boundary can be both, inflow and outflow boundary at the same time. I recommend to rephrase the general motivation in this context, in order to avoid the impression that the proposed formulation of the boundary conditions solves the issue in general.

AC Our ultimate goal is to be able to nest DALES in mesoscale models. We agree that mesoscale-nested simulations involve time-varying boundary conditions and this has played an important part in how we defined our boundary conditions. We acknowledge that the presented test setup does not include all the challenges of a mesoscale-nested simulation. However, we believe that the presented setup is a first necessary set of tests that the implementation needs to pass before moving to more complicated test cases in future publications as they may mask basic problems with the open boundary implementation. We are aware of the instabilities that can arise with radiation boundary conditions that use slab averages on time-varying boundaries. This is why we chose not to use slab averages, but instead defined the integration length scales over which we calculate the phase velocity and mass flux correction term. The integration length can conveniently be chosen to be the resolution of the "mother" model. This choice gives maximum freedom to the boundary conditions given the constrains imposed by the mother model (see *boundary implementation* section for more on this). We believe, that the mass correction term plays an important role in preventing any instabilities from building up. From other comments we do realize that the description of this correction

term was not clear and we have elaborated on it in the *open boundary implementation* section. We do not believe the presented implementation can only be used in idealized setups and as this is also not our ultimate goal, we do not want to phrase it this way. The goal to be able to do mesoscale-nested simulations has motivated our implementation choices and we therefore do want to mention it. However, we do agree with the reviewer that the presented test case is not sufficient to claim that the setup will work in a mesoscale-nested setup and we will remove or adjust any such claims (l109-l111) and mention that further testing is required (l691). We also would like to note that although the test case does not represent a changing mesoscale system, the north and south boundaries do vary between in- and outflow locally with time. Since the submission of this manuscript, we have used this implementation to nest DALES in a mesoscale model (with time-varying boundaries), we will leave these results however for later publication.
* * *
RC2  I like the idea of a big-brother simulation to investigate the impact of the open boundary conditions in a systematic manner. However, the performance of the open BC is not sufficiently supported by the test case and the analysis. The authors only used a single setup for a convective boundary layer with a fixed inflow and outflow boundary. However, convection may easily masked systematic effects because instantaneous fluctuations may superimpose weaker systematic biases. For this purpose I think the evaluation of the model need to be extended towards purely neutral flows. Moreover, I think the test scenario should be also extended to a case with changing inflow conditions with respect to the wind speed to i) evaluate the performance of the mass-conservation scheme and ii) to demonstrate that proposed time-dependent relaxation time-scale algorithm works properly. Also a test case with changing wind direction is required to demonstrate that the boundary conditions can also deal with such situations.

AC  The goal of this paper is to describe the current implementation of open boundary conditions in DALES and present a first necessary set of tests. We agree with the reviewer that the proposed cases all test and show different aspects and we have conducted some of them in the past (neutral and mesoscale-nested), however for readability we do not want to include them in the current manuscript and we will leave them for future publications. As mentioned before we will remove any claims that can not be supported by the current test case or state that they require further testing. We will also remove the reference to simulations not presented in the manuscript (l365-367).

**2  Introduction**

RC2  l8: The first part of the sentence sounds strange and should be rephrased.

AC  We rephrased l9; The results show that when the ratio between input and model resolution increases to When smoothing is applied over larger/longer spatial/temporal scales
* * *
RC2  l12: I wouldn't say LES exists to study small scale weather phenomena but would formulate this in a more general way, e.g. to study turbulent motions.

AC  Agree, we rephrased l14; small scale weather phenomena to turbulent motions
* * *
RC2  l25: What do the authors mean by the term "fields"?

AC  We mean the variables and have changed fields into variables in l29.
* * *
RC2  l43-45: It would be useful for the reader if the authors would be more specific, i.e. which model uses which kind of BC. The way the sentence if phrased is too general in my opinion. Also, concerning a description of inflow/outflow BC, the Maronga et al. (2015, https://doi.org/10.5194/gmd-8-2515-2015) paper is more suited reference.

AC  We have added a table with information on the different open boundary condition options in the mentioned models l60. We have also referenced Maronga et al. [2015] (l53). Maronga et al. [2015] describes the fixed in and outflow setting present in PALM 4.0, Maronga et al. [2020] however also describes the new possibility of self-and-rans nesting, for which they use prescribed boundary conditions, so we have referenced both.
* * *
RC2  l48-l49: In addition to the Mazzaro paper it would be nice to add the original literature (Mirocha et al., 2014, https://doi.org/10.1175/MWR-D-13-00064.1, plus the follow-up literature - see also references in Mazzaro et al., 2017) of the cell perturbation method too. Also, to my knowledge, Heinze et al. (2017) used no prescribed boundary conditions as stated in the follow sentence but periodic boundary conditions in combination with a large-scale forcing term inferred from mesoscale model output.

RC2  l54-55: The reference to Heinze et al. (2017) at this point is misleading and not correct. As mentioned before, the study used period BC and the relaxation therein does not refer of a relaxation in space but in time, formulated as a nudging term.

AC  Mirocha et al. [2014] has been added as a reference (l63). Heinze et al. [2017] describes the use of ICON-LES nested in COSMO for realistic simulations over Germany. The main simulation uses prescribed boundary

conditions as described in the first paragraph of section 2: *Model description, set-up and simulation output.* For reference they do include results of smaller doubly periodic simulations, but they are only included for validation and are not the main simulation of the paper. There is another Heinze et al. (2017) paper, however the one referenced here does use prescribed boundary conditions.

RC2 Intro: The manuscript would profit if the authors add some more text to introduce the term "open BC" and distinguish it from period boundary conditions with respect to its advantages and disadvantages. For example, also with periodic boundary conditions you can study larger-scale phenomena, even over heterogeneous land surfaces in particular cases.

AC We have added some text on this at l30-33; Having the ability to use open BCs makes an LES model much more versatile in simulating a range of phenomena, especially over heterogenous terrain. While periodic BCs can sometimes be used to study large-scale phenomena over such terrain, the large domains required to do so quickly become computationally prohibitive.

RC2 l53: What do the authors mean with the term "numerical boundary layer"?

AC We mean a thin layer upstream of the boundary where wiggles and perturbations are formed as a result from the very strict Dirichlet boundary condition. We have changed a numerical boundary layer to perturbations at l67.

RC2 Moreover, a formulation like "often accompanied" is inappropriate here. The authors should be more specific in terms which model uses which strategy to mitigate boundary effects.

AC The references in brackets indicated the models that report that they used a relaxation/nudging technique. For clarity we have rephrased The prescribed boundary condition is therefore often accompanied with a relaxation zone (Moeng et al., 2007; Zhu et al., 2010; Heinze et al., 2017), to For this reason Moeng et al. (2007); Zhu et al. (2010); Heinze et al. (2017) use a relaxation zone in combination with a prescribed boundary condition in l68-71. We have also included this information in the table at l60.

**3   Open boundary implementation**

RC1 Eq. (1) does not make sense, since it adds scalar values, like $\partial u_n / \partial t$ or $\epsilon$, and a vector value $\hat{z}$

AC We agree that the notation is wrong. We have split the equation for the lateral and top boundaries (l150);

$$\frac{\partial u_n}{\partial t} = \begin{cases} -\frac{U}{\rho}\frac{\partial \rho u_n}{\partial n} + \epsilon, & \text{for lateral boundaries} \\ -\frac{U}{\rho}\frac{\partial \rho u_n}{\partial n} + g\frac{\theta - \langle \theta \rangle}{\langle \theta \rangle} + \epsilon, & \text{for top boundary} \end{cases}$$

RC1 Line 139: $x_n - \hat{x} \cdot \hat{n} \Delta x_n$ is a location, not a cell.

AC We changed the wording in l172 from the grid cell directly to location one gridsize.
* * *
RC1 Line 143, *Equation (2) is discretised using a second order forward scheme*: what does it mean exactly? Please provide the expression of the numerical scheme. Idem for the discretisation of (1).

AC Here we made a mistake, this should be a first order upwind scheme and is defined as
$$\left.\frac{\partial u}{\partial n}\right|_i \approx \begin{cases} \frac{u_i - u_{i-1}}{\Delta x_n}, & \text{for } u_B \geq 0 \\ \frac{u_{i+1} - u_i}{\Delta x_n}, & \text{for } u_B < 0 \end{cases}$$
We have corrected this and included the equation in l158-159. For the time derivative discretisation the third order Runga Kutta method used by DALES [Heus et al., 2010] is used. This information has been added in the manuscript at l157; The time derivative is discretised using DALES' third order Runga-Kutta fication scheme (Heus et al., 2010).
* * *
RC1 Line 145, *a Dirichlet boundary condition is used for the boundary-normal velocity components*: I do not agree. A Dirichlet boundary condition for the boundary-normal velocity component would read $u_n = u_n^B$. And a Dirichlet boundary condition for the tendency of the boundary-normal velocity component would read $\frac{\partial u_n}{\partial t} = \frac{\partial u_n^B}{\partial t}$. (3) is actually some kind of nudging of $u_n$ towards $u_n^B$, with a relaxation time scale equal to $\Delta t$. Moreover the time discretisation of (3) should also be indicated.

RC2 2.1.2 Inflow: What does it exactly mean that the Dirichlet condition is implemented as a tendency term? Suppose there is a mesoscale model input which changes over time and the LES model is in between 2 mesoscale model timesteps. How exactly are the BCs for the velocity vector and other quantities computed? I guess at the end DALES requires some kind

of boundary values for each prognostic quantity for the spatial discretization rather than a tendency term? Moreover, as the authors mentioned that a tendency term work well with the pressure solver, at what stage are the boundary values imposed, before or after invoking the pressure solver?

AC We agree that the description given by RC1 is more accurate than the current one and we have changed a Dirichlet boundary condition is used for the boundary-normal velocity components. The Dirichlet condition is implemented as a tendency to work well with the pressure solver used in DALES. to the boundary-normal velocity at the boundary $u_n$ is nudged towards the input value $u_n^B$ with a relaxation time scale equal to the integration time scale used by DALES ($\Delta t$). The discretisation of the time derivative is given by the third-order Runga-Kutta scheme used by DALES (Heus et al., 2010). In the given setup the input was given at the same spatial and temporal resolution as the simulation, so the case described by RC2 where DALES would be in between two input time steps does not occur. However, the current implementation has been used to simulate more realistic cases in which DALES was coupled to a mesoscale model. In this case, the boundary input data is linearly interpolated in time if DALES is in between two mesoscale time steps. This information has been added to the manuscript at l135-137; If the boundary input is not at the same time intervals as the simulation, the input data is linearly interpolated in time to the model time. Boundary input is required for all the prognostic variables of DALES as is mentioned at l132. In the implementation the tendencies are applied before the pressure solver. The order does however not matter as the pressure solver uses homogeneous Neumann boundary conditions $\frac{\partial p}{\partial n} = 0$ and has therefore no influence on the tendencies of the boundary-normal velocity components at the boundaries.
* * *
RC1 Eq. (4): S(B) is not defined.

RC1 Eq. (5): $S^{\text{int}}$ is not defined. I understand that it is a patch around the boundary, but it should be defined exactly.

RC1 Eq. (6) is definitely unclear to me. Is $\epsilon$ a constant or does it depend on space and time? Is the $\epsilon(S^{\text{int}})$ the same as $\epsilon$? If $\epsilon$ is a constant, (6) is indeed only the time derivative of (5), which does not involve any $\epsilon$. The way $\epsilon$ is actually estimated should be rewritten clearly.

RC2 l170: I disagree with this interpretation. The boundary values enter the equations via the resolved- and subgrid-scale advection terms and not via the pressure term.

AC We agree that the section on the mass correction term $\epsilon$ needs clarity, especially since it's to our best knowledge a new approach. The following adjustments have been made:

- To define $S(B)$ we rephrased l194-197; The input boundary normal velocity components satisfy the continuity equation conform to the reference density profile used by DALES, to The input boundary-normal velocity components integrated over the lateral and top boundaries $S(B)$ satisfy the continuity equation conform the reference density profile used by DALES.

- To elaborate on the boundary patches $S^{\text{int}}$ we rephrased l202-206; On boundary patches defined by $\Delta x^{\text{int}}$, $\Delta y^{\text{int}}$, $\Delta z$, the integrated mass flux equals the integrated mass flux given by the input velocities. to The lateral and top boundaries are subdivided into patches $S^{\text{int}}$ defined by $\Delta y^{\text{int}}$ and $\Delta z$ for the west and east boundaries, $\Delta x^{\text{int}}$ and $\Delta z$ for the north and south boundaries and $\Delta x^{\text{int}}$ and $\Delta y^{\text{int}}$ for the top boundary. We enforce that the mass flux integrated over each patch equals the mass flux given by the input velocities integrated over the same patch.

- To clarify how $\epsilon$ is obtained we rephrased l298-215; piecewise continuous on the integration patches. Taking the time derivative of Eq. (8) gives us then an expression for $\epsilon$. to constant (in space) within a single integration patch $S^{\text{int}}$, but can differ between patches. To obtain an expression for the correction term on a particular integration patch $\epsilon\left(S^{\text{int}}\right)$, we take the time derivative of Eq. (8). Further, we define $\frac{\partial \tilde{u}_n}{\partial t} = \frac{\partial \tilde{u}_n}{\partial t} - \epsilon$ as the tendency from either Eq. (2) or (5) minus the correction term. Within DALES the tendencies for the boundary normal velocities are first calculated without the correction term. These tendencies are then used to calculate the correction term $\epsilon$ for each integration patch using Eq. (9). The correction factor is then added to the tendencies before applying them to make sure mass is conserved.

- The interpretation of $\epsilon$ has been rephrased in l217-229; In Eq. (9) $\frac{\partial \tilde{u}_n}{\partial t} = \frac{\partial u_n}{\partial t} - \epsilon$, is the tendency at the boundary without the correction factor. $\epsilon$ can be physically interpreted as the correction required to force the mass flux through the integration area $S^{\text{int}}$ to the mass flux as given by the input. Within one integration patch the mean of the boundary-normal mass flux is forced to the mean of the input mass flux, while the smaller scale perturbations are preserved. Dirichlet conditions are obtained when the integration length scales are set equal to the DALES resolution. to The correction factor $\epsilon$ can be physically interpreted as the correction required to force the mass flux through the integration patch $S^{\text{int}}$ to the mass flux integrated over the patch as given by the input. Since the constrain is set on the integrated quantity, fluctuations smaller than the set integration patch are conserved. Smaller values for $\Delta x^{\text{int}}$ and $\Delta y^{\text{int}}$ impose more strict boundary conditions, with Dirichlet conditions in the limit where $\Delta x^{\text{int}} = \Delta x$ and $\Delta y^{\text{int}} = \Delta y$. When used in a nested simulation, $\Delta x^{\text{int}}$ and $\Delta y^{\text{int}}$ could be set to the gridsize used by the

mother model. In this setup the total mass flux through a mother cell at the boundary of the child model (DALES) is conserved, while the child model is free to generate turbulence on smaller scales. This is illustrated in 2D in Fig. 1 in which the blue cells correspond to the mother model and have a resolution of $\Delta x^{\text{mother}}$ and the brown cells to the child mother (DALES). A illustration has been added on l229 and l230-242 has been rephrased; Since the role of the pressure term in the anelastic approximation is to conserve mass, one can interpret the correction term as a pressure boundary condition. It is possible to use a non-homogeneous Neumann condition for the pressure solver such that the resulting tendency corresponds to the correction term. However, we choose to add the term in the equations for the boundary-normal velocity components and use homogeneous Neumann boundary conditions for the pressure field. This allows us to keep using the Fourier pressure solver, by using cosine basis functions only. to The role of the correction term is to conserve mass integrated over the domain, such that the pressure solver, which needs to find a solution that conserves mass locally, can find a solution. It is possible to implement the tendency from the correction factor as a non-homogeneous Neumann boundary condition for the modified pressure [defined in Heus et al., 2010] $\frac{\partial \pi}{\partial n} = -\epsilon$, such that all the tendencies as a result of the continuity requirement are together. We chose however, to add the term in the equations for the boundary-normal velocity components and use homogeneous Neumann boundary conditions for the modified pressure $\frac{\partial \pi}{\partial n} = 0$, because this allows us to keep using the Fourier pressure solver present in DALES [Heus et al., 2010], by using cosine basis functions only..
* * *
RC2 Headings of 2.1 and 2.2: The logical structure is misleading or the heading is poorly phrased. When 2.1 is about boundary-normal velocity components, I would expect that 2.2 is about boundary-parallel components and not about cell centered variables.

AC The sections are divided between variables located at the boundary, the boundary-normal velocity components, and variables that are located offset from the boundary, since they differ in the implementation of their boundary conditions. We agree that the tangential velocity components are strictly speaking not cell-centred variables and we have rephrased the heading of 2.2 (l248); Cell-centered variables to Boundary-tangential velocity components and cell-centered variables.
* * *
RC2 l190: What does the term "homogeneous Neumann condition" exactly mean? I see it is defined later in Eq. 11, but should be mentioned already when first used.

[Figure]

Figure 1: 2D illustration of a nested setup in which the integration length scales are set to the gridsize of the mother model. In this setup the mass flux through a mother cell (blue) at the boundary of the child model (brown) is conserved, while the child model is free to generate turbulence on smaller scales.

RC1 Eq. (7): why do you choose a zero normal flux condition at outflow for all variables but $u_n$? You could have made other choices: please elaborate a little bit.

AC The term homogeneous Neumann condition means a zero normal flux condition $\frac{\partial \psi}{\partial n} = 0$ and is defined in the referenced equation below the text. We did notice that we use the term in the description of the pressure boundary conditions as well in l235 and added the definition there as well. The following text has been added to elaborate a little bit on the choice in l265-271; The decision to use homogeneous Neumann boundary conditions for all but the boundary-normal velocity components has been based on the results of Sani and Gresho (1994) and Craske and Van Reeuwijk (2013) Sani and Gresho (1994) state that Neumann boundary conditions tend to produce less perturbations in comparison to a boundary condition on the variable itself (Dirichlet). Setting homogeneous Neumann conditions for the boundary-normal velocity components results in a ill-posed system with fluctuations in the pressure field and is not suited for turbulent flows (Sani and Gresho, 1994; Craske and Van Reeuwijk, 2013).
* * *
RC2 l200-201: To my knowledge this is exactly what is done in PALM (see Hellsten et al., 2021; Kadasch et al., 2021) and in WRF (Moeng et al., 2007; Mirocha et al., 2014), which does not seem to cause significant problems in both models. At least the authors should mention this. Furthermore, this raises the need to improve the argumentation why special Robin boundary conditions are required in conjunction to what happens in DALES when large gradients occur at the boundaries.

AC The potential issue of large gradients and tendencies is a result of less strict or "free" outflow boundary conditions. At outflow boundaries the LES can diverge from the mother model due to the radiative and homogeneous Neumann boundary conditions. When an outflow boundary changes to an inflow boundary, Dirichlet boundary conditions instantly force the solution to be equal to the input. This can result in large tendencies. Palm uses prescribed boundary conditions in their nested setup, which means that at outflow boundaries the solution is also restricted by the prescribed values from the mother model. So the LES is not free to diverge, which means that the issue of large gradients/tendencies is not present. In their radiation boundary condition setup the in- and outlfow boundaries are fixed so this problem can't happen either. We however want the LES to be free and force it minimally at outflow boundaries. To clarify that this problem is only present when the solution is allowed to diverge at outflow boundaries (as is the case with radiation boundary conditions and Neumann boundary conditions) we have rephrased l276-282; However, for flows in which boundary cells frequently change between in- and outflow boundaries,such as turbulent flows, Dirichlet boundary conditions can give large gradients over the boundaries which result in extreme tendencies. to

However, for flows in which boundary cells change from in- to outflow boundaries and in which the outflow boundary is free to diverge from the boundary input, Dirichlet boundary conditions can result in large gradients over the boundary when they instantaneously set the value at the boundary to the boundary input value. For models that use radiation boundary conditions, this can result in unrealistic large tendencies at the boundary.
* * *
RC1 Line 206 and Eq. (9), *advection over an inflow boundary nudges the boundary value to a given input value*: this sentence corresponds to the equation

$$\frac{\partial \psi}{\partial t} + u_n \frac{\partial \psi}{\partial n} + \frac{\psi - \psi^B}{\tau} = 0 \tag{1}$$

which is different from what is implemented. Actually (9) corresponds to the nudging inflow condition for $u_n$ (3) (without $\epsilon$, and with a more general relaxation time scale). But since $\psi$ is discretised one half-cell into the domain and not on the boundary, you have to decide what the value of $\psi$ is on the boundary. For this, you assume that $\psi$ is locally transported at speed $u_n$, i.e. $\frac{\partial \psi}{\partial t} = -u_n \frac{\partial \psi}{\partial n}$

RC2 l227: Isn't e usually being defined as the SGS-TKE? If yes, the units do not match (term in brackets needs to be dimensionless). If not, how is a subgrid-velocity being defined? SGS-models usually give estimations for the SGS-TKE but not for the velocities. There are formulations for SGS-velocites (see e.g. Weil et al., 2004; Weil, J.C.; Sullivan, P.P.; Moeng, C.H. The Use of Large-Eddy Simulations in Lagrangian Particle Dispersion Models. J. Atmos. Sci. 2004, 61, 2877–2887), but I have the impression that the authors mean something different.(?)

AC We understand the confusion around the description of the origin of the Robin boundary condition. We haved changed l287-297; We impose that advection over an inflow boundary nudges the boundary value to a given input value $\psi^B$ within a given time scale $\tau$. The tendency at the boundary can be written as:

$$\frac{\partial \psi}{\partial t} = -u_n \frac{\partial \psi}{\partial n} = \frac{\psi^B - \psi}{\tau} \tag{2}$$

Equation (15) to To derive the inflow boundary condition, we assume that advection is the only process taking place at the boundary,

$$\frac{\partial \psi}{\partial t} + u_n \frac{\partial \psi}{\partial n} = 0 \tag{3}$$

We also impose that the boundary value is nudged towards a given input value $\psi^B$ over a timescale $\tau$,

$$\frac{\partial \psi}{\partial t} = \frac{\psi^B - \psi}{\tau} \tag{4}$$

Combining these two constrains gives

$$\frac{\psi^B - \psi}{\tau} + u_n \frac{\partial \psi}{\partial n} = 0 \tag{5}$$

In the definition for the variable timescale, $e$ is a subgrid velocity scale and not necessarily the TKE. We have used the square root of subgrid TKE as the velocity scale. In the revised manuscript we have changed the symbol for the subgrid velocity scale from $e$ to $u_s$ to avoid confusion. We also rephrased l319-322; the subgrid velocity at the boundary. For DALES this can be taken from the TKE subgrid scheme when used. Otherwise an estimate needs to be supplied. to a subgrid velocity scale at the boundary. Here we used the square root of the subgrid turbulent kinetic energy taken from the SFS-TKE scheme used by DALES (Heus et al., 2010). A different estimate can be used as well.
* * *
RC2 l244-245: Can the authors please specify if this is their personal experience, or if it is experience deduced from previous studies? To my knowledge, the current state of literature does not support to make such a statement - there exists no extensive quantitative comparison between different methods so far. Also, I strongly doubt that temperature fluctuations give perse a better solution than just adding perturbations onto the velocity components because the physical mechanisms of turbulence development differ and might not fit to the physical setup. For instance, in purely-shear driven flows this can lead to long persisting streak-like structures.

AC This is from personal experience. In our opinion the problem of starting turbulence at the inflow boundary is similar to spinup of turbulence in a periodic simulation. However, since there is no recycling due to the lack of periodicity, spinup time now equates distance from the inflow boundary. In our personal experience of getting turbulence started in neutral periodic cases, small random temperature fluctuations were more efficient. We have added that this is our personal experience (l344).
* * *
RC2 Equations - general: punctuation is missing

AC We have looked at all the equations and believe they are in order now.

**4   Test case setup description**

RC1 The reference test case is not really described. It is only said that it is a simulation of the development of a dry convective boundary layer, along with a three-line description of the vales of parameters.

RC1 line 271 with periodic boundary conditions: I suppose that periodicity is achieved in the x and y directions, but not in the z direction?

RC2 l265: Can the authors please be more specific? A w* = 1.5m/s can be achieved in different ways, e.g. by altering the surface flux or the boundary-layer depth. What was the prescribed heat flux in the simulations and how was the initial profile of potential temperature being defined?

RC2 l270 and following: If I understand right, you did perform a forcing where the open BC LES is driven by a period LES. In this regard, it is not clear to me how the coupling was realized. Did you take spatially resolved data, or did you only took horizontal mean profiles? Did you prescribed boundary values at all lateral boundary, i.e. the east, west, north, south and top boundary, or only that the west boundary? I might be wrong, but according to Fig. 3 it looks like you used periodic BC along y. So my question: Does the north/south boundary act as inflow/outflow boundary at the same time? Does the left inflow boundary could be also an outflow boundary (in a CBL with 3m/s mean wind this can happen)? Same with the right "outflow" boundary.

RC2 I strongly recommend the authors revise the setup description and add more details to allow for a better understanding what was done. Furthermore, I am interested how the authors realized the coupling technically (some note in the text might be nice). Was is realized by an offline approach where the data is stored in a separate file or via an MPI coupling strategy between the big-brother and the open-BC simulation?

AC We agree with the referees that information is missing in the description of the reference case. We have added information on the initial profiles and forcings l368; The dry convective boundary layer is forced with a constant surface heat flux of $\overline{w'\theta'}_s = 0.115 K m s^{-1}$, a zero surface momentum flux $u^* = 0 m s^{-1}$ and a geostrophic forcing in the east-west direction corresponding to $u_g = 3 m s^{-1}$. The simulation is initialised with an east-west velocity of $U = 3 m s^{-1}$, a north-south velocity of $V = 0 m s^{-1}$ and an initial potential temperature profile that consist of a boundary layer with a temperature of $300 K$, an inversion layer at $950 m$ and an inversion jump of $\Delta \theta = 8 K$ over $120 m$ (linear interpolation between $300 K$ and $308 K$ over $120 m$) with a constant temperature gradient of $\frac{\partial \theta}{\partial z} = 0.003 K m^{-1}$ above.. The initial profiles and the development of the boundary layer over time have also been visualized in the added Fig. 3 (l398). The forcings and setup parameters have been summarized in Table 2 (l398). Information on the coupling process and boundaries has been added at l391-397; The coupling is done offline, which means that the periodic simulation is done first and the boundary output is saved for every time step. This output is then used to force the simulation with open boundary conditions. In this setup the west boundary is (mainly) an inflow boundary, the east boundary (mainly) an outflow and the north and south boundaries will be in- and outflow boundaries changing for each grid cell and with time. The periodic simulation uses periodicity for the lateral boundaries and a

no-stress boundary condition at the top (Heus et al., 2010). The simulation with open boundary conditions uses open boundary conditions for the lateral and top boundaries.
* * *
RC2 l263-264: For demonstrating the benefit of a newly developed method it is inappropriate to say that other test cases are not shown because they yield similar good results. Either you have conducted these tests and show some results of them, or you don't. In my opinion, purely neutral tests give different insights in the performance of a method as just a convective case. Same with cloudy boundary layers, where it is not straightforward how cloud prognostic quantities provided by mesoscale scales are treated in the LES at the boundaries.

AC We agree that every test case will show different features of the implementation. However, the goal of this paper is not to test the boundary conditions in every scenario, but to give a description of the implementation and a first necessary test case that we believe the implementation should pass. The implementation is currently being used in more advanced test cases (such as mesoscale-nested simulations), but we will leave those for future publications. We have removed the statement referencing not-shown cases (l365-367); Different cases such as a cloudy boundary layer and a neutral boundary layer have also been tested, but are not shown here as they show similar performance in the open LBCs.
* * *
RC2 l267: Do the authors have arguments why they used such an anisotropic grid?

AC The anisotropic grid is chosen for computational reasons. More resolution is required in the vertical, since the vertical gradients of the mean temperature, moisture and wind as well as the corresponding turbulent fluxes are much stronger in the vertical than in the horizontal direction. Limiting the horizontal resolution to the same resolution as the vertical would significantly increase the computational costs.
* * *
RC2 l268: Was the dt really fixed to 5s? In a CBL the vertical component can become about 10 m/s. In conjunction with a dz = 20, time steps of 2s would be required to maintain numerical stability of the advection equation.

AC Yes, the time step was fixed at 5s. In these simulations the vertical velocity stays below 5 m/s. For the chosen time integration method (third order Runga Kutta) and the second order central discretisation scheme used, the critical Courant number for one-dimensional advection is $\sqrt{3}$ [Baldauf, 2008] (and not 1). This means that the upper limit for the time step is

around $\frac{20}{5}\sqrt(3) \approx 7s$, which makes the chosen time step of 5s stable. We have redone the simulations with a $2s$ timestep, which did not change the conclusions of the paper. We did see that the optimum timescale for the Robin boundary condition is smaller for the smaller timestep and have included this finding at l471-474; The simulations have also been done with a shorter timestep of 2s, the results for all but the Robin boundary condition time scale remain the same. For the Robin boundary condition the optimum time scale is lower for a shorter time step, which requires further research.

RC1 lines 280 and 306: *boundary conditions* should be *boundary data.*

AC We have replaced it at l404. At line 432 we however believe that *conditions* is more appropriate as it used as a reference to the simulation.

**5 Discussion and presentation of results**

RC1 A better overview of the solution should be given (e.g. some snapshots), and aspects which could have an impact on the performance of the OBCs should be emphasized (e.g. fluctuations in time of the direction - incoming or outgoing - of the flow near the open boundaries).

AC We have added the evolution of the potential temperature, east-west wind velocity, vertical potential temperature flux and east-west wind velocity variance profiles in Fig. 3 (l398). We have also some text on the expected challenges for the Big Brother experiment at l432-444; This setup allows us to investigate the definition and implementation of the boundary conditions. Any disturbances present in the simulation with open boundary conditions must be a direct result of the boundary implementation, as the periodic simulation supplies "perfect" boundary fields. It is a first necessary test that needs to be passed. The challenging areas are mainly the outflow (east) boundary and the north and south boundaries. At the outflow boundary, fields should leave the domain unperturbed and the area affected by reflections upstream of the outflow boundary should be minimal. The north and south boundaries are both in- and outflow boundaries and will therefore challenge the capability of the boundary conditions to switch from in- to outflow in time and space. The results from the simulation with open boundary conditions are compared to the reference case with periodic boundary conditions. We would like the mean field and the turbulence properties such as the length scales and energy distribution to be unaffected by the numerics of the boundary condition implementation. The two simulations don't have to match from a deterministic point of view, as the chaotic nature of the system will result in different placement of eddies between both simulations.

RC1 In my opinion, the critical presentation of the numerical results (Section 4) should be improved, and the conclusions should be strengthened.

RC1 All figures visually compare reference fields with other ones obtained in simulations with OBCs, but no difference is never quantified. For instance: *The TKE field near the outflow boundary is not affected by the smoothing* (line 387) , or *the wavelet cross-section remains close to the periodic cross-section* (line 390). Please quantify.

RC1 The objectives should be explained: what do the authors want from the OBCs ? What are the key properties and diagnostics that should not be impacted by open boundaries? In particular, do you expect to reproduce the behavior of the reference solution from a statistical point of view or from a deterministic point of view? What are then the quantitative criteria that will be used to assess the performance of the OBCs?

AC Ideally the boundary conditions have minimal influence on the solution from a statistical viewpoint. We would like the mean field and the turbulence properties such as the length scales and energy distribution to be unaffected by the numerics of the boundary condition implementation. This information has been added in l432-444 (see statement above) and in l486-487; The simulations don't have to be similar from a deterministic point of view as the smallest differences at the boundaries would result in a different solution due to the chaotic nature of the system. To condense the information in the 2d panels into more directly quantifiable information, we have integrated the TKE cross-sections over the boundary layer and presented and evaluated them as well (Figs. 8, 13 & 18).
* * *
RC2 Fig. 2: It would be easier to understand if you show absolute values rather than differences. Further, did you compute the profiles from the entire xy-domain or did you exclude some areas near the boundaries? In my opinion it does not make much sense to include areas where the flow is potentially affected by the boundaries because this can bias the result, even if the flow features in the interior of the model domain perfectly match.

AC Changing the panels to show absolute values will make it very difficult/impossible to see any differences as they are small compared to the absolute values. Instead, we have referenced to the absolute periodic profiles that are now included in Fig. 3, l449; The profiles for the periodic simulation can be seen in Fig. 3 The profiles are calculated from the entire xy-domain. We do agree that this can be very strict for the reason you mention. However, since we are looking for differences we wanted to include the boundaries as well.
* * *
RC1 Figures 3 to 6: Those figures could be complemented with the difference between the two panels. And the conclusions fully depend on the criteria:

do you want a statistical matching or a deterministic matching between the two panels? How could you quantify it?

AC   The solutions only need to agree in a statistical point of view. Due to the chaotic nature of the system they differ in the locations of their turbulent structures. We therefore think that showing differences does not show any relevant information. The fact that they are so similar in a deterministic point of view only goes to show how small the influence of the boundary conditions is, but it is not a requirement. This criteria has been clarified in l442-444; The profiles for the periodic simulation can be seen in Fig. 3 and l486-487; The simulations don't have to be similar from a deterministic point of view as the smallest differences at the boundaries would result in a different solution due to the chaotic nature of the system
* * *
RC2   l338-339: To thoroughly evaluate this, xz cross-sections are required. It could well be the case the authors just randomly picked a height which is only weakly affected, while other heights show significant up- or downdrafts near the boundaries.

AC   xz-cross sections have been included for all simulations, Figs. 6, 11 & 16.
* * *
RC2   l339-340 and Fig. 4: Resolved or subgrid TKE? In the first case, how did you calculate the TKE (formula, time-averaging of the total fluxes, etc.)? In the next sentence you mention that the TKE is averaged over half an hour, which partly answers my question, but I have the impression that the calculation of TKE is not completely correct in this case. According to what you wrote, you computed instantaneous values of TKE from $\sum_i \left\langle (u_i'(t))^2 \right\rangle$ and average these over time. This only works when $u'$ refers to a phase average where homogeneous conditions along y apply. However, if the north/south boundaries are also in/outflow boundaries, this is strictly speaking not the case. Alternatively, $u_i'^2$ can be computed via a time average.

AC   That is indeed how we compute what we call the TKE. We have added the precise equation used and a comment that it is striclty speaking not exactly equal to the TKE in l496-499; calculating $\frac{1}{2} \left[ \sigma_y^2(u) + \sigma_y^2(v) + \sigma_y^2(w) \right]$ for every time step and averaging it over the last half an hour of the simulation. $\sigma_y^2()$ denotes the variance in the cross-wind $(y)$ direction. This quantity is very close to the definition of turbulent kinetic energy (TKE) and will therefore be referred to as TKE from hereon.
* * *
RC2   caption Fig. 4: How can a black line indicate a "fixed" ratio? I guess you mean something like ratio between horizontal and vertical advection?

AC    We agree that the description is not clear. We have change it to The slope of the solid line corresponds to the ratio of the advective velocity scale $(U = 3ms^{-1})$ and convective velocity scale $(w^* = 1.5ms^{-1})$. We have also changed the description at line 505-506, l567 and l561.

———————————————————————————————————————

RC2  l352-354: Which data was exactly used for the wavelet analysis? Did you use a spatial or a temporal data series for the wavelet analysis. In the latter case, at which distance from the inflow boundary? Did you use timeseries at at single point of time dependent yz cross section data. Which mother wavelet was employed? More specific information is required.

AC    An instantaneous xy-slab has been used to calculate the wavelet analysis. For each x line a (spatial) wavelet analysis is performed. The average of these power spectra is shown. A morlet wavelet is used as the mother wavelet. We have rephrased l520-523 to include this information; A one dimensional wavelet analysis is performed in the along-wind direction at 110m after 6 hours of simulation time. The results are averaged over the cross-wind direction. to A one dimensional wavelet analysis is performed on an instantaneous xy-slab after 6 hours of simulation time. The wavelet analysis is done in the along-wind (x) direction. The results for each along-wind line are averaged over the cross-wind direction. A Morlet wavelet was used as the mother wavelet..

———————————————————————————————————————

RC2  355: I do not understand why the analysis window is outside the domain. Actually the hatched area is defined by the cone-of-influence in the wavelet literature, describing the area in the scalogramm which is not affected by boundary effects. The sentence should be rephrased accordingly.

AC    The reviewer is right that the hatched area is the cone-of-influence. The cone-of-influence describes the area that is potentially affected by boundary effects. These boundary effects result from the stretched wavelet extending beyond the edges of the domain. That's what we meant with "the analysis window is outside the domain". This information has been added by rephrasing l524-528; The hatched area indicates the combination of wavelengths and location for which the analysis window is partly outside of the domain and results within this area should be ignored. to The hatched area indicates the cone of influence (COI), the COI describes the area that is potentially affected by boundary effects. These boundary effects result from the stretched wavelet extending beyond the edges of the domain and results within the COI should therefore be ignored. We have also change the description of Fig. 9; The hatched area is the area where the wavelet window is (partly) outside the domain and should be ignored. to The hatched area is the cone of influence and indicates the area that is potentially affected by boundary effects and results within should be ignored..

RC1 Lines 358-360, ... *shows similar results for both simulations... no clear differences visible...*: in my opinion, this is exaggerated. One should better explain why we can consider that the differences are not significant, which again depends on the criteria that have been chosen.

AC We have clarified the criteria (statistical agreement) in l486-487 The simulations don't have to be similar from a deterministic point of view as the smallest differences at the boundaries would result in a different solution due to the chaotic nature of the system.. We do believe that from a statistical point of view the wavelet analysis shows similar results between the two simulations.

RC1 Section 4.1: boundary data are perfect in this experiment, with the same spatial and temporal resolution as the reference simulation. Dirichlet boundary conditions everywhere would therefore give a perfect result. So it is not surprising that the results are good in the vicinity of the inflow boundary. It is what happens near the output that is a priori the most interesting.

RC2 l363 and following: I agree, but this is not surprising as you simply forced an LES with output from another LES under idealized conditions (no changing wind direction, not much change in mass flux, etc.). The authors should put their statements into the context what their test case really shows.

AC It is true that perfect boundary information is given. Dirichlet conditions are however not employed everywhere so the solution is not predetermined and therefore not necessarily the same as the reference simulation. The inflow boundary in this case is indeed the least interesting and we would expect good results there. It is a sanity check that needs to be passed so that we have a good benchmark from which we can degrade by coarsening the temporal and spatial resolution of the input. Also, disturbances from the eastern, north and south boundaries could still propagate upstream and disturb the solution in the interior of the domain. We think this "best-case" scenario is a test that the implementations has to pass, even though it might not look too interesting. We have added l432-441 to emphasise that this scenario is designed to test the implementation on the most basic level, as due to the ideal boundary information, any disturbances can only be blamed on the implementation; This setup allows us to investigate the definition and implementation of the boundary conditions. Any disturbances present in the simulation with open boundary conditions must be a direct result of the boundary implementation, as the periodic simulation supplies "perfect" boundary fields. It is a first necessary test that needs to be passed. The challenging areas are mainly the

outflow (east) boundary and the north and south boundaries. At the outflow boundary, fields should leave the domain unperturbed and the area affected by reflections upstream of the outflow boundary should be minimal. The north and south boundaries are both in- and outflow boundaries and will therefore challenge the capability of the boundary conditions to switch from in- to outflow in time and space. The results from the simulation with open boundary conditions are compared to the reference case with periodic boundary conditions.
* * *
RC1 Several statements are quite obvious: smoothing the input data results in a reduced TKE downwind of the inflow boundary, and deteriorates the solution; adding synthetic turbulence helps to generate developed turbulence faster... Again defining, from the beginning, clear desirable quantitative criteria would help.

AC We agree that a reduced TKE downwind of the inflow boundary is to be expected when smoothing the input fields. The goal of adding synthetic turbulence is to generate turbulence faster, however this does not mean that it would necessarily work. Since it is impossible to add "real" turbulence that the LES agrees with, it would be a realisitic possibility that the perturbations are dampened and wouldn't help. To make this clearer, we have added the following l645-648; The better performance when using synthetic turbulence may appear trivial. However, as we cannot add turbulence that is directly compatible with the LES solution, the synthetic turbulence could be dampened or generate artefacts near the inflow boundary. The fact that it does not, shows the value of using it in our implementation.. We have also rephrased the introduction for the simulation with added synthetic turbulence to better state what the goals are (l608-612); This section will explore the potential of a synthetic turbulence algorithm to mitigate the wave structures found in Sect. 4.2. to The previous section has highlighted significant issues at the inflow boundary when the boundary values are smoothed in space and/or time, resulting in a more laminar flow near that boundary. A potential approach to reduce these issues (Smirnov et al., 2001) is to add synthetic turbulence to the boundary values. The purpose of this section is to investigate how the results in our simulations are affected by doing so.
* * *
RC1 Lines 385 and 421: it is mentioned that a burst of TKE is observed, but is there an explanation for it?

RC2 l392-397: This is an interesting point because it systematically investigates the overshooting of turbulence also seen in previous studies (Munosz-Esparza and Kosovic, 2018 - https://doi.org/10.1175/MWR-D-18-0077.1 ; Kadasch et al., 2021). I would encourage the authors to also discuss their findings in the context of previous studies.

AC  The studies listed by RC2 have been included and we have included our hypothesis in l563-572; This burst in TKE was also found by Muñoz-Esparza and Kosovic (2018) and Kadasch et al. (2021). Our hypothesis is that the burst in TKE is a result of the clash between non turbulent fields that are mainly governed by information supplied at the lateral inflow boundary and turbulent fields originating from surface convection. We believe that the sudden transition from non turbulent flow to turbulent flow causes an overshoot in TKE. This phenomena is also seen during the spinup time of (periodic) turbulent simulations. During the first hour the turbulence in the boundary layer needs to build up. Only after this is developed it is capable of transporting the accumulated surface moisture and heat flux through the boundary layer causing a peak in TKE but also in cloud fraction if clouds are formed on the top of the boundary layer (e.g. Siebesma and Cuijpers, 1995).
* * *
RC1  Section 4.3: the goal of this section is not clear to me. Do you expect for the solution to reproduce the reference solution from a deterministic point of view, or to have a correct level of turbulence? The key question is perhaps the following: which scales are closer (in some sense to be defined) to the reference ones when this artificial turbulence is added?

AC  The goal is not to reproduce the reference simulation, as it is not possible to recreate the full turbulence field from the few parameters that the synthetic turbulence routine uses. The goal is to speed up the turbulence generation and therefore shorten the turbulence build up length. Since it was found that the lack of developed turbulence resulted in the distortions found in the previous section we also wanted to see if adding synthetic turbulence is enough to mitigate these. The introduction of this section has been rephrased to clarify the goals better l608-612 (see two ACs above).
* * *
RC1  A suggestion: To the best of my knowledge, the introduction of a variable timescale $\tau$ for the inflow condition (Eq. (13)) is something new. In my opinion, this is a possible contribution, that is worth being discussed and emphasized. In other words, you could discuss more in depth this aspect, by comparing results with a Dirichlet inflow condition on $u^B$ ($\tau_0 = 0$), a Dirichlet condition for the tendency $\frac{\partial \psi}{\partial t} = \frac{\partial \psi^B}{\partial t}$, and intermediate conditions with several values of $\tau_0$ and $p$, including $p = 0$ (fixed timescale $\tau = 2\tau_0$). Relevant diagnostics should make it possible to decide if the time and space variability of the timescale has a significant effect.

AC  We agree that these test would be interesting. However, we are afraid that extensive testing of all these different inflow conditions would distract from the main story. Furthermore, since the condition is developed with time-varying boundaries in mind, a different test case might be more suited

for testing the different conditions. The Dirichlet limit is present in the sensitivity profiles.

**6    List of changes**

l9-10  The results show that when the ratio between input and model resolution increases to When smoothing is applied over larger/longer spatial/temporal scales

l14-15  small scale weather phenomena to turbulent motions

l29-30  fields at inflow boundaries and propagate fields to variables at inflow boundaries and propagate variables

l30-33  Having the ability to use open BCs makes an LES model much more versatile in simulating a range of phenomena, especially over heterogenous terrain. While periodic BCs can sometimes be used to study large-scale phenomena over such terrain, the large domains required to do so quickly become computationally prohibitive.

l53  (Maronga et al., 2020) to (Maronga et al., 2015, 2020)

Table 1  Added Table 1.

l63  Mirocha et al.,2014

l67  a numerical boundary layer to perturbations

l68-71  The prescribed boundary condition is therefore often accompanied with a relaxation zone (Moeng et al., 2007; Zhu et al., 2010; Heinze et al., 2017) to For this reason Moeng et al. (2007); Zhu et al. (2010); Heinze et al. (2017) use a relaxation zone in combination with a prescribed boundary condition

l109  , in an idealized setup,

l110-111  emulating a setup where the LES is coupled to to as one would encounter when embedding the LES in

l112-113  potential future

l115  coupling to nesting

l132  at the resolution of the simulation.

l135-137  If the boundary input is not at the same time intervals as the simulation, the input data is linearly interpolated in time to the model time

Eq. (1)  Corrected notation

l153-154  $\hat{z}$ the unit vector in the vertical direction,

| | |
|---|---|
| l156 | at the top boundary |
| l157-158 | The time derivative is discretised using DALES' third order Runga-Kutta scheme (Heus et al., 2010). The spatial derivative is discretised using a first order upwind scheme. |
| Eq. (3) | Added spatial discretisation scheme. |
| l172 | the grid cell directly to location one gridsize |
| 174-175 | $\Delta x^{\text{int}}$(north and south) boundaries) or $\Delta y^{\text{int}}$ (west and east boundaries) |
| l177-178 | second order forward scheme to first order upwind scheme Eq. (3). |
| l180-l184 | a Dirichlet boundary condition is used for the boundary-normal velocity components. The Dirichlet condition is implemented as a tendency to work well with the pressure solver used in DALES. to the boundary-normal velocity at the boundary $u_n$ is nudged towards the input value $u_n^B$ with a relaxation time scale equal to the integration time scale used by DALES ($\Delta t$). The discretisation of the time derivative is given by the third-order Runga-Kutta scheme used by DALES (Heus et al., 2010). |
| l186 | In Eq. (5) $\Delta t$ is the integration time step of DALES. |
| l194-l197 | The input boundary normal velocity components satisfy the continuity equation conform to the reference density profile used by DALES, to The input boundary-normal velocity components integrated over the lateral and top boundaries $S(B)$ satisfy the continuity equation conform the reference density profile used by DALES. |
| Eq. (6) | Changed closed integral sign to open integral sign |
| l202-206 | On boundary patches defined by $\Delta x^{\text{int}}, \Delta y^{\text{
[revised manuscript text omitted]

l323 $e$ to $u_s$

l326 Eq. (22)

Eq. (21) e to $u_s$

l331 Eq. (24)

Eq. (23) $e$ to $u_s$

l344 personal

l365-367 Different cases such as a cloudy boundary layer and a neutral boundary layer have also been tested, but are not shown here as they show similar performance in the open LBCs.

l368-373 of $\overline{w'\theta'}_s = 0.115 Kms^{-1}$, a zero surface momentum flux $u^* = 0ms^{-1}$ and a geostrophic forcing in the east-west direction corresponding to $u_g = 3ms^{-1}$. The simulation is initialised with an east-west velocity of $U = 3ms^{-1}$, a north-south velocity of $V = 0ms^{-1}$ and an initial potential temperature profile that consist of a boundary layer with a temperature of $300K$, an inversion layer at $950m$ and an inversion jump of $\Delta\theta = 8K$ over $120m$ (linear interpolation between $300K$ and $308K$ over $120m$) with a constant temperature gradient of $\frac{\partial\theta}{\partial z} = 0.003 Km^{-1}$ above.

l373 that to This

l373-374 The simulations are initialized with a mean advection velocity of $U = 3ms^{-1}$ and a boundary layer height of $z_i = 900m$.

l377-385 For the advection of all variables DALES' second order central scheme was used [Heus et al., 2010]. This setup is very close to the dry (strong) convective boundary layer shown in Heus et al. [2010], which was already studied by Sullivan et al. (1998). The differences are the addition of a mean background wind, a weaker surface heat flux, a higher horizontal resolution, the use of second-order advection schemes and a fixed integration time step. The initial profiles and the evolution over time of the potential temperature, east-west wind velocity, potential temperature flux and east-west wind variance are shown in Fig. 3.

Table 2 Added table with simulation settings and forcings

Figure 3 Added figure with initial profiles and development over time for potential temperature, east-west velocity, vertical potential temperature flux and east-west wind velocity variance.

l391-3397 The coupling is done offline, which means that the periodic simulation is done first and the boundary output is saved for every time step. This output is then used to force the simulation with open boundary conditions. In this setup the west boundary is (mainly) an inflow boundary, the east boundary (mainly) an outflow and the north and south boundaries will be in- and outflow boundaries changing for each grid cell and with time. The periodic simulation uses periodicity for the lateral boundaries and a no-stress boundary condition at the top [Heus et al., 2010]. The simulation with open boundary conditions uses open boundary conditions for the lateral and top boundaries.

l404 conditions to data

l432-444 This setup allows us to investigate the definition and implementation of the boundary conditions. Any disturbances present in the simulation with open boundary conditions must be a direct result of the boundary implementation, as the periodic simulation supplies "perfect" boundary fields. It is a first necessary test that needs to be passed. The challenging areas are mainly the outflow (east) boundary and the north and south boundaries. At the outflow boundary, fields should leave the domain unperturbed and the area affected by reflections upstream of the outflow boundary should be minimal. The north and south boundaries are both in- and outflow boundaries and will therefore challenge the capability of the boundary conditions to switch from in- to outflow in time and space. The results from the simulation with open boundary conditions are compared to the reference case with periodic boundary conditions. We would like the mean field and the turbulence properties such as the length scales and energy distribution to be unaffected by the numerics of the boundary condition implementation. The two simulations don't have to match from a deterministic point of view, as the chaotic nature of the system will result in different placement of eddies between both simulations.

l449-450 The profiles for the periodic simulation can be seen in Fig. 3.

l453 (within 1%)

l471-474 The simulations have also been done with a shorter timestep of $2s$, the results for all but the Robin boundary condition time scale remain the same. For the Robin boundary condition the optimum time scale is lower for a shorter time step, which requires further research.

l475-477 Figure 5 shows a top view of the potential temperature perturbation with respect to the periodic slab average. to Figures 5 and 6 show a top (xy)

view at $110m$ and a side (xz) view of the potential temperature respectively. The top view is shown as a perturbation with respect to the periodic slab average.

Figure 5  Added location of xz cross section

Figure 6  Added figure with xz cross section

l478  at $110m$ height

l479-487  The location of the xz cross-section within the xy cross-section (and vice versa) is shown by the dashed line. The slope of the solid line in the xz cross-section of the simulation with open boundary conditions corresponds to the ratio of the advective velocity scale ($U = 3ms^{-1}$) and convective velocity scale ($w^* = 1.5ms^{-1}$). Left (upstream) of this line, fields will be mainly dominated by information advected from the inflow boundary, whereas right of the line (downstream) the fields will be mainly influenced by convection originating from the surface boundary. The cross-sections are used to visually inspect the results to see if there are any discrepancies in the mean fields or turbulent structures. The simulations don't have to be similar from a deterministic point of view as the smallest differences at the boundaries would result in a different solution due to the chaotic nature of the system.

l496-500  comparing turbulent kinetic energy (TKE) cross-sections between the two simulations (Fig. 7). to calculating $\frac{1}{2}\left[\sigma_y^2\left(u\right) + \sigma_y^2\left(v\right) + \sigma_y^2\left(w\right)\right]$ for every time step and averaging it over the last half an hour of the simulation. $\sigma_y^2()$ denotes the variance in the cross-wind ($y$) direction. This quantity is very close to the definition of turbulent kinetic energy (TKE) and will therefore be referred to as TKE from hereon. Figure 7 shows cross-sections of TKE for the periodic and open boundary condition experiments.

l502-503  after 6 hours of simulation time. The TKE sections are averaged over the last half hour of the simulation and are derived from the velocity perturbations with respect to the cross-wind line averages.

l505-507  The black line shows the $U/w^*$ ratio to The slope of the solid black line corresponds to the ratio of the advective velocity scale ($U = 3ms^{-1}$ and the convective velocity scale ($w^* = 1.5ms^{-1}$)

Figure 7  The black line indicates the $U/w^*$ ratio to The slope of the solid line corresponds to the ratio of the advective velocity scale ($U = 3ms^{-1}$) and convective velocity scale ($w^* = 1.5ms^{-1}$)

[revised manuscript text omitted]

---

## Referee Report (RR1)

**Review of the revised version of the paper Open Boundary Conditions for Atmospheric Large Eddy Simulations and the Implementation in DALES4.4 by Franciscus Liqui Lung, Christian Jakob, A. Pier Siebesma, and Fredrik Jansson, submitted to GMD**

In this revised version, I found that the authors have answered many questions and remarks made by the reviewers. The descriptions of the boundary conditions as well as the configuration of the simulations are much more detailed, and the analysis of the simulation results has clearer objectives and is enriched with some additional figures and comments. As a whole, I found the paper significantly improved and I recommend its publication in GMD.

A few remaining typos or details:

- Eq. (2): it's  $\partial u_n / \partial n$  or  $\partial u_n / \partial x$ , but not  $\partial u_n / \partial t$
- Line 137: not exactly. This condition is perfect only for waves in the 1-D case, and for  $\epsilon = 0$  (see Blayo and Debreu, 2005).
- Line 147-148: a clearer description could be

$$U^* = U|_{x_n - \dots}^{t - \Delta t} = \dots \qquad \text{then } U = \begin{cases} u_n^B & \text{if } |U^*| \le |u_n^B| \\ U^* & \text{if } |u_n^B| \le |U^*| \le \frac{\Delta x_n}{\Delta t} \\ \operatorname{sign}(U^*) \frac{\Delta x_n}{\Delta t} & \text{if } |U^*| \ge \frac{\Delta x_n}{\Delta t} \end{cases}$$

- Eq. (4): remove the comma
- Line 167: conform  $\underline{to}$
- Line 216: there is a point missing at the end of the line.
- Line 219: an ill-posed...
- Line 294: consists

---

## Editor Decision (ED1)

**Review of the revised version of the paper**
*Open Boundary Conditions for Atmospheric Large Eddy Simulations and the Implementation in DALES4.4*
**by Franciscus Liqui Lung, Christian Jakob, A. Pier Siebesma, and Fredrik Jansson, submitted to** *GMD*

In this revised version, I found that the authors have answered many questions and remarks made by the reviewers. The descriptions of the boundary conditions as well as the configuration of the simulations are much more detailed, and the analysis of the simulation results has clearer objectives and is enriched with some additional figures and comments. As a whole, I found the paper significantly improved and I recommend its publication in GMD.

A few remaining typos or details:

- Eq. (2): it's $\partial u_n/\partial n$ or $\partial u_n/\partial x$, but not $\partial u_n/\partial t$

- Line 137: not exactly. This condition is perfect only for waves in the 1-D case, and for $\epsilon = 0$ (see Blayo and Debreu, 2005).

- Line 147-148: a clearer description could be

$$U^* = U|_{x_n - \ldots}^{t - \Delta t} = \ldots \qquad \text{then } U = \begin{cases} u_n^B & \text{if } |U^*| \leq |u_n^B| \\ U^* & \text{if } |u_n^B| \leq |U^*| \leq \frac{\Delta x_n}{\Delta t} \\ \text{sign}(U^*)\frac{\Delta x_n}{\Delta t} & \text{if } |U^*| \geq \frac{\Delta x_n}{\Delta t} \end{cases}$$

- Eq. (4): remove the comma

- Line 167: conform to

- Line 216: there is a point missing at the end of the line.

- Line 219: an ill-posed...

- Line 294: consists

---

## Author Response (AR2)

**List of changes**

- L54: mother to parent
- L99: big-brother to Big Brother
- L103: mother to parent
- Eq. (2): $\frac{\partial u_n}{\partial t}$ to $\frac{\partial u_n}{\partial n}$ (in response to reviewer)
- L137: The 1-D case of Eq. (1) without the correction factor $\epsilon$ (in response to reviewer)
- Eq. (3) Changed according to advice of reviewer
- Eq. (4): removed , (in response to reviewer)
- L167 conform to (in response to reviewer)
- Figure 1: mother to parent
- L184: mother to parent (2x)
- L186: mother to parent
- L186: $\Delta x^{mother}$ to $\Delta x^{parent}$
- L187: mother to model
- L216: (2013). (in response to reviewer)
- L219: an (in response to reviewer)
- L259: by Eq. (17) (added space between by and Eq)
- L262: by Eq. (18) (added space between by and Eq)
- L294: consist to consists (in response to reviewer)
- L305: Big-Brother to Big Brother
- L324: mother to parent
- L328: mother to parent
- L364: wihtin to within
- We did not change mother in mother wavelet as mother and father wavelet are used to refer to different functions